# Perfect Sampling from Pairwise Comparisons

**Dimitris Fotakis**
NTUA
fotakis@cs.ntua.gr

**Alkis Kalavasis**
NTUA
kalavasisalkis@mail.ntua.gr

**Christos Tzamos**
UW Madison
tzamos@wisc.edu

## Abstract

In this work, we study how to efficiently obtain perfect samples from a discrete distribution $\mathcal{D}$ given access only to pairwise comparisons of elements of its support. Specifically, we assume access to samples $(x, S)$, where $S$ is drawn from a distribution over sets $\mathcal{Q}$ (indicating the elements being compared), and $x$ is drawn from the conditional distribution $\mathcal{D}_S$ (indicating the winner of the comparison) and aim to output a clean sample $y$ distributed according to $\mathcal{D}$. We mainly focus on the case of pairwise comparisons where all sets $S$ have size 2. We design a Markov chain whose stationary distribution coincides with $\mathcal{D}$ and give an algorithm to obtain exact samples using the technique of Coupling from the Past. However, the sample complexity of this algorithm depends on the structure of the distribution $\mathcal{D}$ and can be even exponential in the support of $\mathcal{D}$ in many natural scenarios. Our main contribution is to provide an efficient exact sampling algorithm whose complexity does not depend on the structure of $\mathcal{D}$. To this end, we give a parametric Markov chain that mixes significantly faster given a good approximation to the stationary distribution. We can obtain such an approximation using an efficient learning from pairwise comparisons algorithm (Shah et al., JMLR 17, 2016). Our technique for speeding up sampling from a Markov chain whose stationary distribution is approximately known is simple, general and possibly of independent interest.

## 1 Introduction

Machine Learning questions dealing with pairwise comparisons (PCs) constitute a fundamental and broad research area with multiple real-world applications [SBB$^+$16, Kaz11, LOAF12, SBGW16, FH10, WJJ13]. For instance, the preference of a consumer to choose one product over another constitutes a pairwise comparison between the two products. The two bedrocks of modern ML applications are inference/learning and sampling. The area of learning from pairwise comparisons has a vast amount of work [RA14, SBGW16, DKNS01, NOS12, Hun04, JLYY11, APA18, LSR21, HOX14, SBB$^+$16, VY16, NOS17, VYZ20]. In this problem, there exist $n$ alternatives and an unknown weight vector $\mathcal{D} \in \mathbb{R}^n$ where $\mathcal{D}(i)$ is the weight of the $i$-th element with $\mathcal{D}(i) \geq 0$ and $\sum_{i \in [n]} \mathcal{D}(i) = 1$, i.e., $\mathcal{D}$ is a probability distribution over $[n]$. Roughly speaking, the learner observes pairwise comparisons (or more generally $k$-wise comparisons) of the form $(w, \{u, v\}) \in [n] \times [n]^2$ that compare the alternatives $u \neq v$ and $w$ equals either $u$ or $v$ indicating the winner of the comparison. The most fundamental distribution over pairwise comparisons is the BTL model [BT52, Luc59] where the probability that the learner observes $w = u$ when the items $u, v$ are compared is $\mathcal{D}(u)/(\mathcal{D}(u) + \mathcal{D}(v))$. The learner's goal is mainly to estimate the underlying distribution $\mathcal{D}$ using a small number of pairwise comparisons.

In this work, we focus on questions concerning sampling, the second bedrock of modern ML. In fact, we will deal with a fundamental area of the sampling literature, namely *perfect sampling*. In contrast to approximate sampling, perfect (or exact) sampling from a probabilistic model requires an exact sample from this model. Problems in this vein of research lie in the intersection of computer science and probability theory and have been extensively studied [PW96, PW98, Hub04, Hub16, FGY19,

JSS21, AJ21, GJL19, BC20, HSW21]. Our work poses questions concerning perfect sampling in the area of pairwise comparisons. We ask the following simple yet non-trivial question: Is it possible to obtain a perfect sample $x$ distributed as in $\mathcal{D}$ by observing only pairwise comparisons from $\mathcal{D}$?

Apart from the algorithmic and mathematical challenge of generating exact samples, perfect simulation has also practical motivation, noticed in various previous works [Hub16, JSS21, AJ21]. Crucially, its value does not stem from the fact that the output of the sampling distribution is perfect; usually, one can come up with a Markov chain whose evolution rapidly converges to the stationary measure. Hence, the distance between the (approximate) sampling distribution from the desired one can become small efficiently. This approximate sampling approach has a clear drawback: there is no termination criterion, i.e., the Markov chain has to be run for sufficiently long time and this requires an a priori known bound on the chain's mixing time, which may be much larger than necessary. On the contrary, perfect sampling algorithms come with a stopping rule and this termination condition can be attained well ahead of the worst-case analysis time bounds in practice.

Second, the question that we pose appears to have links with the important literature of truncated statistics [Gal97, DGTZ18]. Truncation is a common biasing mechanism under which samples from an underlying distribution $\mathcal{D}$ over $\mathbb{X}$ are only observed if they fall in a given set $S \subseteq \mathbb{X}$, i.e., one only sees samples from the conditional distribution $\mathcal{D}_S$. There exists a vast amount of applications that involve data censoring (that go back to at least [Gal97]) where one does not have direct sample access to $\mathcal{D}$. Recent work in this literature [FKT20] asked when it is possible to obtain perfect samples from some specific truncated discrete distributions (namely, truncated Boolean product distributions). The area of pairwise comparisons can be identified as an extension of the truncated setting where the truncation set can change (i.e., each subset of alternatives corresponds to some truncation set); while perfect sampling has been a subject of interest in truncated statistics, the neighboring area of pairwise comparisons lacks of efficient algorithms for this important question.

**Problem Formulation.**    Let us define our setting formally (which we call Local Sampling Scheme). This setting is standard (without this terminology) and can be found e.g., at [RA14].

**Definition 1** ([RA14]). *Let $\mathcal{Z}$ be a finite discrete domain. Consider a target distribution $\mathcal{D}$ supported on $\mathcal{Z}$ and a distribution $\mathcal{Q}$ supported on subsets of $\mathcal{Z}$. The sample oracle $\mathrm{Samp}(\mathcal{Q}; \mathcal{D})$ is called* Local Sampling Scheme (LSS) *and each time it is invoked, it returns an example $(x, S)$ such that: (i) the set $S \subseteq \mathcal{Z}$ is drawn from $\mathcal{Q}$ and (ii) $x$ is drawn from the conditional distribution $\mathcal{D}_S$, where $\mathcal{D}_S(x) = \mathcal{D}(x)\mathbf{1}\{x \in S\}/\mathcal{D}(S)$ and $\mathcal{D}(S) = \sum_{x \in S} \mathcal{D}(x)$.*

For simplicity, we mostly focus on the case where $\mathcal{Q}$ is supported on a subset of $\mathcal{Z} \times \mathcal{Z}$, and often refer to $\mathcal{Q}$ as the *pair distribution*. Then, $\mathcal{Q}$ naturally induces an (edge weighted) undirected graph $G_{\mathcal{Q}}$ on $\mathcal{Z}$, where $\{u, v\} \in E(G_{\mathcal{Q}})$, if $\{u, v\}$ is supported on $\mathcal{Q}$, and has weight equal to the probability $\mathcal{Q}(u, v)$ that $\{u, v\}$ is drawn from $\mathcal{Q}$. When we deal with a pair distribution $\mathcal{Q}$, the above generative model lies in the heart of the pairwise comparisons [FV86, Mar96, WJJ13]. The comparisons are generated according to the Bradley-Terry-Luce (BTL) model (where the comparisons have size 2, [BT52, Luc59]): there is an unknown weight vector $\mathcal{D} \in \mathbb{R}^n_+$ ($\mathcal{D}(u)$ indicates the quality of item $u$) and, for two items $u, v$, the algorithm observes the sample $(u, \{u, v\})$, i.e., that $u$ beats $v$ with probability $\frac{\mathcal{D}(u)}{\mathcal{D}(u)+\mathcal{D}(v)}$. Motivated by the importance of perfect simulation, we ask the following question:

**Question 1.** *Is there an efficient algorithm that draws i.i.d. samples $(x, S)$ from $\mathrm{Samp}(\mathcal{Q}; \mathcal{D})$ and generates a single perfect sample $y \sim \mathcal{D}$?*

To put our contribution into context, let us ask another question whose answer is clear from previous work. What is the sample complexity of *learning* $\mathcal{D}$ from a Local Sampling Scheme $\mathrm{Samp}(\mathcal{Q}; \mathcal{D})$? Here, the goal is to estimate $\mathcal{D}$ in some $L_p$ norm given such comparisons. Some works focus on learning the re-parameterization $\mathbf{z} \in \mathbb{R}^n$ with $z_x = \log(\mathcal{D}(x))$. Depending on the context, either the normalization condition $\langle \mathbf{1}, \mathcal{D} \rangle = 1$ or $\langle \mathbf{1}, \mathbf{z} \rangle = 0$ is used. [SBB+16] deal with the problem of learning the re-parameterization $\mathbf{z}$ in the $L_2$ norm when $|S| = 2$ (pairwise comparisons) or $|S| = k$ ($k$-wise comparisons with $k = O(1)$). The sample complexity for learning $\mathbf{z}$ in $L_2$ norm from pairwise comparisons is $n/(\lambda(\mathbf{Q})\epsilon^2)$, where $\lambda(\mathbf{Q})$ is the second smallest eigenvalue of a Laplacian matrix $\mathbf{Q}$, induced by the pair distribution $\mathcal{Q}$ (the learning result is tight in a minimax sense for some appropriate semi-norm [SBB+16]). In particular, the matrix $\mathbf{Q}$ is defined as $\mathbf{Q}_{xy} = -\mathcal{Q}(x, y)$ and $\mathbf{Q}_{xx} = \sum_{y \neq x} \mathcal{Q}(x, y)$. Hence, in the pairwise comparisons setting, the sample complexity of

learning incurs an overhead associated with $\lambda(\boldsymbol{Q})$ (Table 1). The algorithm of [SBB+16] is a pivotal component for our main result, as we will see later. [APA18] provide a similar result for learning $\mathcal{D}$ in TV distance using a random-walk approach. We remark that learning from pairwise comparisons requires some mild conditions concerning the sampling process (LSS in our case). We review them shortly below.

Table 1: Learning and Exact Sampling from PCs of size 2. $\widetilde{O}(\cdot)$ subsumes logarithmic factors.

| Sample Access | Learning ($\epsilon > 0$) | Exact Sampling |
|---|---|---|
| Definition 1 with Pairwise Comparisons | $(\boldsymbol{z}, L_2)$ $\quad O(n/\epsilon^2) \cdot \frac{1}{\lambda(\boldsymbol{Q})}$ [SBB+16] | $\widetilde{O}(n^2) \cdot \frac{1}{\lambda(\boldsymbol{Q})}$ (Theorem 3) |

**Conditions for Local Sampling Schemes.** We provide a standard pair of conditions for Local Sampling Schemes (these or similar conditions are also needed in the learning problem). We are interested in LSSs that satisfy both of these conditions. However, some of our results still hold when only the first condition is true; this will be more clear later. Assumption 1 is a necessary information-theoretic condition for the pair distribution $\mathcal{Q}$.

**Assumption 1** (Identifiability). *The support of the pair distribution $\mathcal{Q}$ of the Local Sampling Scheme contains a spanning tree, i.e., the induced graph $G_{\mathcal{Q}}$ is connected. We define $\mathcal{E} := E(G_{\mathcal{Q}})$.*

Intuitively, the pair distribution $\mathcal{Q}$ needs to be supported on a spanning tree. Equivalently, the associated Laplacian matrix $\boldsymbol{Q}$ should have positive Fiedler eigenvalue $\lambda(\boldsymbol{Q}) > 0$. Assumption 2 is a condition about the support of $\mathcal{Q}$ *and* the target distribution $\mathcal{D}$; it allows efficient learning of $\mathcal{D}$ from pairwise comparisons and is related to the difficulty of estimating small $\mathcal{D}$-probabilities.

**Assumption 2** (Efficiently Learnable). *There exists a constant $\phi > 1$ such that target distribution $\mathcal{D}$ satisfies $\frac{1}{\phi} \leq \max_{(x,y) \in \mathcal{E}} \frac{\mathcal{D}(x)}{\mathcal{D}(y)} \leq \phi$, where $\mathcal{E}$ is the support of the distribution $\mathcal{Q}$.*

From a dual perspective, Assumption 2 says that $\mathcal{Q}$ is supported only on edges where the two corresponding $\mathcal{D}$-ratios are well controlled and implies that any conditional distribution $\mathcal{D}_{\{x,y\}}$ has bounded variance, i.e., $(1/\phi)/(1 + \phi)^2 \leq \frac{\mathcal{D}(x)\mathcal{D}(y)}{(\mathcal{D}(x)+\mathcal{D}(y))^2} \leq \phi/(1 + 1/\phi)^2$. A quite similar property has been proposed in the problem of learning from pairwise comparisons [SBB+16].

**Contributions and Techniques.** We provide an algorithmic answer to Question 1 regarding perfect sampling (under the above mild conditions) by showing the following:

*There exists an efficient algorithm that draws $\widetilde{O}(n^2/\lambda(\boldsymbol{Q}))$ samples from $\mathrm{Samp}(\mathcal{Q}; \mathcal{D})$ and outputs a perfect sample $x \sim \mathcal{D}$.*

The tool that makes this possible (and algorithmic) is a novel variant of the elegant Coupling from the Past (CFTP) algorithm of [PW96]. CFTP is a technique developed by Propp and Wilson ([PW96, PW98]), that provides an exact random sample from a Markov chain, that has law the (unique) stationary distribution (we refer to Appendix A.2.3 for an introduction to CFTP). To make our contribution clear, we provide two theorems; Informal Theorem 1 that directly and "naively" applies CFTP (indicating the challenges behind efficient sample correction) and a more sophisticated Informal Theorem 2 that *combines CFTP and learning from pairwise comparisons* in order to get the result of Table 1. We show how to efficiently generate "global" samples from the true distribution $\mathcal{D}$ from few samples from a Local Sampling Scheme $\mathrm{Samp}(\mathcal{Q}; \mathcal{D})$. To this end, our novel algorithm combines the CFTP method [PW96, PW98], a remarkable technique that generates exact samples from the stationary distribution of a given Markov chain, with (i) an efficient algorithm that estimates the parameters of Bradley-Terry-Luce ranking models from pairwise (and, in general, $k$-wise) comparisons [SBB+16] and (ii) procedures based on Bernoulli factories [DHKN17, MP05, NP05] and rejection sampling.

At a technical level, we observe (Section 2) that a Local Sampling Scheme $\mathrm{Samp}(\mathcal{Q}; \mathcal{D})$ naturally induces a canonical ergodic Markov chain on $[n]$ with transition matrix $\boldsymbol{M}$. The probability of a transition from a state $x$ to a state $y$ is equal to the probability $\mathcal{Q}(x, y)$, that the pair $\{x, y\}$ is drawn from $\mathcal{Q}$, times the probability $\mathcal{D}(y)/(\mathcal{D}(x) + \mathcal{D}(y))$, that $y$ is drawn from the conditional distribution $\mathcal{D}_{\{x,y\}}$. Then, the unique stationary distribution of the resulting Markov chain coincides with the true

distribution $\mathcal{D}$. Applying (a non-adaptive version of) the Coupling from the Past algorithm, we obtain the following result under Assumption 1 and Assumption 2 for a target distribution $\mathcal{D}$ over $[n]$. For the matrix $M$, we let $\Gamma(M)$ be its absolute spectral gap (see Appendix A.1 for extensive notation).

**Informal Theorem 1** (Direct Exact Sampling from LSS). *With an expected number of $\widetilde{O}\left(\frac{n^2}{\Gamma(M)}\right)$ samples from a Local Sampling Scheme $\mathrm{Samp}(\mathcal{Q};\mathcal{D})$, there exists an algorithm (Algorithm 1) that generates a sample distributed as in $\mathcal{D}$.*

The sample complexity of Informal Theorem 1 is, for instance, attained by the instance of Figure 1, which we will discuss shortly.[1] We remark that Informal Theorem 1 holds more generally for Local Sampling Schemes that only satisfy Assumption 1 and not Assumption 2; for a formal version of the above result, we refer to Theorem 2 and for a discussion on the resulting sample complexity, we refer also to Remark 1. Let us now see why Informal Theorem 1 is quite unsatisfying.

In Informal Theorem 1, the matrix $M$ contains the transitions induced by $\mathrm{Samp}(\mathcal{Q};\mathcal{D})$ and the transition $u \to v$ is performed with probability $\mathcal{Q}(u,v)\mathcal{D}(v)/(\mathcal{D}(u)+\mathcal{D}(v))$ and $\Gamma(M)$ is the absolute spectral gap of the Markov chain's transition matrix $M$ on which CFTP is performed. Note that both $M$ and $\Gamma(M)$ depend on $\mathcal{Q}$ and the target distribution $\mathcal{D}$. In many natural cases, e.g., if $\mathcal{D}$ is multimodal or includes many low probability points, the CFTP-based algorithm of Informal Theorem 1 can be quite inefficient! For instance, Figure 1 is a bad instance for the standard CFTP algorithm. It corresponds to an instance where the support of $\mathcal{Q}$ is the path graph over $[n]$ and the target distribution $\mathcal{D}$ is bimodal, satisfying Assumption 1 and Assumption 2 with $\phi = 2$. In this case, $M$ captures the transitions of a negatively biased nearest-neighbor random walk on the path, i.e., the probability of moving to the boundary is larger than moving to the interior. The coalescence probability of the two extreme points is of order $\alpha^n$ for some $0 < \alpha < 1$. This probability is connected to the maximum hitting time and, consequently, we get that the mixing time is $\Omega(\alpha^{-n})$ [LP17] and so the coalescence time is exponentially large. Informal Theorem 1 depends on $\Gamma(M)$ and hence the *exponential dependence on $n$* is reflected in the conductance of the chain, which is exponentially small. So, can we eliminate the bias due to Definition 1 efficiently?

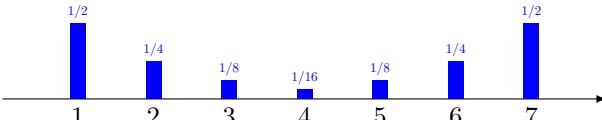

Figure 1: A bad instance for Informal Theorem 1.

As a technical contribution, we provide a novel exact sampling algorithm whose convergence does not depend on $\mathcal{D}$. We show how to improve the efficiency of CFTP by removing the dependence of its sample (and computational) complexity on the structure of the true distribution $\mathcal{D}$. In particular, in Section 3, we show the following under Assumption 1 and Assumption 2:

**Informal Theorem 2** (Exact Sampling from LSS using Learning). *There exists an algorithm (Algorithm 2 & Algorithm 3) that draws an expected number of $\widetilde{O}\left(\frac{n^2}{\lambda(\mathbf{Q})}\right)$ samples from a Local Sampling Scheme $\mathrm{Samp}(\mathcal{Q};\mathcal{D})$, and generates a sample distributed as in $\mathcal{D}$.*

In the above, $\lambda(\mathbf{Q})$ is the second smallest eigenvalue (a.k.a., the Fiedler eigenvalue) of $\mathbf{Q}$, the weighted Laplacian matrix of the graph $G_{\mathcal{Q}}$ induced by $\mathcal{Q}$; this is the same matrix that appears in the learning result of Table 1. *Crucially, the sample complexity in the above result is polynomial in $n$*, provided that the pair distribution $\mathcal{Q}$ is reasonable. For the path graph of Figure 1 with $\mathcal{Q}$ the uniform distribution, Informal Theorem 1 yields an exponential sample complexity but the spectrum of the matrix of Informal Theorem 2 is $\mathrm{spec}(\mathbf{Q}) = \{\frac{1}{n}\Theta(1 - \cos(\pi i/n))\}_{i=0}^{n-1}$. Hence, since $1 - \cos(x) = \sin^2(x/2) \approx x^2$, we get that the sample complexity is $\mathrm{poly}(n)$. Moreover, the runtime of the algorithm is sample polynomial.

---

[1]For the instance of Figure 1 one can reduce the sample complexity of our results by a factor of $n$, since the state space has a natural ordering and one could apply monotone CFTP (it suffices to coalesce two random processes and not $n$).

**Technical Overview.** A first key idea towards establishing Informal Theorem 2 is to exploit *learning* in order to accelerate the convergence of CFTP. We first apply (a modification of) the gradient-based algorithm of [SBB$^+$16] to the empirical log-likelihood objective, which estimates the parameters of BTL ranking models from PCs, with samples from $\mathrm{Samp}(\mathcal{Q};\mathcal{D})$. As a result of this learning algorithm (whose efficiency requires Assumption 2, as in [SBB$^+$16]), we get a probability distribution $\widetilde{\mathcal{D}}$, which approximates $\mathcal{D}$ with small relative error, in the sense that $\mathcal{D}(x) \in (1 \pm \epsilon) \cdot \widetilde{\mathcal{D}}(x)$, for any $x$ in $\mathcal{D}$'s support. As we saw in Table 1, the sample complexity will be roughly of order $O(n/(\lambda(\boldsymbol{Q})\epsilon^2)))$. To be precise, this learning phase gives us a sample complexity term of order $O(n^2/\lambda(\boldsymbol{Q}))$ and corresponds to a single execution of the above learning algorithm for the target $\mathcal{D}$ in relative error with accuracy $\epsilon = \Theta(1/\sqrt{n})$ (the accuracy's choice will be more clear later).

The new idea is that we can use the output of the learning algorithm $\widetilde{\mathcal{D}}$ and a Bernoulli factory mechanism to transform the Markov chain induced by $\mathrm{Samp}(\mathcal{Q};\mathcal{D})$ into an ergodic chain with almost uniform stationary probability distribution, whose transition probabilities (and mixing time) essentially do not depend on $\mathcal{D}$! To do so, for each pair of states $x, y$, with $(x,y) \in \mathcal{E}$ and $\widetilde{\mathcal{D}}(x) > \widetilde{\mathcal{D}}(y)$, we can use Bernoulli downscaling (which constitutes an essential building block for general Bernoulli factories, see e.g., [DHKN17, MP05, NP05]) to transform the (unknown) probability of a transition from $y$ to $x$ to $\mathcal{D}(x)/(\mathcal{D}(x) + \mathcal{D}(y)) \cdot \widetilde{\mathcal{D}}(y)/\widetilde{\mathcal{D}}(x)$ (conditional that the edge $(x,y)$ is drawn). Since $\widetilde{\mathcal{D}}(x) \approx \mathcal{D}(x)$, for all states $x$, this makes the probability of a transition from $x$ to $y$ almost equal to the probability of a transition from $y$ to $x$, which, in turn, implies that *the stationary distribution of the modified Markov chain is almost uniform.* Thus, we speed up the CFTP algorithm by removing the modes due to the landscape of the target $\mathcal{D}$ and can get an exact sample from the stationary distribution of the modified Markov chain with $O\left(n\log(n)/\lambda(\boldsymbol{Q})\right)$ samples. Namely, the sample complexity of this parameterized variant of the CFTP algorithm does not depend on the structure of $\mathcal{D}$ anymore (of course, it does depend on the size $n$ of $\mathcal{D}$'s support). The above constitute our second key idea and the intuition behind this re-weighting comes from the method of simulated tempering [MP92, GLR18b, GLR18a].

Our third idea deals with the output of the algorithm. Since we want an exact/clean sample from $\mathcal{D}$, we cannot simply print the output of the above process. As a last step, we use rejection sampling, guided by the distribution $\widetilde{\mathcal{D}}$, and apply the "reverse" modification, bringing the distribution of the resulting sample back to $\mathcal{D}$ (see Algorithm 2). With high probability, this last rejection sampling will succeed after $\Theta(n)$ executions of the parameterized CFTP algorithm. However, note that we do not have to execute the learning algorithm again. Hence, the total sample complexity is equal to the cost of the learning phase (that is performed only once and its output is given as input to the parameterized CFTP algorithm; see Algorithm 2) and the cost of the modified CFTP multiplied by $\Theta(n)$ (due to rejection sampling), which yields a total of $\widetilde{O}(n^2/\lambda(\boldsymbol{Q}))$ samples. The above modification of the Markov chain induced by $\mathrm{Samp}(\mathcal{Q};\mathcal{D})$, which improves possibly significantly its mixing time, is simple and general and the parameterized variation of the standard CFTP algorithm (Algorithm 2) might be of independent interest. In Appendix G, we discuss how to extend the analysis of the main Algorithm 2 to sets of size larger than 2.

**Related Work.** Our results are related to and draw ideas from several areas of theoretical computer science, which we review below.

**Exact Sampling.** Exact sampling comprises a well-studied field [Ken05, Hub16] with numerous applications in theoretical computer science (e.g., to approximate counting [Hub98]). One of our main tools, the Coupling From the Past algorithm, is the breakthrough result of [PW96, PW98], which established the applicability of perfect simulation, and made efficient simulation of challenging problems (e.g., Ising model) feasible. After this fundamental result, the literature of perfect simulation flourished and various exact sampling procedures have been developed, see e.g., [PW96, GM99, Wil00, HS00, FH00, Hub04]), and the references therein.

**Learning and Ranking from Pairwise Comparisons.** There is a vast amount of literature on ranking and parameter estimation from pairwise comparisons (see e.g., [FV86, Mar96, Cat12] and the references therein). A significant challenge in that field is to come up with models that accurately capture uncertainty in pairwise comparisons. The standard ones are the Bradley-Terry-Luce (BTL) model [BT52, Luc59] and the Thurstone model [Thu27]. Our work is closely related to previous work on the BTL model [RA14, SBGW16, DKNS01, NOS12, Hun04, JLYY11, APA18, LSR21] and our sample complexity rates depend on the Fiedler eigenvalue of a pairwise comparisons' matrix, as in [HOX14, SBB$^+$16, VY16, NOS17, VYZ20].

**Truncated Statistics.** Our work falls in the research agenda of efficient learning and exact sampling from truncated samples [Bre96, Coh16]. Recent work has obtained efficient algorithms for efficient learning from truncated [DGTZ18, KTZ19, DGTZ19, IZD20, DRZ20, BDNP21, NP19, FKT20, DKTZ21] and censored [MMS21, FKKT21, Ple21] samples. Closer to our work, [FKT20] also deal with the question of exact sampling from truncated samples.

**Correction Sampling.** Sample correction is a field closely related to our work. This area of research deals with the case where input data are drawn from a noisy or imperfect source and the goal is to design *sampling correctors* [CGR18] that operate as filters between the noisy data and the end user. These algorithms exploit the structure of the distribution and make "on-the-fly" corrections to the drawn samples. Moreover, the problem of local correction of data has received much attention in the contexts of self-correcting programs, locally correctable codes, and local filters for graphs and functions (see e.g., [BLR93, SS10, JR13, ACCL04, BGJ+10, GLT20]).

**Conditional Sampling.** Conditional sampling deals with the *adaptive* learning analog of LSS, where the goal is again to learn an underlying discrete distribution from conditional/truncated samples, but the learner can change the truncation set on demand and is extensively studied [Can15, FJO+15, ACK15b, BC18, ACK15a, GTZ17, KT19, CCK+19].

## 2 An Exact Sampling Algorithm using CFTP

Consider a discrete target distribution $\mathcal{D}$. For pair distributions $\mathcal{Q}$, the Local Sampling Scheme $\mathrm{Samp}(\mathcal{Q};\mathcal{D})$ can be regarded as a graph $G = (V, E)$ with $V = \mathrm{supp}(\mathcal{D})$ and $\mathcal{E} = \mathrm{supp}(\mathcal{Q})$. We let $V = [n]$. Let $\boldsymbol{M}$ denote the canonical transition matrix of the Markov chain, associated with the oracle $\mathrm{Samp}(\mathcal{Q};\mathcal{D})$. The entries of $\boldsymbol{M} = [\boldsymbol{M}_{xy}]_{x,y\in[n]}$ are defined as:

$$\boldsymbol{M}_{xy} = \mathcal{Q}(x,y)\frac{\mathcal{D}(y)}{\mathcal{D}(x) + \mathcal{D}(y)} \text{ for any } x \neq y \text{ and } \boldsymbol{M}_{xx} = 1 - \sum_{y\neq x}\boldsymbol{M}_{xy} \text{ for } x \in [n]. \quad (1)$$

Observe that the transition from $x$ to $y$ is performed when both the edge $\{x, y\}$ is drawn from $\mathcal{Q}$, with probability proportional to $\mathcal{Q}(x, y)$ and $y$ is drawn from the conditional distribution $\mathcal{D}_{\{x,y\}}$. The Markov chain, whose transitions are described by $\boldsymbol{M}$, has some notable properties: it has $\mathcal{D}$ as stationary distribution and is ergodic. We can invoke the fundamental CFTP algorithm over the Markov chain of [PW96, PW98] to obtain a perfect sample from $\mathcal{D}$.

---

**Algorithm 1** Exact Sampling from Local Sampling Schemes of Informal Theorem 1 & Theorem 2

1: **procedure** EXACTSAMPLER()        ▷ *Sample access to the LSS oracle* $\mathrm{Samp}(\mathcal{Q};\mathcal{D})$.
2:      $t \leftarrow 0$
3:      $F_0(x) \leftarrow x$, for any $x \in [n]$
4:      **while** $F_t$ has not coalesced **do**        ▷ *While no coalescence has occured.*
5:          $t \leftarrow t - 1$
6:          Draw sample $(i, j, w)$ with $(i, j) \in \mathcal{E}$ and $w \in \{i, j\}$
7:          **for** $x = 1 \ldots n$ **do**        ▷ *In order to update state $x$.*
8:             **if** $x \notin \{i, j\}$ or $x = w$ **then** $F_t(x) \leftarrow F_{t+1}(x)$     ▷ *Append $F_t$ in the past.*
9:             **else** $F_t(x) \leftarrow F_{t+1}(w)$    ▷ *New transition with probability* $\mathcal{Q}(x,w)\frac{\mathcal{D}(w)}{\mathcal{D}(x)+\mathcal{D}(w)}$.
10:          **end**
11:      **end**
12:      Output $F_t(1)$        ▷ *Output the perfect sample.*
13: **end procedure**

---

We extensively discuss CFTP in Appendix A.2.3 for the interested reader. CFTP yields the following (unsatisfying) result, whose proof can be found at the Appendix B.

**Theorem 2** (Direct Exact Sampling from LSS). *Let $\mathcal{D}_{\min}$ be the minimum value of the target distribution $\mathcal{D}$. Under Assumption 1 and for any $\delta > 0$, Algorithm 1 draws, with probability at least $1 - \delta$, $N = O\left(\frac{n\log(1/\mathcal{D}_{\min})}{\Gamma(\boldsymbol{M})}\log(\frac{1}{\delta})\right)$ samples from a Local Sampling Scheme $\mathrm{Samp}(\mathcal{Q};\mathcal{D})$ over $[n]$, runs in time polynomial in $N$ and outputs a sample distributed as in $\mathcal{D}$.*

Note that in the above result (Theorem 2), the confidence parameter $\delta$ corresponds to the number of samples required and not the quality of the output sample (the sample is perfect with probability

1). We comment that the proof of the above Theorem is nearly straightforward and follows from the analysis of the CFTP algorithm; the purpose of Theorem 2 is to indicate that a direct application of CFTP would potentially lead to an exponential number of samples due to the shape of the target distribution. Our main technical contribution begins in the next section: We give an exact sampler that surpasses the dependence on the target's structure using learning and rejection sampling.

## 3 Improving the Exact Sampling Algorithm using Learning

The main drawback of Theorem's 2 algorithm is that its sample complexity depends on both $\boldsymbol{Q}$ and $\boldsymbol{D}$ and consequently in many natural scenaria the sample complexity may be exponentially large in $n$. Next, we present Algorithm 2, which will allow us to remove the dependence on $\boldsymbol{D}$ for the Markov chain and reduce the sample complexity to polynomial in the domain size. The crucial differences compared to Algorithm 1 are the orange lines. Algorithm 2 is a parameterized extension of CFTP and its input is a vector $\boldsymbol{p} \in [0,1]^n$. We remark that when $\boldsymbol{p}$ is the all-ones vector $\mathbf{1}$, we obtain the previous Algorithm 1.

---

**Algorithm 2** Parameterized Extension of Algorithm 1

---
1: **Input:** $\boldsymbol{p} \in [0,1]^n$      ▷ *For instance, this may be the output of Algorithm 5.*
2: **procedure** EXACTSAMPLER($\boldsymbol{p}$)     ▷ *Sample access to the LSS oracle* $\mathrm{Samp}(\mathcal{Q}; \mathcal{D})$.
3:   $t \leftarrow 0$, $F_0(x) \leftarrow x$, for any $x \in [n]$
4:   **while** $F_t$ has not coalesced **do**     ▷ *While no coalescence has occured.*
5:    $t \leftarrow t - 1$
6:    Draw sample $(i, j, w)$ with $(i, j) \in \mathcal{E}$ and $w \in \{i, j\}$
7:    **for** $x = 1 \ldots n$ **do**       ▷ *In order to update state* $x$.
8:     **if** $x \notin \{i, j\}$ or $x = w$ **then** $F_t(x) \leftarrow F_{t+1}(x)$
9:     **else** $F_t(x) \leftarrow \begin{cases} F_{t+1}(w), & \text{with probability } \min\{p(x)/p(w), 1\} \\ F_{t+1}(x), & \text{otherwise} \end{cases}$
10:    **end**
11:   **end**
12:   Draw $C \sim \mathrm{Be}(p(F_t(1)))$      ▷ $\mathrm{Be}(p)$ *is a $p$-biased Bernoulli coin.*
13:   **if** $C = 1$ **then** Output $F_t(1)$ **else** Output $\perp$   ▷ *Output the perfect sample or Fail.*
14: **end procedure**

---

We perform Algorithm 2 using as input parameter $\boldsymbol{p}$ an estimate $\widetilde{\mathcal{D}}$ of the target $\mathcal{D}$ with (relative) error of order $\epsilon = 1/\sqrt{n}$ (this estimate is obtained via Theorem 4 & Algorithm 5 using $O(n^2/\lambda(\boldsymbol{Q}))$ samples from $\mathrm{Samp}(\mathcal{Q}; \mathcal{D})$, as we will see later). Under the perspective of LSSs as random walks on an irreducible Markov chain, Algorithm 2 proceeds by executing CFTP. Recall that in each draw from $\mathrm{Samp}(\mathcal{Q}; \mathcal{D})$, a transition can be realized by the matrix $\boldsymbol{M}$. The fact that this transition depends on the target $\mathcal{D}$ is pathological, as we observed previously. The algorithm, instead of performing this transition (as the non-adaptive CFTP algorithm of Theorem 2 does), performs a Bernoulli factory mechanism (downscaling), using the estimates $\widetilde{\mathcal{D}}$ in order to make each transition almost uniform (see Line 9 of Algorithm 2). This change introduces some (known) bias to the random walk and changes the stationary distribution from $\mathcal{D}$ to an almost uniform one. By performing the CFTP method, the algorithm iteratively reconstructs the past of the infinite simulation, until all the simulations have coalesced at time $t = 0$. When coalescence occurs, the algorithm has an exact sample from the biased (almost uniform) stationary distribution. Since the introduced bias is known (it corresponds to the inverse of the estimates $\widetilde{\mathcal{D}}$), the algorithm accepts the sample using rejection sampling, guided by $\widetilde{\mathcal{D}}$, so that the final sample is distributed according to the target distribution $\mathcal{D}$ (see Line 13 of Algorithm 2). Specifically, we show (for the proof, see Appendix C) that:

**Theorem 3** (Exact Sampling from LSS using Learning). *Under Assumption 1 and Assumption 2, for any $\delta > 0$, there exists an algorithm (Algorithm 3) that draws, with probability at least $1 - \delta$, $N = O\left(n^2 \log^2(n)/\lambda(\boldsymbol{Q}) \cdot \log(1/\delta)\right)$ samples from a Local Sampling Scheme $\mathrm{Samp}(\mathcal{Q}; \boldsymbol{Q})$, runs in time polynomial in $N$, and outputs a sample distributed as in $\mathcal{D}$.*

We proceed with the analysis of Algorithm 3, which can be decomposed into two parts: the Learning Step (Line 2) and several iterations of Algorithm 2 (Line 5).

---
**Algorithm 3** Exact Sampling from Local Sampling Schemes of Informal Theorem 2 & Theorem 3
---
1: **procedure** EXACTSAMPLERWITHLEARNING($\delta$)                     ▷ *The algorithm of Theorem 3.*
2:     $\widetilde{\mathcal{D}} \leftarrow$ LEARN($\epsilon := 1/\sqrt{n}, \delta$)     ▷ *Learn $\mathcal{D}$ in relative error with Algorithm 5 (Theorem 4).*
3:     x $\leftarrow \perp$
4:     **while** x $= \perp$ **repeat**
5:         x $\leftarrow$ EXACTSAMPLER($\widetilde{\mathcal{D}}$)                          ▷ *Call Algorithm 2.*
6:     Output x
7: **end procedure**
---

**Learning Step.** For two distributions $\mathcal{D}, \widetilde{\mathcal{D}}$ with ground set $[n]$, we introduce the sequence/list (of length $n$) $1 - \mathcal{D}/\widetilde{\mathcal{D}} := (1 - \mathcal{D}(x)/\widetilde{\mathcal{D}}(x))_{x \in [n]}$. Observe that the pair of sequences $(1 - \mathcal{D}/\widetilde{\mathcal{D}}, 1 - \widetilde{\mathcal{D}}/\mathcal{D})$ captures the relative error between the two distributions. The sample complexity of the task of learning $\mathcal{D}$ in $\epsilon$-relative error is summarized by the following:

**Theorem 4** (Learning Phase). *For any $\epsilon, \delta > 0$, there exists an algorithm (Algorithm 5) that draws $N = O\left(\frac{n}{\lambda(\boldsymbol{Q})\epsilon^2} \log(\frac{1}{\delta})\right)$ samples from $\mathrm{Samp}(\mathcal{Q}; \mathcal{D})$ satisfying Assumptions 1 and 2, runs in time polynomial in $N$, and, with probability at least $1 - \delta$, computes an estimate $\widetilde{\mathcal{D}}$ of the target distribution $\mathcal{D}$, that satisfies the relative error guarantee $\max\left\{\|1 - \mathcal{D}/\widetilde{\mathcal{D}}\|_\infty, \|1 - \widetilde{\mathcal{D}}/\mathcal{D}\|_\infty\right\} \leq \epsilon$.*

We execute *once* the learning algorithm (Algorithm 5) that estimates $\mathcal{D}$ in relative error and we choose accuracy $\epsilon = 1/\sqrt{n}$. We would like to shortly comment on the choice of the accuracy value: An accuracy of constant order would still potentially yield a biased random walk. After the re-weighting step, our goal is to show that the target distribution becomes almost uniform and that the absolute spectral gap of the reformed matrix is of order $\lambda(\boldsymbol{Q})$ (see the proof of Lemma 15). In order to ensure this, any accuracy that vanishes with $n$ is sufficient. We chose to set $\epsilon = 1/\sqrt{n}$ so that the complexity of learning $(\frac{n}{\lambda(\boldsymbol{Q})\epsilon^2})$ matches that of the exact sampling procedure $(\frac{n^2}{\lambda(\boldsymbol{Q})})$. This learning step costs $n^2/\lambda(\boldsymbol{Q})$ samples and the estimate $\widetilde{\mathcal{D}}$ is used as input to Algorithm 2. For the proof of Theorem 4, we refer to the Appendix E. We can now move to Line 5.

**Executing Algorithm 2.** We next focus on a single execution of the parameterized CFTP algorithm. Our goal is to transform the Markov chain of the Local Sampling Scheme defined in Equation (1) to a modified Markov chain with an almost uniform stationary distribution. We are going to provide some intuition. Applying Theorem 4, we can consider that, for any $x \in [n]$, there exists a coefficient $\widetilde{\mathcal{D}}(x) \approx \mathcal{D}(x)$. The idea is to use $\widetilde{\mathcal{D}}(x)$ and make the stationary distribution of the modified chain close to uniform. Intuitively, this transformation should speedup the convergence of the CFTP algorithm. The downscaling method of Bernoulli factories gives us a tool to do this modification.

We can implement the modified Markov chain via downscaling as follows: Consider an edge $\{x, y\}$ with transition probability pair $(p_{xy}, p_{yx})$. Without loss of generality, we assume that $\widetilde{\mathcal{D}}(y) > \widetilde{\mathcal{D}}(x)$ (which intuitively means that we should expect that $p_{xy} > p_{yx}$). Then, the downscaler leaves $p_{yx}$ unchanged and reduces the mass of $p_{xy}$ to make the two transitions almost balanced. Consider the LSS transition matrix $\boldsymbol{M}$ with $\boldsymbol{M}_{xy} = \mathcal{Q}(x, y)\frac{\mathcal{D}(y)}{\mathcal{D}(x)+\mathcal{D}(y)}$ and $\boldsymbol{M}_{xx} = 1 - \sum_{y \neq x} \boldsymbol{M}_{xy}$. Also, let $\widetilde{\mathcal{D}}$ be an estimate for the distribution $\mathcal{D}$ (Theorem 4). For the pair $(x, y)$, we modify the transition probability $p_{xy} := \boldsymbol{M}_{xy}$, only if $p_{xy} > p_{yx}$, to be equal to the following: $\widetilde{p}_{xy} = \mathcal{Q}(x, y)\frac{\mathcal{D}(y)}{\mathcal{D}(x)+\mathcal{D}(y)}\frac{\widetilde{\mathcal{D}}(x)}{\widetilde{\mathcal{D}}(y)} \approx \mathcal{Q}(x, y)\frac{\mathcal{D}(x)}{\mathcal{D}(x)+\mathcal{D}(y)} = p_{yx}$, where we use that $\widetilde{\mathcal{D}}(x) \approx \mathcal{D}(x)$ and $\widetilde{\mathcal{D}}(y) \approx \mathcal{D}(y)$. The transition probability from $x$ to $y$ corresponds to a Bernoulli variable $\mathrm{Be}(p_{xy})$, which is downscaled by $\widetilde{\mathcal{D}}(x)/\widetilde{\mathcal{D}}(y) < 1$. The modified transition probability $\widetilde{p}_{xy}$ can be implemented by drawing a $\Lambda \sim \mathrm{Be}(\widetilde{\mathcal{D}}(x)/\widetilde{\mathcal{D}}(y))$ and then drawing a $P \sim \mathrm{Be}(p_{xy})$ (from $\mathrm{Samp}(\mathcal{Q}; \mathcal{D})$) and, finally, realizing the transition from $x$ to $y$ only if $\Lambda P = 1$. This implementation is valid since the two sources of randomness are independent. So, we perform a downscaled random walk based on the transition matrix $\widetilde{\boldsymbol{M}}$ with $\widetilde{\boldsymbol{M}}_{xy} = \widetilde{p}_{xy}$ (the "small" probability $p_{yx}$ remained intact). We call $\widetilde{\boldsymbol{M}}$ the $\widetilde{\mathcal{D}}$-scaling of $\boldsymbol{M}$. Lemma 5 summarizes the key properties of $\widetilde{\boldsymbol{M}}$. The definition of the mixing time, used below, can be found at the notation Appendix A.1 and the full proof can be found at the Appendix D.

**Lemma 5** (Properties of Rescaled Random Walk). *Let $\mathcal{D}$ be a distribution on $[n]$ and consider an $\epsilon$-relative approximation $\widetilde{\mathcal{D}}$ of $\mathcal{D}$, as in Theorem 4 with $\epsilon = 1/\sqrt{n}$. Consider the transition matrix $M$ of the Local Sampling Scheme $\mathrm{Samp}(\mathcal{Q}; \mathcal{D})$ (see Equation (1)), and let $\widetilde{M}$ be the $\widetilde{\mathcal{D}}$-scaling of $M$. Then, the transition matrix $\widetilde{M}$ (i) has stationary distribution $\widetilde{\pi}_0$ with $\widetilde{\pi}_0(x) = \Theta(1/n)$ for all $x \in [n]$ and (ii) the mixing time of the transition matrix $\widetilde{M}$ is $T_{\mathrm{mix}}(\widetilde{M}) = O\left(\frac{\log(n)}{\lambda(\mathbf{Q})}\right)$.*

*Proof Sketch.* Set $J_{xy} = \mathcal{D}(x) + \mathcal{D}(y)$. For a pair $(x, y) \in \mathcal{E}$, before the downscaling, the pairwise $(x, y)$-comparison corresponds to the random coin $(\mathcal{D}(y)/J_{xy}, \mathcal{D}(x)/J_{xy})$ and let $\mathcal{D}(y) > \mathcal{D}(x)$. After the downscaling, this coin becomes (locally) almost fair, i.e., $(\mathcal{D}(x)/J_{xy} + \xi_{xy}, \mathcal{D}(x)/J_{xy})$. Hence, the modified transition matrix $\widetilde{M}$ can be written as $\widetilde{M} = \widetilde{D} \otimes \mathbf{Q} + \mathbf{Q} \circ [\xi_{xy}]$, where $\widetilde{D}$ is a symmetric matrix with $\widetilde{D}_{xy} = \min\{\mathcal{D}(x), \mathcal{D}(y)\}/(\mathcal{D}(x) + \mathcal{D}(y)) = \widetilde{D}_{yx}$, $\mathbf{Q}$ is the Laplacian matrix of the graph $G_{\mathcal{Q}}$ and $[\xi_{xy}]$ is a matrix with entries $\xi_{xy}$ where $\xi_{xy}$ is only non-zero in the modified transitions $x \to y$, i.e., if $p_{xy} > p_{yx}$ then $\widetilde{M}_{xy} = \mathcal{Q}(x, y) \cdot (\mathcal{D}(x)/J_{xy} + \xi_{xy})$; for this pair, we set $\xi_{yx} = 0$. Finally, $\mathbf{Q} \circ [\xi_{xy}]$ denotes the modified Hadamard product[2] between $\mathbf{Q}$ and $[\xi_{xy}]$. Due to the accuracy $\epsilon$ of the learning algorithm, we have that $|\xi_{xy}| = O(1/\sqrt{n})$.

We analyze (i) the stationary distribution and (ii) the mixing time. For **Part (i)**, the chain $\widetilde{M}$ remains irreducible and the detailed balance equations of $\widetilde{M}$ satisfy $\widetilde{p}_{xy}/\widetilde{p}_{yx} = \mathcal{D}(y) \cdot \widetilde{\mathcal{D}}(x)/(\widetilde{\mathcal{D}}(y) \cdot \mathcal{D}(x))$. Since $\widetilde{\mathcal{D}}$ is an $\epsilon$-relative approximation of $\mathcal{D}$ with $\epsilon = O(1/\sqrt{n})$, it holds that for any $x \in [n]$: $\mathcal{D}(x)/\widetilde{\mathcal{D}}(x) \in [1 - \epsilon, 1 + \epsilon]$. Hence, the unique stationary distribution $\widetilde{\pi}_0$ satisfies: $\widetilde{\pi}_0(x)/\widetilde{\pi}_0(y) \in [1 - \Theta(1/\sqrt{n}), 1 + \Theta(1/\sqrt{n})]$ for any $x, y \in [n]$. So, it must be the case that $\widetilde{\pi}(x) = \Theta(1/n)$.

For **Part (ii)**, in order to show that the mixing time is of order $O(\log(n)/\lambda(\mathbf{Q}))$, it suffices to show (by Lemma 9) that (i) the minimum value of the stationary distribution is $O(1/n)$ and (ii) the absolute spectral gap $\Gamma(\widetilde{M})$ of $\widetilde{M}$ satisfies $\Gamma(\widetilde{M}) = \Omega(\lambda(\mathbf{Q}))$, where $\lambda(\mathbf{Q})$ is the minimum non-zero eigenvalue of the Laplacian matrix $\mathbf{Q}$. The first property follows directly from the closeness of the stationary distribution of $\widetilde{M}$ to the uniform one. We now focus on the latter. Our goal is to control the absolute spectral gap $\Gamma(\widetilde{M})$. In fact, this matrix can be seen as a perturbed version of the matrix $\widetilde{D} \otimes \mathbf{Q}$ as we discussed in the beginning of the proof sketch. The perturbation noise matrix $\mathbf{E} = \mathbf{Q} \circ [\xi_{xy}]$ is a matrix whose entries contain (among others) the approximation errors $[\xi_{xy}]$. Hence, we can use Weyl's inequality [Tao12] in order to control the absolute spectral gap of $\widetilde{M}$. This inequality gives a crucial perturbation bound for the singular values for general matrices. Since the matrix $\widetilde{M}$ can be seen as a perturbation (as we discussed previously), we can (informally speaking) use Weyl's inequality to control the singular values After some algebraic manipulation, we derive that $\Gamma(\widetilde{M}) = \Omega(\lambda(\mathbf{Q}))$. $\square$

So, a *single iteration* of the parameterized CFTP algorithm with transition matrix $\widetilde{M}$ (see Line 5 of Algorithm 3) guarantees that with an expected number of $N = \widetilde{O}(nT_{\mathrm{mix}}(\widetilde{M})) = \widetilde{O}(n \log(n)/\lambda(\mathbf{Q}))$ samples, it reaches Line 12 with a sample $y = F_t(1)$ that is generated by the parameterized CFTP algorithm (the iterations in Line 4). We then have to study the *quality of the output sample $y$*. Crucially, by the utility of the standard CFTP algorithm, the law of $y$ is the normalized measure that is proportional to $\mathcal{D}/\widetilde{\mathcal{D}}$, i.e., $\widetilde{\pi}_0(y) = (\mathcal{D}(y)/\widetilde{\mathcal{D}}(y))/\sum_{x \in [n]} \mathcal{D}(x)/\widetilde{\mathcal{D}}(x)$ (this is the stationary distribution of the modified random walk). Hence, we cannot simply output $y$. For this reason, we perform rejection sampling. Specifically, as we will see, a single execution of Algorithm 2 either produces a single sample distributed as in $\mathcal{D}$ with some known probability $p$ or fails to output any sample with probability $1 - p$ (this is because we do not sample from $\mathcal{D}$ but from an almost uniform distribution). If the rejection sampling process is successful, then the while loop (Line 4) of Algorithm 3 will break and the clean sample will be given as output. One can show that $\Theta(n)$ iterations of the parameterized CFTP algorithm suffice to generate a perfect sample. Hence, the total sample complexity is of order $O(n^2 \log(n) \log(n/\delta)/\lambda(\mathbf{Q}))$ (at each step $\delta' = \Theta(\delta/n)$). We continue with a sketch behind the rejection sampling process of Line 13 of Algorithm 2.

---

[2]We have that $\left(\mathbf{Q} \circ \mathbf{A}\right)_{xy} = \mathcal{Q}_{xy} A_{xy}$, $\left(\mathbf{Q} \circ \mathbf{A}\right)_{xx} = -\sum_{y \neq x} \mathcal{Q}_{xy} A_{xy}$.

**Claim 1** (Sample Quality and Rejection Sampling). *Algorithm 2, with input the output of the Learning Phase, outputs a state $x \in [n]$ with probability proportional to $\mathcal{D}(x)$ or outputs $\perp$. Moreover, Algorithm 3 outputs a perfect sample from $\mathcal{D}$ with probability 1.*

To see this, set $A = \sum_{y \in [n]} \mathcal{D}(y)/\widetilde{D}(y)$. At the end of Line 11, Algorithm 2 outputs $x \propto \mathcal{D}(x)/\widetilde{D}(x)$, since the unique stationary distribution is $\widetilde{\pi}(x) = (\mathcal{D}(x)/\widetilde{D}(x))/A$ and the utility of the CFTP guarantees that the outputs state follows the stationary distribution. For the sample $x$, we perform a rejection sampling process, with acceptance probability $\widetilde{\mathcal{D}}(x)$. Algorithm 2 has $n+1$ potential outputs: It either prints $x \in [n]$ or $\perp$ indicating failure. The arbitrary sample $x \in [n]$ is observed with probability $\widetilde{\pi}(x) \cdot \widetilde{\mathcal{D}}(x) = \mathcal{D}(x)/A$. The remaining probability mass is assigned to $\perp$. We claim that the output of Line 6 of Algorithm 3 has law $\mathcal{D}$. Observe that the whole stochastic process of Algorithm 3 outputs $x \in [n]$ with probability $\sum_{i=0}^{\infty} \Pr[\text{Reject}]^i \Pr[x \text{ is Accepted}] = \mathcal{D}(x)/A \cdot \sum_{i=0}^{\infty} \Pr[\text{Reject}]^i$, where $\Pr[\text{Reject}] = \frac{1}{A} \sum_{y \in [n]} \frac{\mathcal{D}(y)}{\widetilde{\mathcal{D}}(y)}(1 - \widetilde{\mathcal{D}}(y)) = 1 - \frac{1}{A} \in (0,1)$. As a result, we get that the probability that $x$ is the output of the above stochastic process is $\mathcal{D}(x)$.

## 4 Conclusion

We close with an open question: Is the bound $\widetilde{O}(n^2/\lambda(\boldsymbol{Q}))$ tight? The main difficulty is that it is not clear how to obtain a lower bound against *any possible* exact sampling algorithm. We would like to mention that the dependence on $1/\lambda(\boldsymbol{Q})$ is expected and intuitively unavoidable. This term is connected to the mixing time. Let us become more specific. The (sample) complexity of our proposed method is $\widetilde{O}(n^2/\lambda(\boldsymbol{Q}))$. The term $1/\lambda(\boldsymbol{Q})$ appears due to the mixing time of the (transformed) Markov chain. The complexity of CFTP is $\widetilde{\Theta}(nT_{\text{mix}})$. Hence, any CFTP-based exact sampler (or more broadly random walk based algorithm) will have that kind of spectral dependence.

Similarly, a dependence on the support's size $n$ is necessary in general. Note that the structure of the target distribution $\mathcal{D}$ is crucial: if the target is uniform (or generally "flat"), then a bound of order $\widetilde{O}(n/\lambda(\boldsymbol{Q}))$ is obtained by the naive CFTP method; for distributions with modes, our algorithm achieves quadratic dependence on $n$. The sample complexity of a potential (not necessarily random-walk based) exact sampling algorithm will depend on $\mathcal{Q}$, but might achieve a sample complexity not directly dependent on $\lambda(\boldsymbol{Q})$. The unclear part is our algorithm's dependence on $n$. While our algorithm attains a bound of $\widetilde{O}(n^2)$, it is not obvious whether an algorithm with sample complexity $o(n^2)$ exists.

Similarly, it is interesting to ask whether there are (efficient) ways to transform the chain in order to get a faster sample corrector. Our learning method finds a vector $\boldsymbol{p}$ (input of Algorithm 2) that makes the re-weighted distribution close to uniform. Is it possible to find another $\boldsymbol{p}$ (without our learning algorithm) such that (i) when transforming the chain with $\boldsymbol{p}$, we obtain a fast mixing chain; and (ii) the success probability in the rejection sampling step will be sufficiently high?

This work provided a theoretical understanding to perfect sampling from pairwise comparisons. We believe it is an interesting direction to experimentally evaluate our proposed methodology.

We mention that this work is purely theoretical and has no negative societal impact.

## Acknowledgments and Disclosure of Funding

We thank the anonymous reviewers for useful remarks and comments on the presentation of our manuscript. This work was supported by the Hellenic Foundation for Research and Innovation (H.F.R.I.) under the "First Call for H.F.R.I. Research Projects to support Faculty members and Researchers and the procurement of high-cost research equipment grant", project BALSAM, HFRI-FM17-1424.

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
