# A  Additional Notation and Preliminaries

## A.1  Notation

In this section, we provide the basic notation we are going to use in the technical part.

**General Notation.**  We define $[n] := \{1, \ldots, n\}$. We denote vectors with small bold letters $\boldsymbol{x}$ and matrices with bold letters $\boldsymbol{P} = [\boldsymbol{P}_{xy}]$. For a vector $\boldsymbol{x}$, we let $x_i = x(i)$ be its $i$-th entry. For $p \in \{1, 2, \infty\}$, we denote the $L_p$ norm by $\|.\|_p$. For a matrix $\boldsymbol{A}$, let $\|\boldsymbol{A}\|_2$ denote its spectral norm (largest singular value). We consider graphs $G = (V, E)$, with $|V| = n$ vertices, whose associated Laplacian matrix is denoted $\boldsymbol{\Lambda}(G)$. The edge set of the graph $G$ is the set $E(G)$. In Appendix A.2.2, we have included some basic definitions and facts about random walks and Markov chains. We define the weighted Laplacian matrix of $G_{\mathcal{Q}}$ for a distribution $\mathcal{Q}$ as

$$\boldsymbol{Q}_{xy} = -\mathcal{Q}(x, y) \text{ for any } x \neq y \text{ and } \boldsymbol{Q}\mathbf{1} = \mathbf{0}, \tag{2}$$

**Distributions and Distances.**  The probability simplex is denoted by $\Delta^n$. We let $\mathrm{supp}(\mathcal{D})$ denote the support of a distribution $\mathcal{D}$. The total variation distance between two discrete distributions $\mathcal{D}_1, \mathcal{D}_2$ over $[n]$ is equal to $\mathrm{TV}(\mathcal{D}_1, \mathcal{D}_2) = \max_{S \subseteq [n]} (\mathcal{D}_1(S) - \mathcal{D}_2(S)) = \frac{1}{2}\|\mathcal{D}_1 - \mathcal{D}_2\|_1$. We let $\mathcal{D}(S) = \sum_{x \in S} \mathcal{D}(x)$ and $\mathcal{D}_S$ is the conditional distribution on the set $S$, i.e., $\mathcal{D}_S(x) = \frac{\mathcal{D}(x)}{\mathcal{D}(S)}\mathbf{1}\{x \in S\}$. The vector $\boldsymbol{z} \in \mathbb{R}^n$ with $z_i = \log(\mathcal{D}(i))$ is called the natural parameter vector of $\mathcal{D}$.

**Random Walks.**  For a reversible transition matrix $\boldsymbol{P}$, let its (real) spectrum be $1 = \lambda_1 \geq \lambda_2 \geq \cdots \geq \lambda_n \geq -1$. We define the *absolute spectral gap* of $\boldsymbol{P}$ to be the difference $\Gamma(\boldsymbol{P}) = 1 - \max\{\lambda_2(\boldsymbol{P}), |\lambda_n(\boldsymbol{P})|\}$. If $\boldsymbol{P}$ is aperiodic and irreducible, then $\Gamma(\boldsymbol{P}) > 0$. One could also define the *spectral gap* $1 - \lambda_2(\boldsymbol{P})$. The two gaps are equal when the chain is lazy, i.e., when $\forall i \in [n]$, $\boldsymbol{P}_{ii} \geq 1/2$. We let $\lambda(\boldsymbol{L})$ be the second *smallest* eigenvalue of a Laplacian matrix $\boldsymbol{L}$, a.k.a., its Fiedler eigenvalue. The Laplacian matrix $\boldsymbol{L} \in \mathbb{R}^{n \times n}$ is positive semi-definite and induces a semi-norm on $\mathbb{R}^n$ with $\|\boldsymbol{v}\|_L := \sqrt{\boldsymbol{v}^T \boldsymbol{L} \boldsymbol{v}}$ for $\boldsymbol{v} \in \mathbb{R}^n$. Recall that a semi-norm differs from a norm in that the semi-norm of a non-zero element is allowed to be zero. In Algorithm 1, Algorithm 2 and Algorithm 7, we use the notation $F_t(x) \leftarrow F_{t+1}(w)$ for some $t < 0$. This means that we append the path of $x$ at time $t$ by adding the transition $x \to w$ and then the path of $w$ from time $t + 1$ up to $0$.

**Matrix Operators.**  We denote by $\boldsymbol{A} \odot \boldsymbol{B} = (\boldsymbol{A}_{ij}\boldsymbol{B}_{ij})_{1 \leq i \leq j \leq n}$ the standard Hadamard matrix product and by $\boldsymbol{A} \circ \boldsymbol{B}$ a variation of the Hadamard matrix product, where the off-diagonal entries are equal to those of the standard Hadamard product $\boldsymbol{A} \odot \boldsymbol{B}$, but the diagonal entries correspond to the diagonal matrix with entries $(-\sum_{j \neq i} \boldsymbol{A}_{ij}\boldsymbol{B}_{ij})_{i \in [n]}$. Finally, we let $\boldsymbol{A} \otimes \boldsymbol{B} = \boldsymbol{I} - \boldsymbol{A} \circ \boldsymbol{B}$.

## A.2  Preliminaries

In this section, we provide some preliminaries and some useful tools about (i) concentration of random matrices, (ii) random walks and (iii) CFTP. The reader may skip this section.

### A.2.1  Random Matrices

We continue with some definitions for random matrices, needed for the proof of Lemma 17. The following can be found at [Tro15].

**Definition 6.**  *Let $(\Omega, \mathcal{F}, \mu)$ be a probability space. A random matrix $\boldsymbol{X}$ is a measurable map $\boldsymbol{X} : \Omega \to \mathcal{M}^{n_1 \times n_2}$.*

The entries of $\boldsymbol{X}$ may be considered complex random variables that may or may not be correlated with each other. A finite sequence $\{\boldsymbol{X}_i\}$ of random matrices is independent when

$$\mu(\boldsymbol{X}_k \in F_k \text{ for each } k) = \prod_k \mu(\boldsymbol{X}_k \in F_k),$$

for any collection $\{F_k\}$ of Borel subsets of $\mathcal{M}^{n_1 \times n_2}$.

**Proposition 7** (Hermitian Matrix Chernoff Bounds (see [Tro15])). *Consider a finite sequence $\{\boldsymbol{X}_i\}$ of $m$ independent, random, Hermitian square matrices with common dimension $n$. Assume that $0 \leq \lambda_{\min}(\boldsymbol{X}_i) \leq \lambda_{\max}(\boldsymbol{X}_i) \leq M$ for any $i \in [m]$. Let $\boldsymbol{Y} = \sum_i X_i$. Then, for $\epsilon \in [0, 1)$:*

$$\Pr\left[\lambda_{\min}(\boldsymbol{Y}) \leq (1-\epsilon)\lambda_{\min}(\mathbb{E}\boldsymbol{Y})\right] \leq n(e^{-\epsilon}/(1-\epsilon)^{1-\epsilon})^{\lambda_{\min}(\mathbb{E}\boldsymbol{Y})/M} \leq n\exp\left(-\epsilon^2 \lambda_{\min}(\mathbb{E}\boldsymbol{Y})/(2M)\right).$$

### A.2.2 Random Walks and Markov Chains

This section is mostly based on [LP17] and we refer the reader to [LP17] for a thorough exposition.

**Markov Chains.** Let $\Omega$ be a finite state space. A Markov chain is a sequence of random variables $X_0, X1, \ldots$ that satisfy the Markov property, i.e.,

$$\Pr[X_{t+1} = x | X_0 = x_0, \ldots, X_t = x_t] = \Pr[X_{t+1} = x | X_t = x_t].$$

A Markov chain is called time-homogeneous, if the RHS of the above equation does not depend of $t$. Such a chain is associated with a transition matrix $\boldsymbol{P} = \{\boldsymbol{P}(x, y)\}$, where $(x, y) \in \Omega \times \Omega$. It holds that

$$\boldsymbol{P}(x, y) = \Pr[X_{t+1} = y | X_t = x] \text{ for all } x, y \in \Omega, t \in \mathbb{N}.$$

A Markov chain is *ergodic* if there exists a time $t \in \mathbb{N}$ such that $\boldsymbol{P}^{(t)}(x, y) > 0$ for any $x, y \in \Omega$, i.e., there exists a finite time $t$ so that the probability of going from any vertex to any other in $t$ steps is positive. For finite state Markov chains, ergodicity is equivalent to irreducibility and aperiodicity. A Markov chain is *irreducible* if for any two states $x, y \in \Omega$, there exists a time step $t$ such that $\boldsymbol{P}^{(t)}(x, y) > 0$ and is *aperiodic* if, for any state $x$, it holds that $\gcd\{t | \boldsymbol{P}^{(t)}(x, x) > 0\} = 1$. Observe that a Markov chain with self-loops is aperiodic.

**Stationary Distribution.** A stationary distribution $\boldsymbol{\pi} \in \Delta^n$, i.e., the $(n-1)$-dimensional simplex whose vertices are the $n$ standard unit vectors, is defined as the fixed point of the transition matrix $\boldsymbol{P}$, that is $\boldsymbol{\pi}^T \boldsymbol{P} = \boldsymbol{\pi}^T$. An ergodic Markov chain has a *unique stationary distribution* $\boldsymbol{\pi}$ and, as $t$ increases, it converges to it. An interesting property of Markov chains is the reversibility. A Markov chain is *reversible* if there exists a distribution $\boldsymbol{\pi}$ that satisfies the detailed balanced equations:

$$\boldsymbol{\pi}(x)\boldsymbol{P}(x, y) = \boldsymbol{\pi}(y)\boldsymbol{P}(y, x) \text{ for all } x, y \in \Omega. \tag{3}$$

In this case, we can verify that $\boldsymbol{\pi}$ is a stationary distribution. We can simply write $\pi$ for the stationary distribution with associated probability vector $\boldsymbol{\pi}$.

**Mixing Time.** It is important to understand the convergence time of a Markov chain to its stationary distribution $\pi$. A crucial random variable for convergence time is the *mixing time*.

**Definition 8** (Mixing Time). *Let $0 < \epsilon < 1/2$. Let $M$ be an ergodic Markov chain on a finite state space $\Omega$ with stationary distribution $\pi$. Then, the mixing time with accuracy $\epsilon$ of $M$ equals:*

$$T_{\mathrm{mix}}(\boldsymbol{P}; \epsilon) = \inf\{t > 0 : \max_{x \in \Omega} \mathrm{TV}(\boldsymbol{P}_x(X_t), \pi) \leq \epsilon\},$$

*where $\boldsymbol{P}_x(X_t)$ is the distribution of the state $X_t$ at time $t$ for starting state $x \in \Omega$.*

For a reversible transition matrix $\boldsymbol{P}$, let its spectrum be:

$$1 = \lambda_1 \geq \lambda_2 \geq \cdots \geq \lambda_n \geq -1.$$

Note that $\lambda_2 < 1$ if and only if the chain is irreducible (exactly one connected component) and $\lambda_n > -1$ if and only if the chain is aperiodic (e.g., not a bipartite graph). We define the *absolute spectral gap* of $\boldsymbol{P}$ to be the difference: $\Gamma(\boldsymbol{P}) = 1 - \max\{|\lambda_2(\boldsymbol{P})|, |\lambda_n(\boldsymbol{P})|\}$. It holds that if $\boldsymbol{P}$ is aperiodic and irreducible, then $\Gamma(\boldsymbol{P})$ is strictly positive. One could also define the spectral gap $\lambda(\boldsymbol{P}) = 1 - \lambda_2(\boldsymbol{P})$. The two gaps are equal when the chain is lazy, i.e., when for any state $x \in \Omega$, it holds that $\boldsymbol{P}(x, x) \geq 1/2$.

**Lemma 9** (Bounding $T_{\mathrm{mix}}$, see [LP17]). *Let $0 < \epsilon < 1$. Assume that the transition matrix $\boldsymbol{P}$ is aperiodic, irreducible and reversible with respect to $\pi$. Then, it holds that:*

$$(t_{\mathrm{rel}} - 1)\log\left(\frac{1}{2\epsilon}\right) \leq T_{\mathrm{mix}}(\boldsymbol{P}; \epsilon) \leq \log\left(\frac{1}{\epsilon\pi_{\min}}\right)t_{\mathrm{rel}},$$

*where $t_{\mathrm{rel}} = 1/\Gamma(\boldsymbol{P})$ is the relaxation time of the Markov chain, i.e., the inverse absolute spectral gap.*

**Coalescencing Random Walks.** Let $\Omega = [n]$. In a coalescing random walk, a set of $n$ particles perform independent discrete time random walks on an undirected connected graph $\mathcal{G} = (V, E)$ with $|V| = n$, with each particle initially placed at a single (distinct) vertex $x \in V$. In each time step, all particles move simultaneously. Whenever two or more particles meet at a vertex, they unite to form a single particle which then continues to make a random walk through the graph. The *coalescence time* $T_{\mathrm{coal}}$ is a random variable and is the first time when all particles coalesce. More formally, we can define *coalescence* as follows.

**Definition 10** (Coalescence of Stochastic processes (see [Hub16])). *Let $\mathcal{A}$ be a collection of stochastic processes, defined over a common index set $\mathcal{I}$ and common state space $\Omega$. If there is an index $i \in \mathcal{I}$ and state $x \in \Omega$ such that, for all stochastic processes $X \in \mathcal{A}$, it holds that $X_i = x$, then we say that the stochastic processes have coalesced.*

**Coalescence Time of Random Walks.** Consider the random variable $X_t^{(i)}$, that indicates the position of the $i$-th particle at time $t$. The coalescence time is equal to:

$$T_{\mathrm{coal}} = \inf_{t > 0} \left\{ X_t^{(i)} = X_t^{(j)} \text{ for any } i \neq j, i, j \in [n] \right\}.$$

We conclude this section with a folklore result, that deals with the possible locations in the complex plane of the eigenvalues of a square matrix $\boldsymbol{A} \in \mathcal{M}_n$.

**Lemma 11** (Geršgorin's Theorem (see [HJ12])). *Let $\boldsymbol{A} = [\boldsymbol{A}_{ij}] \in \mathcal{M}_n$. For any $i \in [n]$, let*

$$R_i(\boldsymbol{A}) = \sum_{j \neq i} |\boldsymbol{A}_{ij}|,$$

*and consider the $n$ Geršgorin disks*

$$B_i = \{ z \in \mathbb{C} : |z - \boldsymbol{A}_{ii}| \leq R_i(\boldsymbol{A}) \}.$$

*Then, the eigenvalues of $\boldsymbol{A}$ are in the union of the Geršgorin disks, i.e.,*

$$\mathrm{spec}(\boldsymbol{A}) \subseteq \bigcup_{i=1}^{n} B_i.$$

### A.2.3 Exact Sampling and CFTP Preliminaries

Markov chain Monte Carlo (MCMC) methods constitute a class of algorithms for sampling from probability measures and arise naturally in various fields of science such as theoretical computer science (e.g., approximation algorithms for #P-complete problems (see [Jer03, JSV04])), statistical physics (e.g., in order to understand phase transition phenomena for Ising models) and statistics. However, the theory of MCMC, in terms of exact sampling, can be seen as an asymptotic analysis (i.e., in the limit, the total variation distance vanishes). Perfect simulation is analogous to MCMC and deals with techniques for designing algorithms that return exact draws from the target distribution, instead of long-time approximations. Exact sampling comprises a well-studied field (see [Ken05, Hub16]) and has numerous applications in computer science (e.g., in approximate counting [Hub98]). The importance of exact simulation gave rise to various procedures in order to generate perfect samples. A small sample of this line of research ([PW96, GM99, Wil00, HS00, FH00, Hub04]).

Coupling from the past (CFTP) is a technique developed by Propp and Wilson ([PW96, PW98]), that provides an exact random sample from a Markov chain, that has law the (unique) stationary distribution. The algorithm assumes that a particle has been running on the Markov chain since time $-\infty$ (arbitrarily long in the past) and we are concerned with the location of the particle at (the fixed) time $t = 0$. The fact that the stopping time is deterministic and not a random variable (as in MCMC) is crucial. Since the particle performs a random walk in the Markov chain infinitely long, one would intuitively believe that the particle at time $t = 0$ is distributed according to the stationary distribution.

We define $F_{(t,0]}$ as the $t$-step evolution of the Markov chain from time $t \in \mathbb{Z}_{\leq 0}$ to the fixed time $T = 0$ (evolution *from the past*). Note that we can decompose the evolution of the chain into $t$ independent applications of the random function $f$, which encodes the information of the random walk, i.e., $\Pr[f(x) = y] = \boldsymbol{P}_{xy}$ for any pair of states $x, y \in \Omega$. Hence, we have that

$$F_{(t,0]} = f_{-1} \circ f_{-2} \circ \ldots \circ f_t.$$

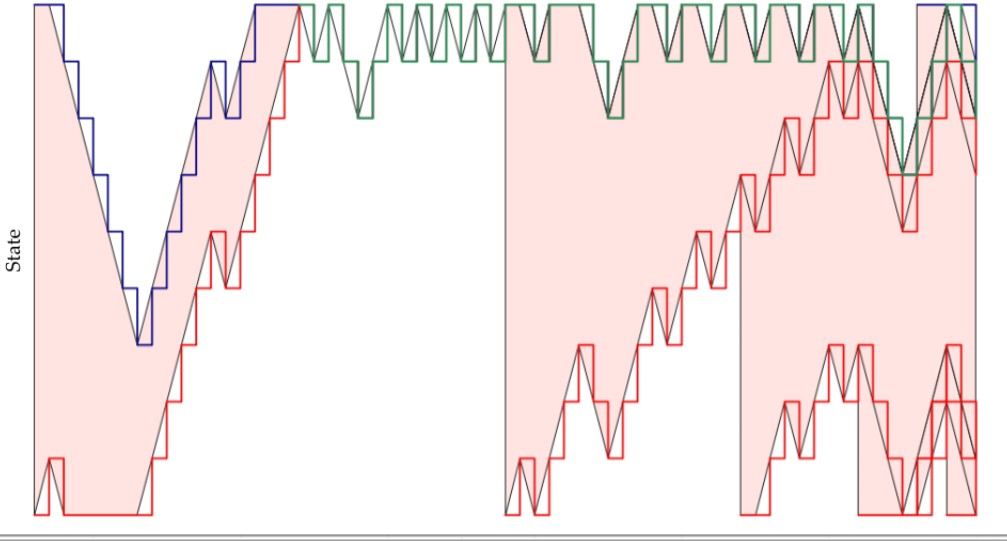

Figure 2: Execution of the CFTP Algorithm for a random walk on the path graph. Observe that it suffices to execute the random procedure only for the two extreme points, since their trajectories dominate any intermediate point of the path. After a series of executions which did not coalesce (see the right part of Figure 2), we observe that the two stochastic evolutions, the blue and the red one in the left part of Figure 2, coalesce and continue as a single trajectory until $t = 0$ (see the green trajectory). A sample execution code can be found at `https://warwick.ac.uk/fac/sci/statistics/staff/academic-research/kendall/personal/perfect_programs/`

---

**Algorithm 4** Coupling From The Past Algorithm

---

 1: **procedure** COUPLINGFROMTHEPAST()  ▷ *Assuming (adaptive) access to the Markov oracle.*
 2:    $t \leftarrow 0$
 3:    $F_{(t,0]}(i) \leftarrow i$, for any $i \in [n]$
 4:    **while** $F_{(t,0]}$ has not coalesced **do**
 5:       $t \leftarrow t - 1$
 6:       $f_t \leftarrow \text{Markov}()$
 7:       $F_{(t,0]} \leftarrow F_{(t+1,0]} \circ f_t$
 8:    **end**
 9:    **return** $F_{(t,0]}(1)$
10: **end procedure**

---

We can think of the data structure $F_{(t,0]}$ as a list of $n$ values, each one capturing the evolution of the Markov chain, initiated in a different state at time $t$ and stopped at time 0. Similarly, the oracle call $\text{Markov}()$ corresponds to $n$ adaptive calls, each one giving a transition from a fixed state $x \in \Omega$, where the next state is distributed according to $\boldsymbol{P}(x, \cdot)$. Hence, $f_t$ is an $n$-dimensional vector with values in $\Omega$. We refer the reader to Figure 2 for an execution of the CFTP algorithm.

The following fact is the main result behind CFTP.

**Fact 1** ([PW96]). *With probability* 1*, the Coupling from the Past algorithm returns a value, and this value is distributed according to the stationary distribution of the Markov chain.*

The following proposition provides an upper bound on the convergence of the CFTP algorithm.

**Proposition 12** ([ZCKM20]). *Let $\pi$ be the stationary distribution of an ergodic Markov chain $\boldsymbol{P}$ with $n$ states. We run $n$ simulations of the chain each starting at a different state. When two or more simulations coalesce, we merge them into a single simulation. With probability at least $1 - \delta$, all $n$ chains have merged after at most $O(nT_{\text{mix}}(\boldsymbol{P}; 1/4) \log(1/\delta))$ iterations.*

# B    The Proof of Theorem 2 (Direct Exact Sampling from LSS)

In this section, we provide the proof of Theorem 2. Let us begin with the algorithm. We remark that in the proof we will use the more technical notation $F_{(a,b]}$ to denote the evolution of the chain from $a$ to $b$. In Algorithm 1, this is abbreviated as $F_t$ to denote $F_{(t,0]}$.

*Proof.* At each iteration, the algorithm draws a single sample from $\mathrm{Samp}(\mathcal{Q}; \mathcal{D})$ Then, the algorithm performs the following procedure for each state $v \in [n]$: let the drawn sample be $(i, j, w)$, where $(i, j) \in \mathcal{E}$ and $w \in \{0, 1\}$ indicating the winning node between $i$ and $j$, i.e., the node $i$ stays in $i$ (if $w = 1$) or node $i$ moves to $j$ (if $w = 0$). For any $v \in [n]$, the algorithm checks whether $v \in \{i, j\}$. If not, the state $v$ remains at $v$; otherwise, assume that $i = v$. Then, the transition $v \to j$ is implemented only if $w = 0$, otherwise it remains at $v$. We claim that this process simulates the transitions of the matrix $\boldsymbol{M}$ and has unique stationary distribution $\mathcal{D}$. The probability that the transition $v \to j$ occurs is

$$\Pr[v \to j] = \Pr[(v, j, w) \text{ is drawn and } w = 0] = \Pr[(v, j) \text{ is drawn}] \cdot \Pr[w = 0|(v, j) \text{ is drawn}]$$

$$= \mathcal{Q}(v, j) \cdot \frac{\mathcal{D}(j)}{\mathcal{D}(v) + \mathcal{D}(j)} = \boldsymbol{M}_{vj}.$$

Hence, for $v \in [n]$ and by the above process, the transition probabilities are given by the matrix $\boldsymbol{M}$, that is a transition matrix with unique stationary distribution $\mathcal{D}$. We will see that this random process gives a perfect sample.

Consider the state space $\Omega = [n]$ and let $T$ be a positive integer. Let $(X_t)_{t \in \mathbb{Z}}$ be a random walk on $[n]$ with transition probabilities as in $\boldsymbol{M}$ (see Equation (1)). Let $F_{(a,b]} : [n] \to [n]$ be such that $F_{(-t_0, T]}(x_0)$ denotes the state of the walk at time $t = T$, i.e., $X_T \in [n]$, when the random walk begun at time $t = -t_0$ (i.e., *in the past*) with initial state $X_{-t_0} = x_0$. Specifically, for each $t_0, T$, we have that $F_{(-t_0, T]} : [n] \to [n]$ is a random map and $F_{(-t_0, T]}(x_0)$ is a realization of the Markov chain, with initial state $x_0$, executed from time $-t_0$ to time $T$. We call $F_{(-t_0, T]}$ a *stochastic flow*.

The CFTP method is initialized at time $t = -t_0$ into the past and places $n$ particles (i.e., simulates $n$ concurrent random walks), one at each state $x \in [n]$. CFTP terminates when the stochastic flow $F_{(-t_0, 0]}$ is constant (it has a single point random image, i.e., $F_{(-t_0, 0]}(x) = F_{(-t_0, 0]}(y)$, for all $x, y$). Namely, CFTP terminates when all the random walks that start at time $t = -t_0$ have coalesced at time $t = 0$ at a single $z \in [n]$. Then, the algorithm outputs the (common) state $z = F_{(-t_0, 0]}(x)$, for any $x \in [n]$.

Since the transitions of each random walk are performed with respect to $\boldsymbol{M}$, the resulting sample $z$ is distributed according to the stationary distribution $\mathcal{D}$. This can be shown as follows (see also [PW96, Hub16]): Let $\mathcal{L}(X)$ denote the distribution of the random variable $X$. Since the chain is aperiodic and irreducible, coalescence occurs almost surely. Specifically, assume that coalescence occurs when the simulation starts at a finite time $t = -t_0$. First, observe that if coalescence occurs at time $-t_0$, then if CFTP starts at any time $-t < -t_0$, we end up in the same state (because the randomness in $[-t_0, 0]$ remains the same), i.e., $F_{(-t, 0]}(x) = F_{(-t_0, 0]}(x)$ for any $-t < -t_0$ and any state $x \in [n]$. Since, coalescence has occurred, we can work with a fixed starting state $x_0$ and let $F_{(-t, 0]} = F_{(-t, 0]}(x_0)$. Hence, it holds that $\mathcal{L}(F_{(-t, 0]}) = \mathcal{L}(F_{(-t_0, 0]})$ and so

$$\mathrm{TV}(\mathcal{L}(F_{(-t_0, 0]}), \mathcal{D}) = \lim_{t \to \infty} \mathrm{TV}(\mathcal{L}(F_{(-t, 0]}), \mathcal{D}).$$

We note that the distribution of the time-backward coalescing random walk is equal to the distribution of the time-forward random walk, i.e., $\mathcal{L}(F_{(-t, 0]}) = \mathcal{L}(F_{(0, t]})$. Thus, $\mathrm{TV}(\mathcal{L}(F_{(-t_0, 0]}), \mathcal{D}) = \lim_{t \to \infty} \mathrm{TV}(\mathcal{L}(F_{(0, t]}), \mathcal{D}) = 0$. So, the output sample of the CFTP algorithm is distributed according to $\mathcal{D}$. Using Proposition 12, we get that, with high probability, coalescence occurs after $O(n T_{\mathrm{mix}}(\boldsymbol{M}; 1/4))$ iterations. From Lemma 9 with $\epsilon = 1/4$, we get that the mixing time of the Markov chain is $O(\log(1/\mathcal{D}_{\min})/\Gamma(\boldsymbol{M}))$. Finally, note that in order to perform a single iteration of the CFTP method, it requires to draw a single sample from the Local Sampling Scheme oracle. This concludes the analysis of the sample complexity. The time complexity is also polynomial in the number of samples [PW96]. $\qquad \square$

**Remark 1.** *As we mentioned in the discussion after Informal Theorem 1, in order to obtain Theorem 2 only Assumption 1 is required. Under only this assumption, a quite large amount of Local Sampling*

Schemes is allowed and the resulting sample complexity is $\widetilde{O}\left(\frac{n\log(1/\mathcal{D}_{\min})}{\Gamma(\boldsymbol{M})}\right)$. However, if we additionally assume that Assumption 2 holds, as we did in the statement of Informal Theorem 1, we accept only LSSs whose target distribution satisfies $\mathcal{D}_{\min} = \Omega(2^{-n})$. This yields the (non-tight) bound $\widetilde{O}(n^2/\Gamma(\boldsymbol{M}))$ of Informal Theorem 1. We preferred to omit this detail in Informal Theorem 1 and emphasize only on the exponential dependence due to $\Gamma(\boldsymbol{M})$ for many natural instances.

## C  The Proof of Theorem 3 (Exact Sampling from LSS using Learning)

The main result of this section is the convergence analysis of the Coupling From the Past algorithm when the algorithm performs the rescaling transformation internally in each transition.

**Theorem 13** (Convergence of CFTP). *Assume that Assumption 1 and Assumption 2 both hold. Algorithm 2, assuming access to a Local Sampling Scheme* $\mathrm{Samp}(\mathcal{Q};\mathcal{D})$ *with weighted Laplacian matrix $\boldsymbol{Q}$ (see Equation (2)) associated with the pair distribution $\mathcal{Q}$ over $[n] \times [n]$, satisfies the following:*

($i$) *Algorithm 2 reaches Line 12 with probability* 1.

($ii$) *For any $\delta \in (0,1)$, a single execution of of Algorithm 2 uses $\widetilde{O}\left(\frac{n\log(n)\log(1/\delta)}{\lambda(\boldsymbol{Q})}\right)$ samples with probability $1 - \delta$ and the running time is polynomial in the number of samples.*

($iii$) *Algorithm 2, with input the output of the Learning Phase, outputs a state $x \in [n]$ with probability proportional to $\mathcal{D}(x)$ or outputs $\perp$. Moreover, Algorithm 3 outputs a perfect sample from $\mathcal{D}$.*

*Using the above properties, for any $\delta > 0$, Algorithm 3 draws, with probability at least $1 - \delta$, $N = O\left(n^2 \log^2(n)/\lambda(\boldsymbol{Q}) \cdot \log(1/\delta)\right)$ samples from a Local Sampling Scheme $\mathrm{Samp}(\mathcal{Q};\mathcal{D})$, runs in time polynomial in $N$, and outputs a sample distributed as in $\mathcal{D}$.*

Proof of *Theorem 13*. Consider a target distribution $\mathcal{D}$, supported on $[n]$ and sample access to a Local Sampling Scheme $\mathrm{Samp}(\mathcal{Q};\mathcal{D})$ for a pair distribution $\mathcal{Q}$. Let $\boldsymbol{M} = \boldsymbol{Q} \otimes \boldsymbol{D}$ be the transition matrix of the Markov chain associated with the Local Sampling Scheme graph. From Theorem 4 with accuracy $\epsilon > 0$, there exists an algorithm that, with high probability, computes the rescaling of the Markov chain that corresponds to $\boldsymbol{M}$, which we denote by $\widetilde{\boldsymbol{M}}$, the transition matrix of the rescaled (by $1/\widetilde{\mathcal{D}}$) Markov chain. The learning algorithm uses $\widetilde{O}(n^2/(\lambda(\boldsymbol{Q})))$ with probability $1 - \delta$ and is executed only once. We now analyze a single execution of Algorithm 2.

**Claim 2** (Termination of CFTP). *Algorithm 2 reaches Line 12 with probability* 1.

*Proof.* Since the chain $\widetilde{\boldsymbol{M}}$ is ergodic, there is a $T$ such that for any pair of states $(x,y)$, the probability $\widetilde{\boldsymbol{M}}^T(x,y) > 0$. Similarly to the proof sketch of the naive exact sampling algorithm (see also Theorem 2), let $(X_t)_{t\in\mathbb{Z}}$ be a random walk on $[n]$ and define the mapping $F_{(a,b)} : [n] \to [n]$, so that $F_{(-t_0, T]}(x_0)$ is the state of the random walk at time $t = T$, i.e., $X_T \in [n]$, where the random walk begun at time $t = -t_0$ (in the past) with starting state $X_{-t_0} = x_0$. Then, the $n$ random walks have coalesced after $T$ steps if and only if $|\mathrm{Image}(F_{(-T,0]})| = 1$. By the ergodicity of the chain, each one of the events $E_T = \{|\mathrm{Image}(F_{(-T,0]})| = 1\}$ for $T \in \mathbb{Z}_{\geq 0}$ has strictly positive probability to occur. Since the events are independent, with probability 1, there will be an event $E_{T^\star}$ that occurs. Then, for any $T > T^\star$, the desired property holds. $\square$

**Claim 3** (Sample Complexity of a Single Iteration). *It holds that each execution of Algorithm 2 uses $O\left(\frac{n\log(n)\log(1/\delta)}{\lambda(\boldsymbol{Q})}\right)$ samples from the Local Sampling Scheme with probability $1 - \delta$.*

*Proof.* The CFTP draws a single sample for each iteration and performs a step according to the matrix $\widetilde{\boldsymbol{M}}$ due to the rescaling. First, the unique stationary distribution is almost uniform (Lemma 15) and so $\log(1/\mathcal{D}_{\min}) = \Theta(\log(n))$. This implies, by Lemma 9 and Lemma 15, that the mixing time of $\widetilde{\boldsymbol{M}}$ for some constant accuracy $\epsilon_0 = 1/4$ is $T_{\mathrm{mix}}(\widetilde{\boldsymbol{M}}; 1/4) = O(\log(n)/\lambda(\boldsymbol{Q}))$. Hence, by Proposition 12, we get the desired result. $\square$

Moreover, the update time of each execution of the CFTP algorithm is polynomial in $n$. Note that, since during the Learning Phase, the algorithm uses $O(n^2/\lambda(\boldsymbol{Q}))$ and is executed once, the sample complexity of Algorithm 3 follows, since Algorithm 2 will be executed $\Theta(n)$ times.

**Claim 4** (Rejection Sampling). *Algorithm 2, with input the output of the Learning Phase, outputs a state $x \in [n]$ with probability proportional to $\mathcal{D}(x)$ or outputs $\perp$. Moreover, Algorithm 3 outputs a perfect sample from $\mathcal{D}$.*

*Proof.* Let us set $A = \sum_{y \in [n]} \mathcal{D}(y)/\widetilde{D}(y)$. At the end of Line 11, the parameterized CFTP algorithm (Algorithm 2) outputs a sample $x \propto \mathcal{D}(x)/\widetilde{D}(x)$. This is because the unique stationary distribution is $\widetilde{\pi}(x) = (\mathcal{D}(x)/\widetilde{\mathcal{D}}(x))/A$ and the utility of the CFTP mechanism guarantees that when the Algorithm initiates a CFTP simulation from time $T = -T^\star$, the generated sample $x = F_{(-T^\star,0]} = F_{(-\infty,0]}$ will be distributed according to $\widetilde{\pi}$. This follows from the observation that the distribution of the state $F_{(-\infty,0]}$ is equal to the distribution of the state $F_{[0,+\infty)}$ obtained by running the simulation up to the limit $T \to +\infty$ forward in time. Hence, we have that $x$ has law the unique stationary distribution $\widetilde{\pi}$. We now have to remove the bias induced by the learning step. For the sample $x$, we perform a rejection sampling process, with acceptance probability $\widetilde{\mathcal{D}}(x)$. The Algorithm 2 has $n+1$ potential outputs: It either prints $x \in [n]$ or $\perp$ indicating failure. The sample $x$ is the output of the algorithm with probability $\frac{\mathcal{D}(x)/\widetilde{\mathcal{D}}(x)}{A} \cdot \widetilde{\mathcal{D}}(x) = \mathcal{D}(x)/A$. This holds for any $x \in [n]$. The remaining probability mass is assigned to $\perp$. Hence, a single execution of Algorithm 2 outputs a point $x \in [n]$ with probability $\mathcal{D}(x)/A$ and outputs "reject" with probability $1 - 1/A$. As we will see, the accepted sample is distributed according to $\mathcal{D}$. This is because, conditional on acceptance, its mass is exactly that assigned by $\mathcal{D}$. Specifically, we claim that the output of Line 6 of Algorithm 3 has law $\mathcal{D}$. Observe that the whole stochastic process of Algorithm 3 outputs $x \in [n]$ with probability

$$\sum_{i=0}^{\infty} \Pr[\text{Reject}]^i \Pr[x \text{ is Accepted}] = \sum_{i=0}^{\infty} \Pr[\text{Reject}]^i \cdot \frac{(\mathcal{D}(x)/\widetilde{\mathcal{D}}(x)) \cdot \widetilde{\mathcal{D}}(x)}{\sum_{y \in [n]} \mathcal{D}(y)/\widetilde{\mathcal{D}}(y)}$$

$$= \frac{\mathcal{D}(x)}{\sum_{y \in [n]} \mathcal{D}(y)/\widetilde{\mathcal{D}}(y)} \sum_{i=0}^{\infty} \Pr[\text{Reject}]^i.$$

We have that $\Pr[\text{Reject}] = \frac{1}{A} \sum_{y \in [n]} \frac{\mathcal{D}(y)}{\widetilde{\mathcal{D}}(y)}(1 - \widetilde{\mathcal{D}}(y)) = 1 - \frac{1}{A} \in (0,1)$. Hence, we have that the whole stochastic process of Algorithm 3 outputs $x \in [n]$ with probability

$$\sum_{i=0}^{\infty} \Pr[\text{Reject}]^i \Pr[x \text{ is Accepted}] = \frac{\mathcal{D}(x)}{\sum_{y \in [n]} \mathcal{D}(y)/\widetilde{\mathcal{D}}(y)} \cdot A = \mathcal{D}(x),$$

since $\sum_{i \geq 0} \lambda^i = \frac{1}{1-\lambda}$ for $\lambda \in (-1,1)$. As a result, we get that the probability that $x$ is the output of the above stochastic process is $\mathcal{D}(x)$. $\square$

These claims complete the proof. Specifically, the total sample complexity is derived as follows: Since we have to execute the CFTP iterations $\Theta(n)$ times, we should call each CFTP process with $\delta' = C\delta/n$ for some constant $C$. Proposition 12 guarantees that the CFTP process will terminate using $O(n \log(n)/\lambda(\boldsymbol{Q}) \cdot \log(n/\delta))$ samples with probability $1 - \Theta(\delta/n)$. Hence, applying the the union bound for the $\Theta(n)$ CFTP calls and the single call of the learning algorithm, gives that, with probability $1 - \delta$, a number of $O(n^2 \log^2(n)/\lambda(\boldsymbol{Q}) \cdot \log(1/\delta))$ samples suffices in order to get a perfect sample from $\mathcal{D}$. $\square$

## D   The Proof of Lemma 5 (Properties of Rescaled Random Walk)

In the next section, we discuss some useful steps as a warm-up for the proof of Lemma 5. The proof can be found at the Appendix D.2.

### D.1   Sketch of the Idea

By downscaling the transition probabilities (as we will see below), we can decouple the Markov chain from $\mathcal{D}$. Then, the transition matrix of the modified Markov chain only depends on the pair

distribution $\mathcal{Q}$, and the convergence of the CFTP algorithm is determined by $\lambda(\boldsymbol{Q})$. Hence, we can sample efficiently even from target distributions $\mathcal{D}$ that may be multimodal or has many low probability points. E.g., in case of a multimodal stationary distribution $\mathcal{D}$, the spectral properties of the walk would remind these of a disconnected graph and the sample complexity of Algorithm 1 would be quite high. We use the estimate $\widetilde{\mathcal{D}}$ (which we obtained from our learning algorithm) of the target distribution $\mathcal{D}$ to transform the Markov chain of the Local Sampling Scheme defined in Equation (1) to a modified Markov chain with an almost uniform stationary distribution. This transformation can be viewed as a downscaling mechanism:

**Definition 14** (Downscaling). *Let $p \in (0, 1)$ and let $X \sim \mathrm{Be}(p)$ be a Bernoulli random variable. Let $\lambda \in (0, 1)$. The random variable $Y \sim \mathrm{Be}(\lambda p)$ is called a $\lambda$-downscaler of $X$.*

Applying Theorem 4, we can consider that, for any $x \in [n]$, there exists a coefficient $\widetilde{\mathcal{D}}(x) \approx \mathcal{D}(x)$. The idea is to use $\widetilde{\mathcal{D}}(x)$ and make the stationary distribution of the modified chain close to uniform. Intuitively, this transformation should speedup the convergence of the CFTP algorithm.

**Step 1.** *Implementation of the downscaling and the rescaled matrix $\widetilde{\boldsymbol{M}}$.*

We can implement the modified Markov chain via downscaling as follows: Consider an edge $\{x, y\}$ with transition probability pair $(p_{xy}, p_{yx})$. Without loss of generality, we assume that $\widetilde{\mathcal{D}}(y) > \widetilde{\mathcal{D}}(x)$ (which intuitively means that we should expect that $p_{xy} > p_{yx}$). Then, the downscaler leaves $p_{yx}$ unchanged and reduces the mass of $p_{xy}$ to make the two transitions almost balanced. Our exact sampling algorithm will perform this downscaling phase to the matrix $\boldsymbol{M}$ (see Equation (1)).

Consider the transition matrix $\boldsymbol{M}$ with $\boldsymbol{M}_{xy} = \mathcal{Q}(x, y) \frac{\mathcal{D}(y)}{\mathcal{D}(x) + \mathcal{D}(y)}$ and $\boldsymbol{M}_{xx} = 1 - \sum_{y \neq x} \boldsymbol{M}_{xy}$. Also, let $\widetilde{\mathcal{D}}$ be an estimate for the distribution $\mathcal{D}$ with some sufficiently small accuracy $\epsilon$, to be chosen (see also Theorem 4). For the pair $(x, y)$, we modify the transition probability $p_{xy} := \boldsymbol{M}_{xy}$, only if $p_{xy} > p_{yx}$, to be equal to the following

$$\widetilde{p}_{xy} = \mathcal{Q}(x, y) \frac{\mathcal{D}(y)}{\mathcal{D}(x) + \mathcal{D}(y)} \frac{\widetilde{\mathcal{D}}(x)}{\widetilde{\mathcal{D}}(y)} \approx \mathcal{Q}(x, y) \frac{\mathcal{D}(x)}{\mathcal{D}(x) + \mathcal{D}(y)} = p_{yx} \,,$$

where we use that $\widetilde{\mathcal{D}}(x) \approx \mathcal{D}(x)$ and $\widetilde{\mathcal{D}}(y) \approx \mathcal{D}(y)$. The transition probability from $x$ to $y$ corresponds to a Bernoulli variable $\mathrm{Be}(p_{xy})$, which is downscaled by $\widetilde{\mathcal{D}}(x)/\widetilde{\mathcal{D}}(y) < 1$. The modified transition probability $\widetilde{p}_{xy}$ can be implemented by drawing a $\Lambda \sim \mathrm{Be}(\widetilde{\mathcal{D}}(x)/\widetilde{\mathcal{D}}(y))$ and then drawing a $P \sim \mathrm{Be}(p_{xy})$ (from $\mathrm{Samp}(\mathcal{Q}; \mathcal{D})$) and, finally, realizing the transition from $x$ to $y$ only if $\Lambda P = 1$. This implementation is valid since the two sources of randomness are independent.

The modified transition matrix $\widetilde{\boldsymbol{M}}$ can be written as:

$$\widetilde{\boldsymbol{M}} = \widetilde{\boldsymbol{D}} \otimes \boldsymbol{Q} + \boldsymbol{Q} \circ [\epsilon_{xy}] \,, \tag{4}$$

where $\widetilde{\boldsymbol{D}}$ is a symmetric matrix with $\widetilde{\boldsymbol{D}}_{xy} = \min\{\mathcal{D}(x), \mathcal{D}(y)\}/(\mathcal{D}(x) + \mathcal{D}(y)) = \widetilde{\boldsymbol{D}}_{yx}$ and $\boldsymbol{Q}$ is the Laplacian matrix of the graph associated with the $\mathrm{Samp}(\mathcal{Q}; \mathcal{D})$ and $\boldsymbol{Q} \circ [\epsilon_{xy}]$ denotes the modified Hadamard product[3] between the Laplacian $\boldsymbol{Q}$ and the matrix with the estimation error $\epsilon_{xy}$ is only non-zero in the modified transitions $x \to y$, i.e., if $p_{xy} > p_{yx}$ then $\widetilde{\boldsymbol{M}}_{xy} = \mathcal{Q}(x, y) \cdot (\frac{\mathcal{D}(x)}{\mathcal{D}(x) + \mathcal{D}(y)} + \epsilon_{xy})$. For this pair, we set $\epsilon_{yx} = 0$. We denote this error matrix with $[\epsilon_{xy}]$.

Specifically, for a pair $(x, y)$, before the downscaling, the pairwise $(x, y)$-comparison corresponds to the random coin $\left( \frac{\mathcal{D}(y)}{\mathcal{D}(x) + \mathcal{D}(y)}, \frac{\mathcal{D}(x)}{\mathcal{D}(x) + \mathcal{D}(y)} \right)$ and let $\mathcal{D}(y) > \mathcal{D}(x)$. After the downscaling, this coin becomes (locally) almost fair, i.e., $(\frac{\mathcal{D}(x)}{\mathcal{D}(x) + \mathcal{D}(y)} + \epsilon_{xy}, \frac{\mathcal{D}(x)}{\mathcal{D}(x) + \mathcal{D}(y)})$.

**Step 2.** *Obtaining a simpler matrix $\widetilde{\boldsymbol{M}}$ with absolute spectral gap of the same order.*

Since the coins are locally fair, we can work with the following matrix that has also almost uniform stationary distribution and conductance of the same order (since making each transition $(x, y)$ more lazy (i.e., increasing the probability of $x \to x$ and $y \to y$ by some *constant*) and still keeping the

---

[3]We have that $\left( \boldsymbol{Q} \circ \boldsymbol{A} \right)_{xy} = \mathcal{Q}_{xy} A_{xy}, \left( \boldsymbol{Q} \circ \boldsymbol{A} \right)_{xx} = -\sum_{y \neq x} \mathcal{Q}_{xy} A_{xy}.$

transitions $x \to y$ and $y \to x$ (almost) equal cannot significantly affect the conductance, i.e., the desired symmetry is preserved). Let $\widetilde{\boldsymbol{M}} = \boldsymbol{I} - c \cdot \boldsymbol{Q} + \boldsymbol{Q} \circ [\epsilon_{xy}]$ for some *constant* $0 \leq c \leq 1/2$. This matrix with $c = \min_{(x,y)\in\mathcal{E}} \mathcal{D}(x)/(\mathcal{D}(x) + \mathcal{D}(y))$ can be obtained by further performing downscaling at each transition (and $c$ is constant due to Assumption 2); then each transition will be equal to the minimum global transition probability (potentially with some $O(1/\sqrt{n})$ noise term). For simplicity, we let $c = 1/2$. The next proof We get the following matrix

$$\widetilde{\boldsymbol{M}}_{xy} = \mathcal{Q}(x,y)\left(\frac{1}{2} + \epsilon_{xy}\right), \ \widetilde{\boldsymbol{M}}_{yx} = \mathcal{Q}(x,y)\left(\frac{1}{2} + \epsilon_{yx}\right), \tag{5}$$

and

$$\widetilde{\boldsymbol{M}}_{xx} = 1 - \sum_{y\neq x} \widetilde{\boldsymbol{M}}_{xy} = 1 - \frac{1}{2}\sum_{y\neq x} \mathcal{Q}(x,y) - \sum_{y\neq x} \epsilon_{xy}\mathcal{Q}(x,y).$$

Also, we observe that the generator of the Markov chain $\boldsymbol{I} - \widetilde{\boldsymbol{M}}$ is close up to scaling to the Laplacian $\boldsymbol{Q}$. This explains why the convergence time of the Markov chain with transition matrix $\widetilde{\boldsymbol{M}}$ spectrally depends only on the distribution $\mathcal{Q}$. Specifically, the Laplacian $\boldsymbol{Q}$ corresponds to the dominant component for the convergence of the algorithm and the Hadamard product is the low-order noise induced by the chain transformation. As we will see, the larger the smallest non-zero eigenvalue of the Laplacian matrix and, hence, the larger the spectral gap of the transition matrix of the transformed chain, the faster the Markov chain converges to its stationary distribution.

### D.2 The Proof of Lemma 5

The following lemma summarizes the key properties of the downscaled random walk.

**Lemma 15** (Properties of Rescaled Random Walk). *Let $\mathcal{D}$ be a distribution on $[n]$ and consider an $\epsilon$-relative approximation $\widetilde{\mathcal{D}}$ of $\mathcal{D}$, as in Theorem 4 with $\epsilon = O(1/\sqrt{n})$. Consider the transition matrix $\boldsymbol{M}$ of the Local Sampling Scheme $\mathrm{Samp}(\mathcal{Q}; \mathcal{D})$ (see Equation (1)), and let $\widetilde{\boldsymbol{M}}$ be the $\widetilde{\mathcal{D}}$-scaling of $\boldsymbol{M}$ (see Equation (4)). Then, the following hold.*

  *(i)* *The transition matrix $\widetilde{\boldsymbol{M}}$ has stationary distribution $\widetilde{\pi}_0(x) = \Theta(1/n)$ for all $x \in [n]$.*

  *(ii)* *The absolute spectral gap $\Gamma(\widetilde{\boldsymbol{M}})$ of $\widetilde{\boldsymbol{M}}$ and the minimum non-zero eigenvalue $\lambda(\boldsymbol{Q})$ of the Laplacian matrix $\boldsymbol{Q}$ satisfy $\Gamma(\widetilde{\boldsymbol{M}}) = \Omega(\lambda(\boldsymbol{Q}))$.*

  *(iii)* *For any $\epsilon_0 \in (0,1)$, the mixing time of the transition matrix $\widetilde{\boldsymbol{M}}$ is $T_{\mathrm{mix}}(\widetilde{\boldsymbol{M}}; \epsilon_0) = O\left(\frac{\log(n/\epsilon_0)}{\lambda(\boldsymbol{Q})}\right)$.*

In the above, we can choose $\epsilon_0 = 1/4$. The proof of the Lemma 15 follows.

Proof of *Lemma 15.* We remark that it suffices to work with the matrix of Equation (5), since the results will only change by at most some constant. Observe that the spectral gap of the matrix of Equation (4) is at least as high as one of the matrix $\boldsymbol{I} - c \cdot \boldsymbol{Q} + \boldsymbol{Q} \circ [\epsilon_{xy}]$ with constant $c = \min_{(x,y)\in\mathcal{E}} \mathcal{D}(x)/(\mathcal{D}(x) + \mathcal{D}(y)) = \Theta(\phi)$ and this matrix has spectral gap of the same order as the matrix of Equation (5). We break the proof into three claims.

**Claim 5.** *The transition matrix $\widetilde{\boldsymbol{M}}$ has stationary distribution $\widetilde{\pi}_0$, that satisfies $\widetilde{\pi}_0(x) = \Theta(1/n)$ for any $x \in [n]$.*

*Proof.* The chain $\widetilde{\boldsymbol{M}}$ remains irreducible and the detailed balance equations of the matrix $\widetilde{\boldsymbol{M}}$ satisfy:

$$\frac{\widetilde{p}_{xy}}{\widetilde{p}_{yx}} = \frac{\mathcal{Q}(x,y)\mathcal{D}(y)/\widetilde{\mathcal{D}}(y)}{\mathcal{Q}(y,x)\mathcal{D}(x)/\widetilde{\mathcal{D}}(x)}.$$

Since $\widetilde{\mathcal{D}}$ is an $\epsilon$-relative approximation of $\mathcal{D}$ with $\epsilon = O(1/\sqrt{n})$, it holds that for any $x \in [n]$:

$$\mathcal{D}(x)/\widetilde{\mathcal{D}}(x) \in [1 - \epsilon, 1 + \epsilon],$$

and hence the unique stationary distribution $\widetilde{\pi}_0$ satisfies: $\widetilde{\pi}_0(x)/\widetilde{\pi}_0(y) \in [1 - \epsilon, 1 + \epsilon]$ for any $x, y \in [n]$. So, this holds for $x = \arg\max \widetilde{\pi}_0$ and $y = \arg\min \widetilde{\pi}_0$ and, since it should hold that $\sum_x \widetilde{\pi}_0(x) = 1$, it must be the case that $\widetilde{\pi}(x) = \Theta(1/n)$. $\qquad \square$

**Claim 6.** *For the absolute spectral gap $\Gamma(\widetilde{M})$ of $\widetilde{M}$ and the minimum non-zero eigenvalue $\lambda(Q)$ of the Laplacian matrix $Q$, it holds that: $\Gamma(\widetilde{M}) = \Omega(\lambda(Q))$.*

*Proof.* For $x \neq y$, there exist error estimates $\epsilon_{xy}$ and $\epsilon_{yx}$; one of them is zero and the other's absolute value is of order $O(1/\sqrt{n})$. Our goal is to control the absolute spectral gap $\Gamma(\widetilde{M})$. Recall that the transition matrix can be written as follows

$$\widetilde{M} = I - \frac{1}{2}Q + Q \circ [\epsilon_{xy}],$$

where $\epsilon_{xy} \in [-\epsilon, \epsilon]$ for some $\epsilon = O(1/\sqrt{n})$. Note that $[\epsilon_{xy}]$ denotes the $n \times n$ error matrix and the operator $\circ$ denotes the standard Hadamard product. Note that the matrix $I - \frac{1}{2}Q$ is symmetric, while $Q \circ [\epsilon_{xy}]$ needs not to be.

Let $\lambda_1 > \lambda_2 \geq \cdots \geq \lambda_n$ be the spectrum of the transition matrix $\widetilde{M}$. Also, the matrix is right stochastic with $\widetilde{M}\mathbf{1} = \mathbf{1}$ and $\lambda_1 = 1$. Let $\Gamma(\widetilde{M})$ be the absolute spectral gap, that is strictly positive by aperiodicity and irreducibility. The Weyl's Inequality for general matrices ([HJ12, Tao12]) describes a multiplicative majorization between the ordered absolute eigenvalues and singular values of $A$ and gives a useful perturbation bound. Recall that $\sigma_i^2(A) = \lambda_i(AA^*) = \lambda_i(A^*A)$ for an arbitrary matrix $A \in \mathbb{C}^{m \times n}$.

**Fact 2** (Weyl's Inequality (see [HJ12])). *Consider the matrices $A, E \in \mathbb{C}^{n \times n}$. Define the singular value of $A$ in decreasing order (counting multiplicity) $\sigma_1(A) \geq \ldots \geq \sigma_n(A) \geq 0$. Then, for $k = 1, \ldots, n$, the following hold*

- $\prod_{i \in [k]} |\lambda_i(A)| \leq \prod_{i \in [k]} \sigma_i(A)$ *with* $|\lambda_1(A)| \geq |\lambda_2(A)| \geq \ldots |\lambda_n(A)|$.

- $|\sigma_k(A + E) - \sigma_k(A)| \leq \|E\|_2$.

For the matrix $\widetilde{M}$, we have that $|\lambda_1| = 1$ and the second largest absolute eigenvalue is $\max\{|\lambda_2|, |\lambda_n|\}$. We can apply the first property for the singular values from the Fact 2 and get that

$$\Gamma(\widetilde{M}) = \lambda_1 - \max\{|\lambda_2|, |\lambda_n|\} \geq 1 - \sigma_1(\widetilde{M})\sigma_2(\widetilde{M}),$$

where $\sigma_1, \sigma_2$ correspond to the two largest singular values. Observe that the largest singular value is $\sigma_1(\widetilde{M}) = \max_{\|v\|_2 = 1} \|\widetilde{M}v\| = 1$ and, using the second property of Fact 2, we can control the perturbation of the second largest singular value

$$\sigma_2\left(I - \frac{1}{2}Q\right) - \|Q \circ [\epsilon_{xy}]\|_2 \leq \sigma_2(\widetilde{M}) \leq \sigma_2\left(I - \frac{1}{2}Q\right) + \|Q \circ [\epsilon_{xy}]\|_2.$$

In order to lower bound the absolute spectral gap, it suffices to upper bound the second largest singular value. But, for the second largest singular eigenvalue of the real symmetric matrix $I - Q/2$, it holds that

$$\sigma_2\left(I - \frac{1}{2}Q\right) = \max\left\{\left|\lambda_2\left(I - \frac{1}{2}Q\right)\right|, \left|\lambda_n\left(I - \frac{1}{2}Q\right)\right|\right\}.$$

Since the matrix $I - Q/2$ is symmetric, we have that

$$\sigma_2\left(I - \frac{1}{2}Q\right) = \max\left\{\left|\max_{v \perp \mathbf{1}, \|v\|_2 = 1} v^\top \left(I - \frac{1}{2}Q\right)v\right|, \left|\min_{\|v\|_2 = 1} v^\top \left(I - \frac{1}{2}Q\right)v\right|\right\},$$

and, hence, if we let $0 = \mu_1 \leq \mu_2 \leq \ldots \leq \mu_n$ be the spectrum of the Laplacian matrix $Q$, we have that

$$\sigma_2\left(I - \frac{1}{2}Q\right) = \max\left\{\left|1 - \frac{1}{2}\mu_2(Q)\right|, \left|1 - \frac{1}{2}\mu_n(Q)\right|\right\}.$$

For the Laplacian matrix $Q$, observe that $|Q_{xx}| = |\sum_{y \sim x} Q(x, y)| \leq 1$. Hence, from Gershgorin's Theorem (see Lemma 11), we get that all the eigenvalues of the Laplacian lie on the real axis (since

the matrix is symmetric) and all the disks $B(x, r)$ are centered at points $|x| \leq 1$ and the radii $r$ are upper bounded by 1. Hence, it holds that $0 \leq \mu_2(\boldsymbol{Q}) \leq \mu_n(\boldsymbol{Q}) \leq 2$. So, we can take that $\sigma_2(\boldsymbol{I} - \frac{1}{2}\boldsymbol{Q}) = 1 - \frac{1}{2}\mu_2(\boldsymbol{Q}) \overset{\text{def}}{=} 1 - \frac{1}{2}\lambda(\boldsymbol{Q}) \geq 0$, where $\lambda(\boldsymbol{Q})$ denotes the smallest non-zero eigenvalue of the Laplacian matrix $\boldsymbol{Q}$.

Now, we have that $\|\boldsymbol{Q} \circ [\epsilon_{xy}]\|_2 = \max_{\|\boldsymbol{v}\|_2=1} \|(\boldsymbol{Q} \circ [\epsilon_{xy}])\boldsymbol{v}\|_2 = O(1/\sqrt{n}) \cdot \mu_n(\boldsymbol{Q}) = O(1/\sqrt{n})$ and so we get $\sigma_2(\widetilde{\boldsymbol{M}}) \leq (1 - \Theta(\lambda(\boldsymbol{Q}))) + O(1/\sqrt{n})$. This implies that

$$\sigma_2(\widetilde{\boldsymbol{M}}) \lesssim \max\{1 - \Theta(\lambda(\boldsymbol{Q})), O(1/\sqrt{n})\},$$

and so

$$\Gamma(\widetilde{\boldsymbol{M}}) \gtrsim 1 - \max\{1 - \Theta(\lambda(\boldsymbol{Q})), O(1/\sqrt{n})\}$$

This implies that $\Gamma(\widetilde{\boldsymbol{M}}) = \Omega(\lambda(\boldsymbol{Q}))$ for sufficiently large $n$. The above proof will be similar for any matrix $\widetilde{\boldsymbol{M}} = I - c \cdot \boldsymbol{Q} + \boldsymbol{Q} \cdot [\epsilon_{xy}]$ for $0 \leq c \leq 1/2$ which is obtained by the discussion of **Step 2**. Such a matrix has an absolute spectral gap that is, on the one side, lower bounded by $\lambda(\boldsymbol{Q})$ (as we showed above) and, on the other side, upper bounded by the absolute spectral gap of the original matrix of Equation (4). This completes the proof. □

We continue with the mixing time result of the transformed Markov chain. Since we have shown that the absolute spectral gap is $\Omega(\lambda(\boldsymbol{Q}))$, we can directly get the desired result, whose proof relies on the analysis of Lemma 9 and can be found, e.g., in [LP17].

**Claim 7.** *The $\epsilon_0$-mixing time of the transition matrix $\widetilde{\boldsymbol{M}}$ is equal to*

$$T_{\text{mix}}(\widetilde{\boldsymbol{M}}; \epsilon_0) = O(\log(n/\epsilon_0)/\lambda(\boldsymbol{Q})).$$

In the above claim, we choose $\epsilon_0 = 1/4$.

Combining these claims, the proof is completed. □

# E    The Proof of Theorem 4 (Main Result of Learning Phase)

The first step of Algorithm 3 is to learn the target distribution $\mathcal{D}$ in $\epsilon$-relative error for some $\epsilon > 0$ and pass it as input to Algorithm 2. In this section, we will provide the learning results for abstract $\epsilon$; however, our algorithm applies these results with $\epsilon = 1/\sqrt{n}$. For two distributions $\mathcal{D}, \widetilde{\mathcal{D}}$ with ground set $[n]$, we introduce the sequence/list (of length $n$) $1 - \mathcal{D}/\widetilde{\mathcal{D}} := (1 - \mathcal{D}(x)/\widetilde{\mathcal{D}}(x))_{x \in [n]}$. Observe that the pair of sequences $(1 - \mathcal{D}/\widetilde{\mathcal{D}}, 1 - \widetilde{\mathcal{D}}/\mathcal{D})$ captures the relative error between the two distributions. The sample complexity of the task of learning $\mathcal{D}$ in $\epsilon$-relative error is summarized by the following (restatement of Theorem 4):

**Theorem.** *For any $\epsilon, \delta > 0$, there exists an algorithm (Algorithm 5) that draws $N = O\left(\frac{n}{\lambda(\boldsymbol{Q})\epsilon^2} \log(\frac{1}{\delta})\right)$ samples from a Local Sampling Scheme $\text{Samp}(\mathcal{Q}; \mathcal{D})$ satisfying Assumptions 1 and 2, runs in time polynomial in $N$, and, with probability at least $1 - \delta$, computes an estimate $\widetilde{\mathcal{D}}$ of the target distribution $\mathcal{D}$, that satisfies the following relative error guarantee $\max\left\{\|1 - \mathcal{D}/\widetilde{\mathcal{D}}\|_\infty, \|1 - \widetilde{\mathcal{D}}/\mathcal{D}\|_\infty\right\} \leq \epsilon$.*

Let us sketch the proof of Theorem 4. For an arbitrary weight vector $\boldsymbol{w} \in \mathbb{R}^n_{>0}$, we use the re-parameterization $z_i = \log(w_i)$. When in addition $\boldsymbol{w} \in \Delta^n$, i.e., $\boldsymbol{w}$ is a probability distribution over $[n]$ and is usually denoted by $\mathcal{D}$, we call $\boldsymbol{z}$ the natural parameter vector of $\mathcal{D}$. Recall that $\boldsymbol{Q}$ is a Laplacian matrix where $\boldsymbol{Q}_{xy} = -\mathcal{Q}(x, y)$, i.e., it is the Laplacian matrix of the graph $G_{\mathcal{Q}}$ weighted by the mass assigned by $\mathcal{Q}$. The proof goes as follows: we draw i.i.d. samples from the LSS. We first apply the following result which is a modification of the results of [SBB+16].

**Theorem 16** (Variant of [SBB+16]). *Let $G_{\mathcal{Q}}$ be the graph of the support $\mathcal{E}$ of $\mathcal{Q}$ satisfying Assumption 1 with $|V| = n$ vertices and associated weighted Laplacian matrix $\boldsymbol{Q}$ (see Equation (2)). Let $\boldsymbol{L}$ be the associated empirical Laplacian matrix with $\mathbb{E}[\boldsymbol{L}] = \boldsymbol{Q}$. Consider a vector $\boldsymbol{z} \in \mathbb{R}^n$ with $\langle \mathbf{1}, \boldsymbol{z} \rangle = 0$ satisfying the constraint $\max_{x,y \in \mathcal{E}} |z_x - z_y| \leq \log(\phi)$ for some constant $\phi > 1$. Let $\text{Ex}(\mathcal{Q}; \boldsymbol{z})$ be the oracle that generates the example $\{(x, y), q\}$ as follows: the edge $(x, y)$ is chosen with probability $\mathcal{Q}(x, y)$ and the bit $q$ is set to 1 with probability $\exp(z_x)/(\exp(z_x) + \exp(z_y))$; otherwise it is set*

---
**Algorithm 5** Learn using Shifting from (Pairwise) Local Sampling Schemes
---
1: **procedure** LEARN-SHIFT$(\epsilon, \delta)$        ▷ *Sample access to oracle* $\mathrm{Samp}(\mathcal{Q}; \mathcal{D})$.
2:      Set $N = \Theta\left(\frac{n}{\lambda(\boldsymbol{Q})\epsilon^2} \log(\frac{1}{\delta})\right)$.
3:      Draw $N$ samples of the form $((x_i, y_i), q_i) \in \mathcal{E} \times \{0, 1\}$.
4:      Obtain $\widehat{\boldsymbol{z}}$ using Algorithm 6 (the estimation vector satisfies $\langle \mathbf{1}, \widehat{\boldsymbol{z}} \rangle = 0$ and must be shifted).
5:      Compute $C = \log\left(\sum_{x \in [n]} \exp(\widehat{z}_x)\right)$
6:      Set $z'_x = \widehat{z}_x - C$ for any $x \in [n]$        ▷ *See Appendix F.2.*
7:      Output $\boldsymbol{z}'$        ▷ The output satisfies $\|\boldsymbol{z} - \boldsymbol{z}'\|_\infty \le \epsilon$.
8: **end procedure**
---

*to 0. For any $\epsilon, \delta > 0$, there is a maximum likelihood estimator of $\boldsymbol{z}$ (Algorithm 6) which draws $N = O\left(\frac{n}{\lambda(\boldsymbol{L})\epsilon^2} \log(\frac{1}{\delta})\right)$ samples from $\mathrm{Ex}(\mathcal{Q}; \boldsymbol{z})$ and computes, in time polynomial in the number of samples $N$, an estimate $\widehat{\boldsymbol{z}}$ such that $\|\boldsymbol{z} - \widehat{\boldsymbol{z}}\|_2 \le \epsilon$, with probability at least $1 - \delta$.*

The proof can be found at Appendix F.1. Some comments are in order:

1. In the work of [SBB$^+$16], the target is the (re-parameterized) weights vector $\boldsymbol{z}^\star \in \mathbb{R}^n$ satisfying the conditions $\langle \mathbf{1}, \boldsymbol{z}^\star \rangle = 0$ and $\|\boldsymbol{z}^\star\|_\infty \le B$ for some constant $B$. The provided algorithm minimizes the empirical log-likelihood and, using $O(n/(\lambda(\boldsymbol{Q}) \cdot \epsilon^2)))$ samples, computes an estimate $\widehat{\boldsymbol{z}}$ so that $\|\boldsymbol{z}^\star - \widehat{\boldsymbol{z}}\|_2 \le \epsilon$.

2. Our provided Algorithm 6 has the same guarantees as the algorithm of [SBB$^+$16] but the target weight vector satisfies the conditions $\langle \mathbf{1}, \boldsymbol{z}^\star \rangle = 0$ and $|z_x^\star - z_y^\star| \le B$ for some constant $B$. The algorithm draws sufficiently many samples and then minimizes the empirical negative log-likelihood objective over an appropriately selected constrained set (see Appendix F.1). Observe that for a Local Sampling Scheme with distribution $\mathcal{D}$, the natural parameter vector $\boldsymbol{z}^\star$ does not satisfy $\langle \mathbf{1}, \boldsymbol{z}^\star \rangle = 0$ and so Theorem 16 and Algorithm 6 cannot be directly applied.

3. We will discuss (Appendix F.2) how to apply Algorithm 6 to distributions, i.e., target weight vectors with $\langle \mathbf{1}, \boldsymbol{z}^\star \rangle \le 0$. This results in learning the target vector $\boldsymbol{z}^\star$ in $L_\infty$ norm (which is sufficient for Algorithm 3).

The algorithm of Theorem 16 follows:

---
**Algorithm 6** Learning from (Pairwise) Local Sampling Schemes
---
1: **procedure** LEARN$(\epsilon, \delta)$        ▷ *Sample access to oracle* $\mathrm{Ex}(\mathcal{Q}; \boldsymbol{z})$.
2:      Set $N = \Theta\left(\frac{n}{\lambda(\boldsymbol{Q})\epsilon^2} \log(\frac{1}{\delta})\right)$.
3:      Draw $N$ samples of the form $((x_i, y_i), q_i) \in \mathcal{E} \times \{0, 1\}$.
4:      Compute the empirical negative log-likelihood objective

$$L_N(\boldsymbol{z}; \{((x_i, y_i), q_i)\}_{i \in [N]}) = -\frac{1}{N} \sum_{i=1}^N \mathbf{1}\{q_i = 1\}z_{x_i} + \mathbf{1}\{q_i = 0\}z_{y_i} - \log(\exp(z_{x_i}) + \exp(z_{y_i})).$$

5:      Minimize $L_N$ using gradient descent in the subspace $\Omega_\phi$.        ▷ *See Appendix F.1.*
6:      Output the guess vector $\widehat{\boldsymbol{z}}$.
7: **end procedure**
---

To complete the proof of Theorem 4, we combine the upcoming Lemma 17 and Lemma 18 with the main result of [SBB$^+$16].

Lemma 17 states that, after drawing sufficiently many samples from $\mathrm{Samp}(\mathcal{Q}; \mathcal{D})$, we can construct an empirical Laplacian matrix $\boldsymbol{L}$, whose Fiedler eigenvalue is of the same order as the one of the unknown matrix $\boldsymbol{Q}$ of the Local Sampling Scheme. Moreover, notice that Lemma 17 implies that, with high probability, the graph induced by the matrix $\boldsymbol{L}$ is connected, i.e., $\lambda(\boldsymbol{L}) > 0$.

**Lemma 17** (Concentration of Empirical Fiedler Eigenvalue). *Let $\epsilon > 0$ and $V = \{v \in \{\pm 1, 0\}^n : v = e_i - e_j$ for any $i < j\}$ and let $\mathcal{Q}$ be a distribution over $V$. Let $Q$ be the Laplacian matrix of $\mathcal{Q}$ (see Equation (2)). There exists an algorithm that uses $O(\log(n/\delta)/(\lambda(Q)\epsilon^2))$ samples from $\mathcal{Q}$ and computes a matrix $L$ that satisfies $|\lambda(L) - \lambda(Q)| \leq \epsilon\lambda(Q)$, with probability at least $1 - \delta$, where $\lambda(\cdot)$ denotes the second smallest eigenvalue of a Laplacian matrix.*

*Proof.* Let $\{X_i = v_i v_i^T\}$ be a finite sequence of $m$ independent symmetric square matrices (of common dimension $n$) with $v_i \sim \mathcal{Q}$, let $L = \frac{1}{m}\sum_{i=1}^m X_i$, and let $M = \lambda_{\max}(X_i) = \Theta(1)$. To deal with the second smallest eigenvalue of the Laplacian matrix, we can project the Laplacian matrix $L$ to the orthogonal complement of the vector $\mathbf{1}$. Under this transformation, we can obtain a matrix $L'$ that has two crucial properties. First, it still is a sum of independent positive semidefinite terms and its minimum eigenvalue coincides with the Fiedler eigenvalue of $L$. To achieve this, as in [Tro15], we introduce the transformation $R \in \mathbb{R}^{(n-1)\times n}$, that satisfies $RR^T = I_{n-1}$ and $R\mathbf{1} = \mathbf{0}$. Hence, it holds that $L' = RLR^T$ and $\mathbb{E}[L'] = RQR^T$. Then,

$$\Pr_{v_{1..m}\sim\mathcal{Q}^{\otimes m}}\left[\lambda(L) \leq (1-\epsilon)\lambda(Q)\right] = \Pr_{v_{1..m}\sim\mathcal{Q}^{\otimes m}}\left[\lambda_{\min}(L') \leq (1-\epsilon)\lambda_{\min}(RQR^T)\right].$$

Using Proposition 7, we get that:

$$\Pr_{v_{1..m}\sim\mathcal{Q}^{\otimes m}}\left[\lambda(L) \leq (1-\epsilon)\lambda(Q)\right] \leq n\left(e^{-\epsilon}/(1-\epsilon)^{1-\epsilon}\right)^{\lambda_{\min}(RQR^T)/M} \leq n\exp\left(-\epsilon^2 m\frac{\lambda(Q)}{2M}\right).$$

To upper bound this probability by $\delta$, it suffices to draw $O(\log(n/\delta)/(\epsilon^2\lambda(Q)))$ samples. So, we get that, with probability at least $1 - \delta$, the Fiedler eigenvalue of the empirical matrix $L$ satisfies $\lambda(L) \in (1 \pm \epsilon)\lambda(Q)$. Moreover, the second smallest eigenvalue of the empirical Laplacian matrix $L$ is strictly positive and thus, the induced graph is connected (recall that the number of connected components in the graph is the dimension of the nullspace of the Laplacian matrix). $\square$

The next lemma guarantees that learning the natural parameters of any distribution in $L_\infty$ norm is sufficient for learning the distribution with small relative error over its support.

**Lemma 18** (Stability of Relative Error). *Let $\mathcal{D}_1, \mathcal{D}_2$ be discrete distributions supported on the ground set $[n]$ and let $z_1, z_2$ be the corresponding natural parameter vectors. For any sufficiently small accuracy parameter $\epsilon > 0$, if it holds that $\|z_1 - z_2\|_\infty \leq \epsilon$, then the distributions $\mathcal{D}_1, \mathcal{D}_2$ are close in relative error, i.e., $\max\left\{\|1 - \mathcal{D}_1/\mathcal{D}_2\|_\infty, \|1 - \mathcal{D}_2/\mathcal{D}_1\|_\infty\right\} \leq \epsilon$.*

Proof of *Lemma 18*. Let $\epsilon > 0$ sufficiently small and assume that $\|z_1 - z_2\|_\infty \leq \epsilon$, i.e.,

$$\max_{x\in[n]}\left|\log\left(\frac{\mathcal{D}_1(x)}{\mathcal{D}_2(x)}\right)\right| \leq \epsilon.$$

For $x > 0$, it holds that $1 - \frac{1}{x} \leq \log_e(x)$[4]. Hence, for any $x \in [n]$, we get that

$$1 - \frac{\mathcal{D}_2(x)}{\mathcal{D}_1(x)} \leq \log\left(\frac{\mathcal{D}_1(x)}{\mathcal{D}_2(x)}\right) \leq \epsilon,$$

and

$$1 - \frac{\mathcal{D}_1(x)}{\mathcal{D}_2(x)} \leq \log\left(\frac{\mathcal{D}_2(x)}{\mathcal{D}_1(x)}\right) \leq \epsilon.$$

This gives that

$$\frac{\mathcal{D}_1(x)}{\mathcal{D}_2(x)} \geq 1 - \epsilon \quad \text{and} \quad \frac{\mathcal{D}_2(x)}{\mathcal{D}_1(x)} \geq 1 - \epsilon.$$

This implies that both ratios are upper bounded by $1 + \epsilon$. For contradiction, assume that

$$\frac{\mathcal{D}_1(x)}{\mathcal{D}_2(x)} > 1 + \epsilon \iff \frac{\mathcal{D}_2(x)}{\mathcal{D}_1(x)} < \frac{1}{1+\epsilon}.$$

But, the Taylor expansion of the function $x \mapsto \frac{1}{1+x}$ is equal to $1 - x + O(x^2)$ for $|x| < 1$. Hence, for sufficiently small $\epsilon$, the desired bound follows. $\square$

---

[4]This inequality holds when the base of the logarithm is $e$, the base that we have assumed that we work with.

To wrap up, the proof of Theorem 4 goes as follows:

of *Theorem 4.* Assume that $z^\star \in \mathbb{R}^n$ is the true natural parameter vector for the target distribution $\mathcal{D}$. By Lemma 18, in order to obtain the desired relative error, it suffices to control $z^\star$ in $L_\infty$. Also, let $z \in \mathbb{R}^n$ be a shifted variant of $z^\star$ so that $\langle \mathbf{1}, z \rangle = 0$, i.e., $z^\star = z + C\mathbf{1}$ for some $C$. First, the algorithm draws $N = \Theta\left(\frac{n}{\lambda(\mathbf{Q})\epsilon^2} \log(\frac{1}{\delta})\right)$ i.i.d. samples from the LSS with Laplacian matrix $\mathbf{Q}$. Using the concentration of the Fiedler eigenvalue (Lemma 17), we have that this number of samples is sufficient to apply Theorem 16. This will yield a vector $\widehat{z}$ so that $\|z - \widehat{z}\|_2 \leq \epsilon$ with probability $1 - \delta$. We then apply the transformation described in Appendix F.2 (which corresponds to Line 5-6 of Algorithm 5). This yields the desired $L_\infty$ bound and completes the proof. □

# F   The Proof of Theorem 16 (Variant of [SBB+16])

In this section, we prove a slight variant of the algorithm of [SBB+16] where the target weight vector satisfies the conditions $\langle \mathbf{1}, z^\star \rangle = 0$ and $|z_x^\star - z_y^\star| \leq B$ for some constant $B$.

## F.1   Description of the Learning Algorithm for Abstract Weight Vectors

In this section, we provide a complete description of the learning algorithm of [SBB+16] (with a slight modification). We remark that the technical steps required in order to establish our result follow the analysis of [SBB+16]. It suffices to present an efficient learning algorithm that estimates the target weights vector $z^\star = (z_1^\star, \ldots, z_n^\star)$ in $L_\infty$ norm. *We underline that for this section, the target vector $z^\star$ is an abstract weight vector and not the natural parameter vector induced by a distribution.* We assume that $z^\star$ lies in the subspace $\Omega_\phi$, defined in Equation (6). Hence, it suffices that the learning algorithm computes an estimate $\widehat{z}$ such that $\|z^\star - \widehat{z}\|_2 \leq \epsilon$, with high probability, for some accuracy parameter $\epsilon > 0$.

of *Theorem 16.* The algorithm minimizes the empirical negative log-likelihood over a subspace $\Omega_\phi \subseteq \mathbb{R}^n$ of the natural parameter vectors. The empirical negative log-likelihood objective with $N$ draws from the Local Sampling Scheme corresponds to the function

$$L_N(z; \{((x_i, y_i), q_i)\}_{i\in[N]}) = -\frac{1}{N}\sum_{i=1}^N \mathbf{1}\{q_i = 1\}z_{x_i} + \mathbf{1}\{q_i = 0\}z_{y_i} - \log(\exp(z_{x_i}) + \exp(z_{y_i})),$$

where $x_i, y_i \in [n]$ and $q_i \in \{0, 1\}$ for any $i \in [N]$. We optimize this objective using gradient descent in the subspace $\Omega_\phi$, where

$$\Omega_\phi = \left\{z \in \mathbb{R}^n : \langle \mathbf{1}, z \rangle = 0, \max_{(x,y)\in\mathcal{E}} |z_x - z_y| \leq \log(\phi)\right\}. \tag{6}$$

The first constraint $\langle \mathbf{1}, z \rangle = 0$ is imposed in order to work in the subspace where Laplacian matrices are positive definite (since any Laplacian matrix has its first eigenvalue equal to 0 with corresponding eigenvector $\mathbf{1}$). The second constraint[5] is equivalent to Assumption 2. Since any valid target distribution satisfies $\frac{1}{\phi} \leq \frac{\mathcal{D}(x)}{\mathcal{D}(y)} \leq \phi$ for $(x, y) \in \mathcal{E}$, we have that

$$-\log(\phi) \leq z_x - z_y \leq \log(\phi) \text{ for any } x, y \in \mathcal{E}.$$

Let $L$ be the empirical Laplacian matrix whose Fiedler eigenvalue $\lambda_2(L)$ is close to the true $\lambda_2(\mathbf{Q})$ (see Lemma 17). We introduce the following notation for the quadratic form $\|v\|_L^2 := v^T L v$ for any vector $v \in \mathbb{R}^n$. The key idea of the algorithm of [SBB+16] is to compute an estimate $\widehat{z}$ in the $\|\cdot\|_L$ semi-norm. Afterwards, by the min-max principle for Hermitian matrices, we have that

$$\|z^\star - \widehat{z}\|_L^2 \geq \lambda_2(L)\|z^\star - \widehat{z}\|_2^2. \tag{7}$$

Hence, the estimation of the true natural parameter vector in $L_2$ norm is directly implied. The analysis for the estimation in the $L$ semi-norm is based on the following fact about $M$-estimators. A proof of this result can be found in [SBB+16].

---

[5]As we mentioned, our learning algorithm almost exactly follows the algorithm of [SBB+16]. The only difference appears in the *second* constraint of the set $\Omega_\phi$, where [SBB+16] optimize over vectors with upper bounded $L_\infty$ norm.

**Fact 3** (see [SBB+16]). *Let $\Omega_1 = \{z \in \mathbb{R}^n : \langle 1, z \rangle = 0\}$ and let $\Omega \subseteq \Omega_1$. Consider the $M$-estimator*

$$\widehat{z} \in \operatorname*{argmin}_{z \in \Omega} \ell(z),$$

*and let $\ell$ be a differentiable objective that is $\kappa$-strongly convex[6] at $z^\star \in \Omega$ with respect to the $L$ semi-norm. Then, it holds that*

$$\|z^\star - \widehat{z}\|_L^2 \leq \frac{1}{\kappa} \|\nabla \ell(z^\star)\|_{L^\dagger},$$

*where $L^\dagger$ is the Moore-Penrose pseudo-inverse of $L$.*

Using the above result, it suffices to verify that the empirical negative log-likelihood objective is strongly convex in the true parameter vector $z^\star \in \Omega_\phi$ and to upper bound the dual norm $\|\nabla L_N(z^\star)\|_{L^\dagger} = \nabla L_N(z^\star)^T L^\dagger \nabla L_N(z^\star)$. Recall that $L$ is the empirical estimate of the true Laplacian matrix $Q$. We continue with two claims, that are sufficient in order to control the $L$ semi-norm of the vector $z^\star - \widehat{z}$.

**Claim 8** (Strong Convexity at $z^\star$). *The empirical negative log-likelihood $L_N(z)$ is $\kappa$-strongly convex at $z^\star \in \Omega_\phi$ with respect to the $L$ semi-norm with $\kappa = \operatorname{poly}(1/\phi)$.*

*Proof.* It suffices to lower bound the quadratic form $w^T \nabla L_N(z) w$ for all vectors $w \in \mathbb{R}^n$ and $z \in \Omega_\phi$. As in Lemma 17, we introduce the measurement vector notation $v_i \in \{-1, 0, 1\}^n$ in order to represent each drawn sample $((x_i, y_i), q_i)$ (recall that the edges in this training set lie in $\mathcal{E}$ and $q_i \in \{0, 1\}$). We set $v_i(x_i) = 2q_i - 1, v_i(y_i) = -v_i(x_i)$ and the other coordinates are set to 0. These measurement vectors are the building blocks of the empirical matrix (see Lemma 17), since we have that

$$L = \frac{1}{N} \sum_{i=1}^N v_i v_i^T := \frac{1}{N} X^T X.$$

Note that the matrix $X \in \{-1, 0, 1\}^{N \times n}$ has the $i$-th measurement vector $v_i^T$ as its $i$-th row. Using this notation and following the computations of [SBB+16] for the Hessian of $L_N$, we get that

$$\nabla^2 L_N(z) = \frac{1}{N} \sum_{i=1}^N \left\{ \mathbf{1}\{v_i = 1\} A_1 + \mathbf{1}\{v_i = 0\} A_0 \right\} v_i v_i^T,$$

where

$$A_1 = (\log F)''(\langle z, v_i \rangle), A_0 = (\log(1 - F))''(\langle z, v_i \rangle) \text{ and } F(x) = 1/(1 + \exp(-x)).$$

The function $F(x) = 1/(1 + \exp(-x))$ is the Bradley-Terry function. The function $F$ is strongly-log concave in the interval $[-\log(\phi), \log(\phi)]$. Specifically, we have that

$$\frac{d^2}{dx^2}\left( -\log F(x) \right) = \frac{e^x}{(1 + e^x)^2} \geq \frac{\phi}{(1 + \phi)^2} =: \kappa(\phi),$$

since the mapping $x \mapsto \exp(x)/(1 + \exp(x))^2$ is symmetric and, so, focusing on the interval $[0, \log(\phi)]$, its minimum is attained at $\log(\phi)$.

The physical interpretation of these properties is that for the pair $i, j$, it does not matter whether $i \succ j$ or $j \succ i$ (symmetry) but if the comparison gap (i.e., $|z_i - z_j|$) attains very large values, then the strong convexity will be very low. Intuitively, note also that, since $\exp(x)/(1 + \exp(x))^2 \leq 1/4$, the

---

[6]A function $f : \mathcal{X} \to \mathbb{R}$ is $\kappa$-strongly convex with respect to a norm $\| \cdot \|$ if, for all $x, y$ in the relative interior of the domain of $f$ and $\lambda \in (0, 1)$, we have

$$f(\lambda x + (1 - \lambda) y) \leq \lambda f(x) + (1 - \lambda) f(y) - \frac{1}{2} \kappa \lambda (1 - \lambda) \|x - y\|^2.$$

If the function $f$ is differentiable, then a second definition for $\kappa$-strong convexity with respect to a norm $\| \cdot \|$ is that for all points $x, y$, we have that $f(y) - f(x) - \langle \nabla f(x), y - x \rangle \geq \kappa \|x - y\|^2$. Recall that the relative interior of a set $S$ is defined as its interior within the affine hull of $S$, i.e., $\operatorname{relint}(S) = \{x \in S : \exists \epsilon > 0, \mathcal{N}_\epsilon(x) \cap \operatorname{aff}(S) \subseteq S\}$.

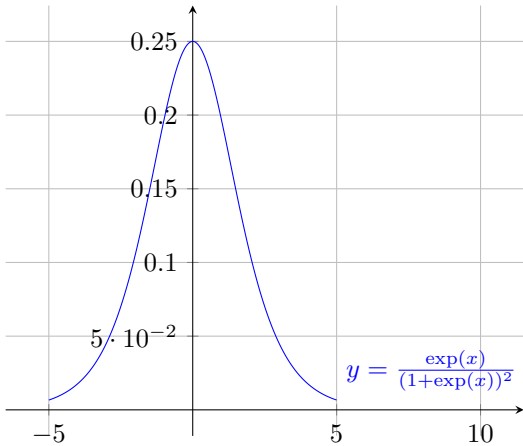

Figure 3: The second derivative of the negative logarithm of the Bradley-Terry function $F$.

desired objective is also smooth and it is known that strong convexity and smoothness of a function over a domain of the form $\{\boldsymbol{z} \in \mathbb{R}^n : \langle \boldsymbol{1}, \boldsymbol{z} \rangle = 0\} \cap \mathcal{Z}$, where $\mathcal{Z}$ is convex, imply that the function satisfies the PL inequality over $\mathcal{Z}$. Hence, for this function, gradient-based methods will converge with fast rates (see e.g., [VYZ20]).

Since any $\boldsymbol{z} \in \Omega_\phi$, we conclude that for any $\boldsymbol{w}$

$$\boldsymbol{w}^T \nabla L_N(\boldsymbol{z}) \boldsymbol{w} \geq \frac{\kappa(\phi)}{N} \|\boldsymbol{X}\boldsymbol{w}\|_2^2,$$

where $\boldsymbol{X}$ the above measurement matrix and $\boldsymbol{z} \in \Omega_\phi$. Hence, if we define $\boldsymbol{\Delta} = \widehat{\boldsymbol{z}} - \boldsymbol{z}^\star$, we get (in order to form the definition of strong convexity) that

$$L_N(\boldsymbol{z}^\star + \boldsymbol{\Delta}) - L_N(\boldsymbol{z}^\star) - \langle \nabla L_N(\boldsymbol{z}^\star), \boldsymbol{\Delta} \rangle \geq \frac{\kappa(\phi)}{N} \|\boldsymbol{X}\boldsymbol{\Delta}\|_2^2 = \kappa(\phi) \|\boldsymbol{\Delta}\|_L^2,$$

since $\|\boldsymbol{\Delta}\|_L^2 = \boldsymbol{\Delta}^T \boldsymbol{L} \boldsymbol{\Delta} = \frac{1}{N} \boldsymbol{\Delta}^T \boldsymbol{X}^T \boldsymbol{X} \boldsymbol{\Delta} = \frac{1}{N} \|\boldsymbol{X}\boldsymbol{\Delta}\|_2^2$. Hence, the empirical negative log-likelihood objective is $\kappa(\phi)$-strongly convex at the true natural parameter vector $\boldsymbol{z}^\star$ with respect to the $\boldsymbol{L}$ semi-norm. $\quad\square$

From the strong-convexity guarantee and, using Fact 3, we get that

$$\|\boldsymbol{\Delta}\|_L^2 \leq \frac{1}{\kappa(\phi)} \|\nabla L_N(\boldsymbol{z}^\star)\|_{L^\dagger}. \tag{8}$$

**Claim 9** (Dual norm). *Let $\boldsymbol{L}$ be the empirical Laplacian estimate matrix of $\boldsymbol{Q}$ and let $\boldsymbol{L}^\dagger$ the Moore-Penrose pseudo-inverse of $\boldsymbol{L}$. There exists a random vector $\boldsymbol{Y} \in \mathbb{R}^n$ such that*

$$\|\nabla L_N(\boldsymbol{z}^\star)\|_{L^\dagger} = \frac{1}{N^2} (\boldsymbol{X}^T \boldsymbol{Y})^T \boldsymbol{L}^\dagger (\boldsymbol{X}^T \boldsymbol{Y}),$$

*where $\boldsymbol{X}$ is the $N \times n$ measurements matrix.*

*Proof.* We first provide a compact form the the gradient $\nabla L_N(\boldsymbol{z}^\star)$. Again, following the analysis of [SBB⁺16], we define a random vector $\boldsymbol{Y} \in \mathbb{R}^n$ with independent coordinates $(Y_1, \ldots, Y_n)$ where we set $Y_i = F'(\langle \boldsymbol{z}^\star, \boldsymbol{v}_i \rangle)/F(\langle \boldsymbol{z}^\star, \boldsymbol{v}_i \rangle)$ with probability $F(\langle \boldsymbol{z}^\star, \boldsymbol{v}_i \rangle)$ and we set $Y_i = -F'(\langle \boldsymbol{z}^\star, \boldsymbol{v}_i \rangle)/(1 - F(\langle \boldsymbol{z}^\star, \boldsymbol{v}_i \rangle))$ otherwise. Recall that $F$, in our setting, is the Bradley-Terry function $F(x) = 1/(1 + \exp(-x))$ and $\boldsymbol{v}_i \in \{-1, 0, 1\}^n$ is the $i$-th comparison vector indicating the drawn edge. Hence, after computing the gradient at the true parameter vector $\boldsymbol{z}^\star$, one gets

$$\nabla L_N(\boldsymbol{z}^\star) = -\frac{1}{N} \boldsymbol{X}^T \boldsymbol{Y}.$$

Claim 9 follows. $\quad\square$

We now introduce the matrix $\boldsymbol{J} := \frac{1}{\kappa(\phi)N^2}\boldsymbol{X}\boldsymbol{L}^\dagger\boldsymbol{X}^T$. Combining Equation (8) with Claim 9, we get that

$$\|\boldsymbol{z}^\star - \widehat{\boldsymbol{z}}\|_L^2 \leq \boldsymbol{Y}^T\boldsymbol{J}\boldsymbol{Y}.$$

Hence, in order to bound the $\boldsymbol{L}$ semi-norm estimation error, it suffices to control the quadratic form of the right hand side. Similarly to [SBB$^+$16], our goal is to apply the Hanson-Wright inequality (see Lemma 19) in order to control the quadratic form.

**Lemma 19** ([HW71, RV13]). *Let $\boldsymbol{Y} \in \mathbb{R}^n$ be a random vector with independent zero-mean components, which are sub-Gaussian with parameter $K$ and let $\boldsymbol{J} \in \mathbb{R}^{n\times n}$ be an arbitrary matrix. Then there is a universal constant $c > 0$ such that*

$$\Pr\left[\left|\boldsymbol{Y}^T\boldsymbol{J}\boldsymbol{Y} - \mathbb{E}[\boldsymbol{Y}^T\boldsymbol{J}\boldsymbol{Y}]\right| > t\right] \leq 2\exp\left(-c\min\left\{\frac{t^2}{K^4\|\boldsymbol{J}\|_F^2}, \frac{t}{K^2\|\boldsymbol{J}\|_2}\right\}\right),$$

*for any $t > 0$.*

Observe that $\mathbb{E}[\boldsymbol{Y}] = 0$ and, for any $i \in [n]$, we have that

$$|Y_i| \leq \sup_{\boldsymbol{z}\in\Omega_\phi}\sup_{\boldsymbol{x}\in\mathbb{X}}\max\left\{\frac{F'(\langle\boldsymbol{z},\boldsymbol{x}\rangle)}{F(\langle\boldsymbol{z},\boldsymbol{x}\rangle)}, \frac{F'(\langle\boldsymbol{z},\boldsymbol{x}\rangle)}{1 - F(\langle\boldsymbol{z},\boldsymbol{x}\rangle)}\right\},$$

where $\mathbb{X} \subseteq \{-1, 0, 1\}^n$ is the space of all valid measurement vectors (they should induce edges in $\mathcal{E}$). Hence, we get that

$$|Y_i| \leq \max_{x\in[-\log(\phi),\log(\phi)]}\max\left\{\frac{\exp(-x)/(1+\exp(-x))^2}{1/(1+\exp(-x))}, \frac{\exp(-x)/(1+\exp(-x))^2}{\exp(-x)/(1+\exp(-x))}\right\} =: u(\phi).$$

Note that $u(\phi) \leq 1$, since, for $0 < b < 1$, it holds that $\max\left\{\frac{1}{b}, \frac{1}{1-b}\right\} \leq \frac{1}{b(1-b)}$. Since each $|Y_i|$ is bounded, we get that the random variables are $u(\phi)$-sub-Gaussian. Also, we have that

$$\mathbb{E}[\boldsymbol{Y}^T\boldsymbol{J}\boldsymbol{Y}] \leq \mathbb{E}\left[\|\boldsymbol{Y}\|_\infty^2\mathrm{tr}(\boldsymbol{J})\right] \leq \frac{u(\phi)^2}{\mathrm{poly}(\kappa(\phi))}\frac{n}{N}.$$

It remains to compute the two matrix norms. As in [SBB$^+$16], one gets that

$$\|\boldsymbol{J}\|_F^2 = \sum_{i,j}|J_{ij}|^2 = \frac{n-1}{\mathrm{poly}(\kappa(\phi))N^2} = \Theta(n/N^2),$$

and

$$\|\boldsymbol{J}\|_2 = \max_{\boldsymbol{x}\neq\boldsymbol{0}}\|\boldsymbol{J}\boldsymbol{x}\|_2/\|\boldsymbol{x}\|_2 = \frac{1}{\mathrm{poly}(\kappa(\phi))N} = \Theta(1/N).$$

Applying the Hanson-Wright concentration inequality, we conclude that

$$\Pr\left[\|\boldsymbol{z}^\star - \widehat{\boldsymbol{z}}\|_L^2 > c\cdot t\frac{u(\phi)^2}{\mathrm{poly}(\kappa(\phi))}\frac{n}{N}\right] \leq e^{-t},$$

for all $t \geq 1$ and some universal constant $c > 0$. Recall that $\phi$ is the constant of Assumption 2 and hence the term $\frac{u(\phi)^2}{\mathrm{poly}(\kappa(\phi))} \in \Theta(1)$. Finally, note that integrating the above tail bound, one gets the desired on expectation bound. $\qquad\square$

### F.2 Application of Algorithm 6 to Distributions

Given a target distribution $\mathcal{D}$, the associated natural parameters vector $\boldsymbol{z} = \log(\mathcal{D})$ satisfies $\langle\boldsymbol{1},\boldsymbol{z}\rangle \leq 0$. Hence, we have to shift it appropriately in order to apply Algorithm 6. Recall that the condition $\langle\boldsymbol{1},\boldsymbol{z}\rangle = 0$ is only useful for identifiability purposes of the BTL model.

Given a target distribution $\mathcal{D} \in \Delta^n$, we convert it to the following weights vector: let $A = \sum_{x\in[n]}\log(\mathcal{D}(x))$ and set $\mathcal{D}'(x) = \exp(-A/n)\mathcal{D}(x)$. So, we have to shift $\boldsymbol{z}$ in the direction $(-A/n)\cdot\boldsymbol{1}$ and obtain the weight vector $\boldsymbol{z}'$. For the weights $(\mathcal{D}'(x))_{x\in[n]}$ with weight vector $\boldsymbol{z}'$, it holds that

1. $\langle \mathbf{1}, \mathbf{z}' \rangle = \sum_{x \in [n]} (-A/n + \log(\mathcal{D}(x))) = 0$ and

2. for any $x, y \in [n]$, it holds that $\mathcal{D}'(x)/\mathcal{D}'(y) = \mathcal{D}(x)/\mathcal{D}(y)$.

Given an estimate for $\mathbf{z}'$, one can extract a good approximation for the natural parameter vector $\mathbf{z}$ of $\mathcal{D}$. In particular, the following properties hold: first, given two distributions $\mathbf{p} \neq \mathbf{q}$, it holds that $\mathbf{z}'(\mathbf{p}) \neq \mathbf{z}'(\mathbf{q})$, i.e., any $\mathbf{z}'$ is uniquely identified by its distribution since one of the following will hold: if $\sum_{i \in [n]} \log(p_i) = A = \sum_{i \in [n]} \log(q_i)$, then since there exists $j \in [n]$ so that $p_j \neq q_j$, we have that $z(\mathbf{p})'_j = \log(p_j) - A/n \neq \log(q_j) - A/n = z(\mathbf{q})'_j$. Otherwise, if, without loss of generality, $A_p = \sum_i \log(p_i) > \sum_i \log(q_i) = A_q$, we have that there exists $j \in [n]$ so that $p_j < q_j$ and so $z(\mathbf{p})'_j = \log(p_j) - A_p/n < \log(q_j) - A_q/n = z(\mathbf{q})'_j$. Hence, any distribution $\mathcal{D}$ uniquely induces a vector $\mathbf{z}'$. Moreover, the actual natural parameter vector $\mathbf{z}$ can be estimated having an estimation for $\mathbf{z}'$. Assume that we have an estimate $\widehat{\mathbf{z}'}$ that satisfies $\|\mathbf{z}' - \widehat{\mathbf{z}'}\|_2 \leq \epsilon$. We can estimate $\mathbf{z}$ using a vector $\widehat{\mathbf{z}}$ (that can be extracted from our estimate $\widehat{\mathbf{z}'}$) as follows: Let $\mathbf{z}' = \mathbf{z} + C\mathbf{1}$, where $C = -A/n$. In order to initiate our exact sampling algorithm, an $L_\infty$ estimate for $\mathbf{z}$ is only required. We have that

$$\|\mathbf{z} - \widehat{\mathbf{z}}\|_\infty = \|(\mathbf{z}' - C\mathbf{1}) - (\widehat{\mathbf{z}'} - C'\mathbf{1})\|_\infty \leq \|\mathbf{z}' - \widehat{\mathbf{z}'}\|_\infty + |C - C'|,$$

where the estimate $\widehat{\mathbf{z}'}$ satisfies $\|\mathbf{z}' - \widehat{\mathbf{z}'}\|_\infty \leq \|\mathbf{z}' - \widehat{\mathbf{z}'}\|_2 \leq \epsilon$ via Algorithm 6 and the shift constant can be estimated since we know that $\mathcal{D}$ is a distribution:

$$\sum_{x \in [n]} \exp(\widehat{z}_x) = 1 \iff \sum_{x \in [n]} e^{\widehat{z'}_x - C'} = 1 \iff e^{C'} = \sum_{x \in [n]} e^{\widehat{z'}_x}.$$

Note that the right hand side of the above equation contains only our estimates. We have that $\sum_{x \in [n]} \exp(z_x) = 1$ and so we set $e^C := \sum_{x \in [n]} e^{z_x}$. Since it holds that $|z'_x - \widehat{z'}_x| \leq \epsilon$ for any $x \in [n]$, we have that $|C - C'| \leq O(\epsilon)$. This gives that for the actual natural parameters it holds that $\|\mathbf{z} - \widehat{\mathbf{z}}\|_\infty \leq O(\epsilon)$.

Hence, given a distribution $\mathcal{D}$, we can learn the weights $\mathbf{z}'$ in $L_2$ norm, satisfying the conditions of Algorithm 6. From this estimation, we can get the estimation for the natural parameters $\mathbf{z}$ by removing the introduced shift. We remark that the above shifting methodology can be applied to any score vector $\mathbf{z}$ satisfying $\langle \mathbf{1}, \mathbf{z} \rangle = B \neq 0$.

## G  Local Sampling Schemes and Hypergraphs

In this section, we discuss how to extend our analysis to sets of size larger that 2. The hypergraph structure of Local Sampling Schemes can be settled as follows:

**Definition 20** (Hypergraph Structure of LSS). *Let $\mathcal{Z}$ be a finite discrete domain and let $\mathcal{Q}$ be a distribution supported on subsets of $\mathcal{Z}$. Then, the hypergraph $G = (V, E)$ with vertex set $V = \mathcal{Z}$ and hyperedge set $E = \mathrm{supp}(\mathcal{Q})$ is called a Local Sampling Scheme hypergraph. If $\mathcal{Q}$ is a pair distribution supported on $\mathcal{Z} \times \mathcal{Z}$, $G$ corresponds to a graph.*

The general Markov Chain for distributions $\mathcal{Q}$, supported on sets $S \subseteq 2^{[n]}$ has the following transition probabilities:

$$p_{xy} = \sum_{S \supseteq \{x,y\}} \mathcal{Q}(S) \frac{\mathcal{D}(y)}{\mathcal{D}(S)} = \mathcal{D}(y) \sum_{S \supseteq \{x,y\}} \frac{\mathcal{Q}(S)}{\mathcal{D}(S)}. \tag{9}$$

For the transition $x \to y$, we can think of a flow $f_{xy}$ with mass $p_{xy}$ and, hence, induce a (simple) graph structure on a flow graph $F = (f_{xy})_{xy}$. Observe that any pair $(x, y)$ shares the same collection of hyperedges and, hence, we have that:

$$\frac{p_{xy}}{p_{yx}} = \frac{\mathcal{D}(y)}{\mathcal{D}(x)} \frac{\sum_{S \supseteq \{x,y\}} \mathcal{Q}(S)/\mathcal{D}(S)}{\sum_{S \supseteq \{x,y\}} \mathcal{Q}(S)/\mathcal{D}(S)} = \frac{\mathcal{D}(y)}{\mathcal{D}(x)}.$$

The ergodicity of the flow graph implies that the stationary distribution corresponds to $\mathcal{D}$ and is unique. For what follows, we consider the case where any hyperedge has size $k$, i.e., $k$-uniform hypergraphs. The Laplacian of the distribution $\mathcal{Q}$ can be generalized over hyperedges and have that

$$\mathbf{Q}_{xy} = - \sum_{S \supseteq \{x,y\}} \mathcal{Q}(S) \text{ for any } x, y \in [n] \text{ and } \mathbf{Q}_{xx} = (k-1) \sum_{S \ni x} \mathcal{Q}(S) \text{ for any } x \in [n].$$

In the following section, we describe some necessary notation.

## G.1 Notation

We denote the set $\binom{[n]}{k}$ the family of size $k$ subsets of the ground set $[n]$. Let $\mathcal{Q}$ be a distribution on $\binom{[n]}{k}$. Such a distribution will be called a $k$-set distribution. Recall that a pair distribution is simply a 2-set distribution. A sample from the Local Sampling Scheme $\mathrm{Samp}_k(\mathcal{Q}; \mathcal{D})$, where $\mathcal{Q}$ is a $k$-set distribution, will be denoted

$$(S, \boldsymbol{v}) \sim \mathrm{Samp}_k(\mathcal{Q}; \mathcal{D}),$$

where $S \in \mathrm{supp}(\mathcal{Q})$ and, in the case where $i \in S$ won between the elements of $S$, we have that $\boldsymbol{v} = \boldsymbol{e}_i \in \mathbb{R}^n$ ($\boldsymbol{v}$ is the indicator vector of the winning node) and we have that $\Pr[\boldsymbol{v} = \boldsymbol{e}_i] = \mathcal{D}(i)/\mathcal{D}(S)\mathbf{1}\{i \in S\}$. For the hypergraph case, the canonical transition matrix induced by the Local Sampling Scheme will be denoted by $\boldsymbol{P}$. In the case $k = 2$, this matrix was denoted by $\boldsymbol{M}$ (recall Equation (1)). Let $\boldsymbol{P}$ denote the transition matrix of the Markov chain, associated with the Local Sampling Scheme $\mathrm{Samp}_k(\mathcal{Q}; \mathcal{D})$, where $\mathcal{Q}$ corresponds to a $k$-set distribution. The entries of $\boldsymbol{P} = [\boldsymbol{P}_{xy}]_{x,y \in [n]}$ are defined as:

$$\boldsymbol{P}_{xy} = \sum_{S \supseteq \{x,y\}} \mathcal{Q}(S) \frac{\mathcal{D}(y)}{\mathcal{D}(S)} \ \text{ for } x \neq y \ \text{ and } \ \boldsymbol{P}_{xx} = 1 - \sum_{y \neq x} \boldsymbol{P}_{xy} \ \text{ for } x \in [n]. \tag{10}$$

Observe that the transition from $x$ to $y$ is performed when

1. a hyperedge $S \in \mathrm{supp}(\mathcal{Q})$ is chosen (with probability $\mathcal{Q}(S)$) and both $x$ and $y$ lie in $S$,
2. and the vertex $y \in [n]$ is the 'winning' node among the nodes of $S$, i.e., $y$ is drawn from the conditional distribution $\mathcal{D}_S$.

There is a natural reduction to the graph case. Consider the marginals of $\mathcal{Q}$ to 2-sets (i.e., edges). Then, one can define a Markov chain over a graph with $[n]$ nodes, whose transition probabilities are described by $\boldsymbol{P}$. The random walk has the following properties: Since any pair of vertices shares the same collection of hyperedges, we get that

$$\frac{\boldsymbol{P}_{xy}}{\boldsymbol{P}_{yx}} = \frac{\mathcal{D}(y)}{\mathcal{D}(x)} \frac{\sum_{S \supseteq \{x,y\}} \mathcal{Q}(S)/\mathcal{D}(S)}{\sum_{S \supseteq \{x,y\}} \mathcal{Q}(S)/\mathcal{D}(S)} = \frac{\mathcal{D}(y)}{\mathcal{D}(x)},$$

for any pair of vertices. Moreover, the Markov chain is ergodic, since it is irreducible, since by the structure of $\mathcal{Q}$ and since $\mathcal{D}$ is supported on $[n]$, one eventually can get from every state to every other state with positive probability; and aperiodic, since it contains self-loops. Hence, it has a unique stationary distribution which coincides with $\mathcal{D}$, using the detailed balance equations.

**Modifications of the LSS Conditions.** We can modify the information-theoretic connectivity condition (Assumption 1) for $k$-uniform hypergraphs and let $\mathcal{E}$ the support of $\mathcal{Q}$. Also, we have to consider the variation of Assumption 2 over the support $\mathcal{E}$ with

$$1/\phi \leq \max_{(x,y) \in S \in \mathcal{E}} \mathcal{D}(x)/\mathcal{D}(y) \leq \phi,$$

for some constant $\phi$. We remark that the learning results of [SBB+16] hold for $k = O(1)$. Similar, our learning tools which are modifications of the work of [SBB+16] hold for the same regime.

## G.2 Learning Phase for Hypergraphs

As in the learning phase of the 2-set case, the learning algorithm for the $k$-set problem is essentially a variation of the analysis of [SBB+16], but the steps are similar. Consider $N$ i.i.d. samples $(S_i, \boldsymbol{v}_i)$ drawn from the sampling oracle $\mathrm{Samp}_k(\mathcal{Q}; \mathcal{D})$ with natural parameter vector $\boldsymbol{z}^\star \in \mathbb{R}^n$. The analysis that follows assumes that any $k$-set sample in the empirical likelihood lies in $\mathcal{E}$, i.e., the support of $\mathcal{Q}$. Our goal is to estimate the true parameter vector. We consider the negative empirical log-likelihood objective

$$L_N(\boldsymbol{z}; \{S_i, \boldsymbol{v}_i\}_{i \in [N]}) = -\frac{1}{N} \sum_{i=1}^N \langle \boldsymbol{z}, \boldsymbol{v}_i \rangle - \log \sum_{j \in S_i} \exp(z_j),$$

and we optimize it over the parameter space

$$\Omega_\phi = \{\boldsymbol{z} \in \mathbb{R}^n : \langle \mathbf{1}, \boldsymbol{z} \rangle = 0, \ \max_{(x,y) \in S \in \mathcal{E}} |z_x - z_y| \leq \log(\phi)\}.$$

**Likelihood Objective and PL model.** We now observe that the above likelihood objective is directly connected to the Plackett-Luce model, which captures the process of choosing a single alternative from a given set, i.e., given a set $S$ of $m$ alternatives with values $w_1, \ldots, w_m$, the likelihood of choosing the $i$-th item is

$$F(w_i, w_1, ..., w_{i-1}, w_{i+1}, ..., w_m) = \exp(w_i) / \sum_{j=1}^{m} \exp(w_j).$$

Note that the negative log-likelihood of the Plackett-Luce model is exactly the same as the single sample version of our objective function. Observe that this function is shift-invariant and its value is independent of the ordering of the last $(m-1)$ elements. Shift invariance is crucial: if one does not work in the subspace $\{z : \langle 1, z \rangle = 0\}$, then neither our problem nor the problem of determining the values of the alternatives in the Plackett-Luce model are identifiable, since any solution of the form $z^\star + c1$ is valid. Technically, shift invariance implies that $1$ lies in the nullspace of the Hessian of the negative log-likelihood (and this is where the first constraint of the set $\Omega_\phi$ arises).

**Hessian matrix.** Let us introduce the vector $e^z(S) = (\exp(z_i))_{i \in S} \in \mathbb{R}^k$ for an arbitrary set $S \subseteq \binom{[n]}{k}$. After standard computations (see also [SBB$^+$16]), we get that

$$\nabla_z^2 \Big( -\langle z, v \rangle + \log \sum_{j \in S} \exp(z_j) \Big) = \frac{\Big( \sum_{j \in S} \exp(z_j) \Big) \mathrm{diag}(e^z(S)) - e^z(S) e^z(S)^T}{\Big( \sum_{j \in S} \exp(z_j) \Big)^2},$$

and hence

$$\nabla_z^2 L_N(z; \{S_i, v_i\}_{i \in [N]}) = \frac{1}{N} \sum_{i=1}^{N} \frac{\langle e^z(S), 1 \rangle \mathrm{diag}(e^z(S_i)) - e^z(S_i) e^z(S_i)^T}{\Big( \langle e^z(S_i), 1 \rangle \Big)^2}.$$

**Quantitative Strong Convexity of PL model.** Consider an arbitrary direction $v \in \mathbb{R}^k$. Without loss of generality, assume that $S = [k]$. Then, it holds that

$$v^T \Big( e^z(S) e^z(S)^T \Big) v = \sum_{i,j=1}^{k} v_i v_j \exp(z_i + z_j) \leq \sum_{i=1}^{k} \exp(z_i) \sum_{j=1}^{k} v_j^2 \exp(z_j) = \langle e^z(S), 1 \rangle \mathrm{diag}(e^z(S)),$$

using the Cauchy-Schwarz inequality, where equality holds if and only if $v \in \mathrm{span}(1)$. Hence, for the Plackett-Luce function $F : \mathbb{R}^k \to \mathbb{R}$, we have that

$$\lambda_2(-\log(F(w))) > 0, \quad \text{for any } w \in \Omega_\phi.$$

In order to quantify the strong convexity parameter, we will make use of the second constraint. For the Plackett-Luce function $F$, we have that

$$\nabla^2(-\log(F(w))) \succeq H,$$

for some $k \times k$ symmetric matrix $H$ with $\lambda_2(H) > 0$. We can observe that the matrix

$$H = \beta(\phi)(I - 11^T),$$

where $\beta(\phi) = \min_{z \in \Omega_\phi} \lambda_2 \left( \frac{\langle e^z(S), 1 \rangle \mathrm{diag}(e^z(S)) - e^z(S) e^z(S)^T}{\Big( \langle e^z(S), 1 \rangle \Big)^2} \right)$, satisfies the strong log-concavity condition for the Plackett-Luce model, i.e., the function $F : \mathbb{R}^k \to \mathbb{R}$.

**Underlying Laplacian matrix and estimation.** Similarly to the case $k = 2$, our goal is to establish strong convexity for the empirical negative log-likelihood around the true parameters $z^\star \in \Omega_\phi$ with respect to the $L$ semi-norm. Hence, we have to first introduce the appropriate Laplacian matrix $L$. In the $k$-set setting, we have that

$$Q_{x,y} = -\sum_{S \ni x,y} \mathcal{Q}(S) \text{ for any } x, y \in [n], \ Q_{x,x} = (k-1) \sum_{S \ni x} \mathcal{Q}(S) \text{ for any } x \in [n].$$

Observe that $\boldsymbol{Q}\mathbf{1} = 0$ (since each set $S \subseteq \binom{[n]}{k}$ that contains $x$, also contains other $k-1$ elements). Recall that for the case $k = 2$, the estimate for the Laplacian matrix $\boldsymbol{Q}$ is given by the empirical matrix $\boldsymbol{L} = \frac{1}{N}\sum_{i=1}^{N}\boldsymbol{v}_i\boldsymbol{v}_i^T = \frac{1}{N}\boldsymbol{X}^T\boldsymbol{X}$ (see Appendix F.1). In the case $k > 2$ case, the Laplacian estimate for the matrix $\boldsymbol{Q}$ is given by the $n \times n$ matrix

$$\boldsymbol{L} = \frac{1}{N}\sum_{i=1}^{N}\boldsymbol{E}_i(k\boldsymbol{I} - \mathbf{1}\mathbf{1}^T)\boldsymbol{E}_i^T\,,$$

where the vectors $\boldsymbol{v}_i$ are substituted by the $(n \times k)$ matrices $\boldsymbol{E}_i$, where each one of the $k$ columns of $\boldsymbol{E}_i$ is a unit vector. The $k$ non-zero elements of $\boldsymbol{E}_i$ correspond to the $k$ elements that lie in the $i$-th drawn set $S_i$, i.e., $\boldsymbol{E}_i = \left[\boldsymbol{e}_{i_1}\middle|\boldsymbol{e}_{i_2}\middle|\ldots\middle|\boldsymbol{e}_{i_k}\right]$ for $S_i = \{i_1, i_2, \ldots, i_k\}$.

In the estimation part[7], our goal is to estimate the unknown matrix $\boldsymbol{Q}$. Using concentration results of the Fiedler eigenvalue of sums of Hermitian matrices [Tro15], using $O(\log(n)/\lambda(\boldsymbol{Q}))$ samples, one can compute an estimate $\boldsymbol{L}$ (that corresponds to the above empirical estimate) that approximates the true $\lambda(\boldsymbol{Q})$ with $(1 + \epsilon)$ multiplicative error.

We have that

$$\boldsymbol{L} = \frac{1}{N}\sum_{i=1}^{N}\boldsymbol{X}_i\,,$$

where[8] $\boldsymbol{X}_i = \boldsymbol{E}_i(k\boldsymbol{I} - \mathbf{1}\mathbf{1}^T)\boldsymbol{E}_i^T$, i.e., the estimate $\boldsymbol{L}$ is a sum of independent Hermitian matrices with $\lambda_{max}(\boldsymbol{X}_i) = \Theta(k)$. Using Proposition 7 in a similar manner as in Lemma 17, we get that

$$\Pr_{\boldsymbol{X}_{1..m}\sim\mathcal{Q}^{\otimes m}}\left[\lambda(\boldsymbol{L}) \le (1-\epsilon)\lambda(\boldsymbol{Q})\right] \le n\exp\left(-\epsilon^2 m\frac{\lambda(\boldsymbol{Q})}{2k}\right)\,.$$

Hence, it suffices to draw at least

$$m = O\left(\frac{k}{\epsilon^2 \cdot \lambda(\boldsymbol{Q})}\log(n/\delta)\right)$$

in order to guarantee the desired concentration bound with confidence at least $1 - \delta$.

**Strong Convexity of Empirical NLL w.r.t.** $\|\cdot\|_L$. Having introduced the empirical Laplacian matrix, we are able to get the desired strong convexity result for the empirical negative log-likelihood with respect to the $\boldsymbol{L}$ semi-norm. Following the analysis of [SBB$^+$16], we get that, for any vector $\boldsymbol{w} \in \mathbb{R}^d$ and parameter $\boldsymbol{z} \in \Omega_\phi$,

$$\boldsymbol{w}^T\nabla^2 L_N(\boldsymbol{z})\boldsymbol{w} \ge \frac{\lambda_2(\boldsymbol{H})}{k}\frac{1}{N}\sum_{i=1}^{N}\sum_{j=1}^{k}\mathbf{1}\{\boldsymbol{v}_i = \boldsymbol{e}_j\}\boldsymbol{w}^T\boldsymbol{E}_i(k\boldsymbol{I} - \mathbf{1}\mathbf{1}^T)\boldsymbol{E}_i^T\boldsymbol{w}\,,$$

where $\boldsymbol{H} = \beta(\phi)(\boldsymbol{I} - \mathbf{1}\mathbf{1}^T)$. Hence, we get that

$$\boldsymbol{w}^T\nabla^2 L_N(\boldsymbol{z})\boldsymbol{w} \ge \frac{\lambda_2(\beta(\phi)(\boldsymbol{I} - \mathbf{1}\mathbf{1}^T))}{k}\|\boldsymbol{w}\|_L^2\,.$$

Note that $\lambda_2(\boldsymbol{I} - \mathbf{1}\mathbf{1}^T) = 1$. [9] Consequently, the empirical negative log-likelihood is $\beta(\phi)/k$-strongly convex around the true parameters $\boldsymbol{z}^\star \in \Omega_\phi$. Using Fact 3, we get that

$$\|\boldsymbol{z}^\star - \widehat{\boldsymbol{z}}\|_L^2 \le \frac{k^2}{\beta(\phi)^2}\|\nabla L_N(\boldsymbol{z}^\star)\|_{L^\dagger}^2\,.$$

---

[7]Recall that the learning algorithm does not require the knowledge of the Laplacian matrix $\boldsymbol{Q}$. The concentration result for the empirical estimation of the matrix $\boldsymbol{Q}$ using the matrix $\boldsymbol{L}$ enables us to express our sample complexity results using the second smallest eigenvalue of the true (population) matrix $\boldsymbol{Q}$ and not its empirical estimate.

[8]For instance, let $n = 5, k = 3$ and $S = \{1, 2, 3\}$.

We have that $\boldsymbol{E} = [\boldsymbol{e}_1|\boldsymbol{e}_2|\boldsymbol{e}_3] = \begin{bmatrix} 1 & 0 & 0 \\ 0 & 1 & 0 \\ 0 & 0 & 1 \\ 0 & 0 & 0 \\ 0 & 0 & 0 \end{bmatrix}$ and $\boldsymbol{X} = \begin{bmatrix} 2 & -1 & -1 & 0 & 0 \\ -1 & 2 & -1 & 0 & 0 \\ -1 & -1 & 2 & 0 & 0 \\ 0 & 0 & 0 & 0 & 0 \\ 0 & 0 & 0 & 0 & 0 \end{bmatrix}$

[9]The characteristic polynomial of $\boldsymbol{A} = \boldsymbol{I} - \mathbf{1}\mathbf{1}^T$ is $\det(\boldsymbol{A} - \lambda\boldsymbol{I}) = (\lambda - 1)^{n-1}(\lambda + n - 1)$.

**Bounding the (expected) dual norm $\|\cdot\|_{L^\dagger}$.** Following the exact analysis of [SBB$^+$16] for the expected value of the dual norm, we get that for our setting

$$\mathbb{E}\Big[\langle \nabla L_N(\boldsymbol{z}^\star), \boldsymbol{L}^\dagger \nabla L_N(\boldsymbol{z}^\star)\rangle\Big] \le \frac{n}{N} \sup_{\boldsymbol{z}\in\mathbb{R}^k : |z_x - z_y| \le \log(\phi)} \|\nabla \log F(\boldsymbol{z})\|_2^2\,,$$

where $F$ is the Plackett-Luce function and assuming that $S = [k]$ (and $S$ satisfies the modified mass condition, i.e., $S$ lies in the support of $\mathcal{Q}$). We have that

$$\nabla_{\boldsymbol{z}} \log F(\boldsymbol{z}) = \boldsymbol{e}_i - \frac{(\exp(z_1),...,\exp(z_k))^T}{\sum_{j\in[k]}\exp(\boldsymbol{z}_j)}\,,$$

for some $i \in [k]$. Hence, we get that

$$\|\nabla_{\boldsymbol{z}}\log F(\boldsymbol{z})\|_2^2 = \left(1 - \frac{e^{z_i}}{\sum_{j\in[k]}e^{z_i}}\right)^2 + \frac{\sum_{j\in[k]\setminus\{i\}}e^{2z_j}}{(\sum_{j\in[k]}e^{z_j})^2} = \frac{(\sum_{j\in[k]\setminus\{i\}}\exp(z_j))^2 + \sum_{j\in[k]\setminus\{i\}}\exp(2z_j)}{(\sum_{j\in[k]}\exp(z_j))^2}$$

Over the optimization space $\Omega_\phi$ and since the natural parameters $\boldsymbol{z}_x = \log(\mathcal{D}(x)) \le 0$, we have that

$$\frac{1}{k\phi} \le \frac{\mathcal{D}(x)}{k\max_{j\in S}\mathcal{D}(j)} \le \frac{\mathcal{D}(x)}{\sum_{j\in S=[k]}\mathcal{D}(j)} \le \frac{\mathcal{D}(x)}{k\min_{j\in S}\mathcal{D}(j)} \le \phi/k$$

Hence, we have that

$$\left(\frac{\sum_{j\in[k]\setminus\{i\}}\mathcal{D}(j)}{\sum_{j\in[k]}\mathcal{D}(j)}\right)^2 \le ((1-1/k)\phi)^2\,,$$

and

$$\frac{\sum_{j\in[k]\setminus\{i\}}\mathcal{D}(j)^2}{(\sum_{j\in[k]}\mathcal{D}(j))^2} = \sum_{j\in[k]\setminus\{i\}} \frac{\mathcal{D}(j)}{\sum_{j\in[k]}\mathcal{D}(j)}\frac{\mathcal{D}(j)}{\sum_{j\in[k]}\mathcal{D}(j)} \le (k-1)\frac{\phi^2}{k^2}\,.$$

Finally, we get that

$$\|\nabla_{\boldsymbol{z}}\log F(\boldsymbol{z})\|_2^2 \le \phi^2 \frac{k-1}{k}\,.$$

**Conclusion of the Learning Phase in $k$-sets.** We get that we can estimate the true parameters in $\boldsymbol{L}$ semi-norm, i.e.,

$$\|\boldsymbol{z}^\star - \widehat{\boldsymbol{z}}\|_L^2 \le \frac{k^2}{\beta(\phi)^2}\frac{n}{N}\phi^2\frac{k-1}{k}.$$

Hence, we can transfer this result to an $L_2$ bound and get

$$\|\boldsymbol{z}^\star - \widehat{\boldsymbol{z}}\|_2^2 \le \frac{k^2}{\beta(\phi)^2}\frac{n}{N\lambda_2(\boldsymbol{L})}\phi^2\frac{k-1}{k}.$$

Using $N = O\Big(\frac{nk^2}{\lambda_2(\boldsymbol{L})\beta(\phi)^2}\cdot\frac{1}{\epsilon^2}\Big) =_\epsilon O\Big(\frac{nk^2}{\lambda(\boldsymbol{Q})\beta(\phi)^2}\cdot\frac{1}{\epsilon^2}\Big)$ samples from the oracle $\mathrm{Samp}_k(\mathcal{Q};\mathcal{D})$ on the support set $\mathcal{E}$, we can learn the parameter vector in $L_2$. At this point, the proof of the learning phase is similar to the $k = 2$ case: We can apply the trick of Appendix F.2 and get an $L_\infty$ bound. Hence, using Lemma 18, we can estimate the target distribution with relative error.

### G.3  Downscaling Phase for Hypergraphs

Let $\widehat{\boldsymbol{z}}$ be the estimate of the natural parameters vector and let $\widetilde{\mathcal{D}}$ the correspond distribution. We use the estimate $\widetilde{\mathcal{D}}$ of the target distribution $\mathcal{D}$ to transform the Markov chain of the $k$-set Local Sampling Scheme into another almost uniform stationary distribution, as in the case $k = 2$. Using the Bernoulli downscaling mechanism, for the pair $(x, y)$, we have that

$$\widetilde{p}_{xy} = \mathcal{D}(y)\frac{\widetilde{\mathcal{D}}(x)}{\widetilde{\mathcal{D}}(y)}\sum_{S\ni x,y}\frac{\mathcal{Q}(S)}{\mathcal{D}(S)} \approx \mathcal{D}(x)\sum_{S\ni x,y}\frac{\mathcal{Q}(S)}{\mathcal{D}(S)} = p_{yx}\,.$$

We have that

$$\frac{\widetilde{p}_{xy}}{\widetilde{p}_{yx}} = \frac{\mathcal{D}(y)\frac{\widetilde{\mathcal{D}}(x)}{\widetilde{\mathcal{D}}(y)}\sum_{S\ni x,y}\frac{\mathcal{Q}(S)}{\mathcal{D}(S)}}{\mathcal{D}(x)\sum_{S\ni x,y}\frac{\mathcal{Q}(S)}{\mathcal{D}(S)}} = \frac{\mathcal{D}(y)/\widetilde{\mathcal{D}}(y)}{\mathcal{D}(x)/\widetilde{\mathcal{D}}(x)}.$$

The modified transition matrix (as in the $k = 2$ case) has an almost uniform stationary distribution. Also, it has an absolute spectral gap of same order as the matrix $\widetilde{\boldsymbol{P}}$ that can be expressed as

$$\widetilde{\boldsymbol{P}}_{xy} = \left(\frac{1}{2} + \epsilon_{xy}\right)\sum_{S\ni x,y}\mathcal{Q}(S), \quad \widetilde{\boldsymbol{P}}_{yx} = \left(\frac{1}{2} + \epsilon_{yx}\right)\sum_{S\ni x,y}\mathcal{Q}(S),$$

and $\widetilde{\boldsymbol{P}}_{xx} = 1 - \sum_{y\neq x}\widetilde{\boldsymbol{P}}_{xy}$. Hence, we get that

$$\widetilde{\boldsymbol{P}} = \boldsymbol{I} - \frac{1}{2}\boldsymbol{Q} + \boldsymbol{Q}\circ[\epsilon_{xy}].$$

The expression of the modified transition matrix $\widetilde{\boldsymbol{P}}$ is exactly similar to the $k = 2$ case (see Equation (5)). Spectral analysis as in Lemma 15 of this matrix will result to the following properties:

(i) The transition matrix $\widetilde{\boldsymbol{P}}$ has stationary distribution $\widetilde{\pi}_0(x) = \Theta(1/n)$ for all $x \in [n]$, i.e., an *almost* uniform probability measure.

(ii) The absolute spectral gap $\Gamma(\widetilde{\boldsymbol{P}})$ of $\widetilde{\boldsymbol{P}}$ and the minimum non-zero eigenvalue $\lambda(\boldsymbol{Q})$ of the Laplacian matrix $\boldsymbol{Q}$ satisfy $\Gamma(\widetilde{\boldsymbol{P}}) = \Omega(\lambda(\boldsymbol{Q}))$.

(iii) The mixing time of the transition matrix $\widetilde{\boldsymbol{P}}$ is $T_{\mathrm{mix}}(\widetilde{\boldsymbol{P}};1/4) = O(\log(n)/\lambda(\boldsymbol{Q}))$, where $\lambda(\boldsymbol{Q})$ is the minimum non-zero eigenvalue of the Laplacian matrix $\boldsymbol{Q}$.

### G.4 CFTP Phase for Hypergraphs

The structure of the $k$-set algorithm remains similar to the $k = 2$ case, with the exception that the drawn samples are now hyperedges. We continue with the modified Algorithm 7. Recall that, for each drawn sample $(S, \boldsymbol{v}) \sim \mathrm{Samp}_k(\mathcal{Q};\mathcal{D})$, the vector $\boldsymbol{v} \in \{\boldsymbol{e}_1, ..., \boldsymbol{e}_n\}$ is the indicator vector of the winning node of the set $S \subseteq \binom{[n]}{k}$.

### G.5 Conclusion

Let $N_{\mathrm{Learn}}$ and $N_{\mathrm{CFTP}}$ be the expected number of samples required for a single execution of the learning algorithm for $k$-sets and of the parameterized CFTP algorithm respectively (see Algorithm 7). Under Assumption 1 (for the distribution $\mathcal{Q}$ over hyperedges) and the modified version of Assumption 2, the expected sample complexity of the generalized algorithm for the $k$-set case is

$$N_{\mathrm{Learn}}(\epsilon := 1/\sqrt{n}) + \Theta(n)\cdot N_{\mathrm{CFTP}} = O\left(\frac{k\log(n)}{\lambda(\boldsymbol{Q})} + \frac{n^2k^2}{\lambda(\boldsymbol{Q})\beta(\phi)^2}\right) + \widetilde{O}\left(\frac{n^2}{\lambda(\boldsymbol{Q})}\right).$$

Note that $\beta(\phi)$ is constant under Assumption 2. As explained in [SBB$^+$16], it is natural to consider the regime where $k = O(1)$. In this regime, the learning bounds match the CFTP dependence on $n$ under $\widetilde{O}$ notation.

**Corollary 21** (Exact Sampling from $k$-Set LSS). *For any positive constant integer $k > 2$, under Assumption 1 (for the distribution $\mathcal{Q}$ over $k$-sets) and Assumption 2, there exists an algorithm (Algorithm 7) that draws an expected number of $\widetilde{O}\left(\frac{n^2k^2}{\lambda(\boldsymbol{Q})}\right)$ samples from a $k$-Set Local Sampling Scheme $\mathrm{Samp}_k(\mathcal{Q};\mathcal{D})$, and generates a sample distributed as in $\mathcal{D}$.*

---

**Algorithm 7** Exact Sampling from $k$-set Local Sampling Schemes

---

1: **procedure** EXACTSAMPLER-$k$-SET($\boldsymbol{p}$)  ▷ *Sample access to the LSS oracle* $\mathrm{Samp}_k(\mathcal{Q};\mathcal{D})$.
2:  $t \leftarrow 0$
3:  $F_0(x) \leftarrow x$, for any $x \in [n]$
4:  **while** $F_t$ has not coalesced **do**  ▷ *While no coalescence has occured.*
5:   $t \leftarrow t - 1$
6:   Draw $(S, \boldsymbol{v})$ where $S \in \mathcal{E}$ and $\boldsymbol{v} = \boldsymbol{e}_i$ for some $i \in [n]$
7:   $y \leftarrow$ position where $\boldsymbol{v}[y] = 1$  ▷ *Find the winning node $y \in S$.*
8:   **for** $x = 1 \ldots n$ **do**  ▷ *In order to update state $x$.*
9:    **if** $x \notin S$ or $x = y$ **then** $F_t(x) \leftarrow F_{t+1}(x)$
10:    **else**

$$F_t(x) \leftarrow \begin{cases} F_{t+1}(w), & \text{with probability } \min\{p(x)/p(w), 1\} \\ F_{t+1}(x), & \text{otherwise} \end{cases}$$

11:   **end**
12:  **end**
13:  Draw $C \sim \mathrm{Be}(p(F_t(1)))$  ▷ *Remove bias using rejection sampling.*
14:  **if** $C = 1$ **then** Output $F_t(1)$ **else** Output $\perp$  ▷ *Output the perfect sample or reject.*
15: **end procedure**

16: **procedure** EXACTSAMPLER-$k$-SET-WITHLEARNING($\delta$)  ▷ *The algorithm of Corollary 21.*
17:  $\widetilde{\mathcal{D}} \leftarrow$ LEARN($\epsilon := 1/\sqrt{n}, \delta$)  ▷ *Learn $\mathcal{D}$ in relative error as in Appendix G.2.*
18:  x $\leftarrow \perp$
19:  **while** x $= \perp$ **repeat**
20:   x $\leftarrow$ EXACTSAMPLER-$k$-SET($\widetilde{\mathcal{D}}$)
21:  Output x
22: **end procedure**

---