# OpenReview forum: "Perfect Sampling from Pairwise Comparisons"
_NeurIPS.cc/2022/Conference — NeurIPS 2022 Accept_

### Official Review · Reviewer_RUQw · 2022-06-30

**Rating:** 8
**Confidence:** 2
**Soundness:** 4 excellent
**Presentation:** 3 good
**Contribution:** 3 good

**Summary:**

The paper studies the problem of obtaining perfect samples from a discrete distribution with access to pairwise comparisons of elements. More specifically, let $\mathcal{D}$ be a distribution on a finite set $X$ and $\mathcal{Q}$ be a distribution on a subset of $X\times X$. Accessible samples are generated as below. A pair $(x,y)$ is first sampled from $\mathcal{Q}$ and then an element $w\in\{x,y\}$ is returned, where $w=x$ with probability $\mathcal{D}(x)/(\mathcal{D}(x)+\mathcal{D}(y))$ and $w=y$ with probability $\mathcal{D}(y)/(\mathcal{D}(x)+\mathcal{D}(y))$. The task is to sample an element from $\mathcal{D}$ using as few sample triples $(x,y,w)$ as possible.

The paper solves this problem with $\tilde{O}(n^2/\lambda(Q))$ samples of the triples, here $n = |X|$, $Q$ is a matrix defined by $Q_{xy} = -\mathcal{Q}(x,y)$ and $Q_{xx} = \sum_{y\neq x} \mathcal{Q}(x,y)$ and $\lambda(Q)$ is the second smallest eigenvalue of $Q$.

The major technical innovation is a new variant of the Coupling from the Past (CFTP) algorithm that allows the perfect sampling with $\tilde{O}(n\log(1/\mathcal{D}_{\min})/\Gamma(M))$ samples, where $M$ is the transition matrix of a Markov chain (depending on both $\mathcal{Q}$ and $\mathcal{D}$) and $\Gamma(M)$ the spectral gap from 1. To remove the dependence on $\mathcal{D}$, the authors use $\tilde{O}(n^2/\lambda(Q))$ samples to estimate the density function of $\mathcal{D}$ up to an $(1\pm 1/\sqrt{n})$-factor (another main technical contribution of the paper) and then modify the new CFTP algorithm to accommodate having just an approximation to $\mathcal{D}$ instead of the exact distribution.

**Questions:**

The full version of the paper is well-written. The following are some minor points:
- Lines 72-79: it is not clear at this point whether you are getting a comparison element w in {x,y}
- Line 88: in L_2 norm -> in the L_2 norm
- Line 217: give the full name of TCS
- Algorithm 1, line 4: coallesced -> coalesced
- Algorithm 1, line 12: define Be(...) (the definition only appears in Line 334, which is too late)
- Line 309, why do you need to take the max? Doesn’t |1-f|<=eps imply |1-1/f| <= eps? (after rescaling eps)
- Supplementary, Algorithm 4, line 4: coallesced -> coalesced
- Supplementary, line 816: seems that w=1 and w=0 are inverted? Staying at node i should correspond to w=0 and moving to j correspond to w = 1.
- Supplementary, lines 877, 1152, 1172: missing the word ‘Proof’ at the beginning of the line
- Supplementary, Algorithm 5, line 7: shouldn’t it be just output z’? |z-z’|_infty <= eps is the guarantee. It is also not clear what z is at this point unless one reads Appendix F.2.


**Limitations:**

N.A.

**Strengths And Weaknesses:**

Strengths: Technically strong, solid theoretical results. Obtained precise sampling not just approximate sampling.

Weaknesses: I don’t see weakness in the results per se. It would be better if the authors can show a lower bound, say, why Omega(1/\lambda(Q)) samples are necessary.

Additional comments: This reads like a TCS paper and probably fits better at a more theoretical conference like COLT. This submission was reorganised from a longer paper to fit within NeurIPS's strict page limits, and as a result, it is incoherent in presentation at various places.  For example, the definitions and notations are relegated to the appendix, making the paper difficult to read; Section 2 refers to Algorithm 4, which also serves as a foundation for the main Algorithm 1 but is relegated to the appendix; a main technical theorem (Theorem 4) refers to Algorithm 5, which is also relegated to the appendix. If the paper is to be accepted, it must be thoroughly reorganised into a more coherent form within the page constraints.

---

> ### Author Response · Authors · 2022-08-02
> **Response to Reviewer RUQw**
>
> We thank the reviewer for the positive feedback and the provided comments and corrections. We will fix all the mentioned minor issues in the first revision of our work. Moreover, as the reviewer mentions, we chose to refer to the Appendix for the notation and some preliminaries and algorithms due to space constraints. We are going to reorganize the material towards a more coherent presentation, as the reviewer proposes. We would like to respond to the reviewer's questions and comments, as follows:
>
> > *Weaknesses: I don’t see weakness in the results per se. It would be better if the authors can show a lower bound, say,why Omega(1/\lambda(Q)) samples are necessary.*
>
> We would like to mention that the dependence on $1/\lambda(Q)$ is expected and intuitively unavoidable. This term is connected to the mixing time. We will add a discussion in the paper. Let us become more specific. The (sample) complexity of our proposed method is $ \widetilde{O}(n^2/\lambda(Q))$. The term $1/\lambda(Q)$ appears due to the mixing time of the (transformed) Markov chain. The complexity of CFTP (Zahavy et al. 2020) is $\widetilde{\Theta}(n T_{mix}).$ Hence, any CFTP-based exact sampler (or more broadly random walk based algorithm) will have that kind of spectral dependence.
>
> Similarly, a dependence on the support’s size $n$ is necessary in general. Note that the structure of the target distribution D is crucial: if the target is uniform (or generally “flat”), then a bound of order $\widetilde{O}(n /\lambda(Q))$ is obtained by the naive CFTP method; for distributions with modes, our algorithm achieves quadratic dependence on $n$. However, as we mention in the conclusion of our paper, it is not clear how to obtain a lower bound against **any** possible exact sampling algorithm. The sample complexity of a potential (not necessarily random-walk based) exact sampling algorithm will depend on $Q$ (since it draws samples from $Q$ and $D$ ’s support should be observed at least once), but might achieve a sample complexity not directly dependent on $\lambda(Q)$. The unclear part is our algorithm’s dependence on $n$. While our algorithm attains a bound of $\widetilde{O}(n^2)$, it is not obvious whether an algorithm with sample complexity $\widetilde{O}(n)$ exists. It is moreover quite unclear how to obtain a potential $\widetilde{\Omega}(n^2)$ against any potential sampling algorithm.
>
> As a result, we believe that the main open question left for future work from our result is whether the quadratic dependence on the support’s size $n$ can be reduced to $\widetilde{O}(n)$. In this direction, it is interesting to ask whether there are efficient ways to transform the Markov chain we are dealing with in order to get a faster perfect sampling algorithm. Our learning method finds a vector $\vec p$ (input of our modified CFTP Algorithm) that makes the re-weighted distribution close to uniform. Is it possible to find another $\vec q$ (without our learning algorithm) such that (i) when transforming the chain with $\vec q$, we obtain a fast mixing chain; and (ii) the success probability in the rejection sampling step will be sufficiently high? This may lead to an improved sample complexity bound. To summarize, even though getting a matching lower bound is an interesting pursuit for future work, we believe that our algorithmic results require various new ideas and constitute a good addition to the existing literature.

---

> > ### Comment · Reviewer_RUQw · 2022-08-03
> > **Thanks for the response**
> >
> > It is a nice response, and it would be good to include this discussion in the paper. Thanks for the interesting paper!

---

### Official Review · Reviewer_PftA · 2022-07-09

**Rating:** 7
**Confidence:** 4
**Soundness:** 4 excellent
**Presentation:** 4 excellent
**Contribution:** 3 good

**Summary:**

The paper studies perfect sampling from pairwise comparisons. The setup is the following:

There is an unknown distribution D supported on [n], and the goal is to produce a sample from this distribution. The access to D is via 'pairwise comparisons'. There is a distribution Q supported on pairs [n]x[n], and the algorithm observes samples of the following form: first, one draws (x, y) from Q and then w is drawn according to D conditioned on it being either x or y. (A small remark: the algorithm doesn't need to know the distribution Q, but the complexity guarantees will depend on the structure of Q).

At a very high level, the line of work this explores is algorithmic problems on distributions when access model is not the standard independent samples. One could imagine settings where pairwise comparisons are the natural way one interacts with a distribution -- while the authors don't quite provide specific examples, they point to previous work exploring these kinds of questions. For instance, the question of learning a distribution in the same model was studied by [SBB+16] (and will also play a prominent role in the algorithm presented).

Very Straight-forward Approach: There is a very natural way to produce a sample from a distribution which is very close to D via building a natural Markov chain. One starts at an arbitrary element z in [n], and starts drawing samples of the from ({x,y}, w). If z is one of x or y, then we transition to w. The stationary distribution of this Markov chain is D, so given enough steps, one gets a draw from a distribution extremely close to D. The time it takes to produce a sample which is eps-close to D is essentially the time it takes for the Markov chain to mix, which depends on the structure of the Markov chain, which is specified by Q and D. Note that {x, y} is drawn from Q, and w from D conditional on {x,y}, so if M is the Markov chain, the time is proportional to 1 / Gamma(M), where Gamma(M) is the spectral gap of the Markov chain.

Straight-forward Approach: The problem with the approach above is that it is only produces samples from an approximately close distribution. There is actually a technique from Markov chains to produce perfect samples, called 'coupling from the past' (CFTP), which essentially runs n Markov chains backwards until they are all in one element ('coalesced'). Using this technique one can get a perfect sample in n / Gamma(M) (from my understanding, the factor n slowdown is due to the fact one considers n Markov chains?).

There is one important aspect about the above two approaches which falls short. The running time depends on the spectral gap of the Markov chain M, which depends on Q as well as the unknown target distribution D. Furthermore, the paper provides a relatively natural and simple examples where the resulting Markov chain M takes exponential running time in n in order to mix, and applying the above two natural approaches would lead to algorithms running in exponential time. One natural question would be to have the computational complexity depend solely on the structure of comparisons (via the distribution Q), but not the unknown distribution D. This is what this work achieves.

They give an algorithm which produces a perfect sample from D. The number of pairwise comparison samples is n^2 times 1 / Gamma(Q), where Gamma(Q) is the spectral gap of the distribution Q. The complexity is now worst-case for any unknown distribution D.

The starting point is the prior work of [SBB+16] on learning D, where an algorithm is presented for learning with n / (eps^2 Gamma(Q)) samples up to small error eps. The main insight of this work is that one may use the learning algorithm to later speed up the sampling algorithm. The idea proceeds as follows: one first learns the distribution D using prior work and small eps. Suppose that ~D is the learned approximation to D. The algorithm uses ~D to bias the Markov chain M to a different Markov chain M' which is close to the behavior of Q (and now mixing time will depend on Gamma(Q)). However, since the bias introduced by ~D is known, one can use a rejection sampling step to remove this bias at the cost of a factor n more samples.

**Questions:**

Is it possible to give a concrete application of PC in the paper? I would imagine this would help the presentation of the paper. While I could artificially imagine some instances, I'm not aware of an application where your new algorithm could be useful.

**Limitations:**

The authors addressed this.

**Strengths And Weaknesses:**

Strengths:

It is very nice that the paper starts with a very straight-forward approach which obtains undesirable guarantees, and shows that different (but similar) techniques can get different algorithms. The paper is very well-written.

Weaknesses:

I don't see any major weakness.

---

> ### Author Response · Authors · 2022-08-02
> **Response to Reviewer PftA**
>
> We thank the reviewer for the positive feedback concerning our work and the presentation of the paper. We would like to respond to the reviewer's questions and comments, as follows:
>
> > *Is it possible to give a concrete application of PC in the paper? I would imagine this would help the presentation of the paper. While I could artificially imagine some instances, I'm not aware of an application where your new algorithm could be useful.*
>
> Our approach could be seen as a general methodology for boosting the convergence of various random walk procedures. In applications where either (i) the transitions are essentially unknown but learning the underlying distribution is feasible or (ii) the transitions are known (and no learning is needed), then one can use our technique to smoothen the landscape of the random walk and obtain faster convergence results. For instance, we believe that such re-weighting and rejection sampling ideas (whose motivation comes from the simulated tempering technique) are applicable even for approximate sampling algorithms (e.g., when the (known) target distribution is far from being uniform, some speed-up might be possible).
>
> A different perspective of our results comes from the field of truncated statistics, where a fundamental task is to design efficient procedures that remove the truncation/censoring bias from the data. Our algorithm can be considered as a way to “clean up” the bias induced from pairwise comparisons (by viewing the various induced conditional distributions as truncations of the original target distribution).
>
> A potential speculation is that this approach could be used in general settings where there exists some kind of bias in the data and not necessarily truncation (e.g., assume that we observe samples from an Ising model and some coordinates are missing). If one could construct a natural Markov chain, she could remove the induced bias potentially in an efficient way by using our re-weighting technique.
>
> The pairwise comparisons (or in general $k$-wise comparisons) setting appears in natural applications such as recommendation systems and web advertisement. In many real-life applications, obtaining approximate samples from the actual distribution is a fundamental problem and it may be preferred over perfect simulation. Our exact sampling approach may nevertheless have some advantages that motivate its real-world applicability. In the approximate sampling approach, there is no termination criterion, i.e., the Markov chain has to be run for sufficiently long time and this requires an a priori known bound on the chain's mixing time, which may be much larger than necessary. Perfect sampling algorithms come with a stopping rule and this termination condition can be attained well ahead of the worst-case analysis time bounds in practice.
>
> In general, we believe that the CFTP/sampling community will find our parameterized approach as an interesting extension and addition to the current literature.

---

### Official Review · Reviewer_y8bp · 2022-07-11

**Rating:** 8
**Confidence:** 3
**Soundness:** 4 excellent
**Presentation:** 4 excellent
**Contribution:** 4 excellent

**Summary:**

This paper considers perfect sampling from a discrete set D given data in the form of pairwise comparisons. A pair of points is sampled from the discrete set, but only the winner of the comparison is revealed (where the winner is distributed according to the conditional distribution over the two points). The goal is to generate a sample from D given only the results of the pairwise comparisons. To develop their sampling scheme, the authors use Coupling From the Past (CFTP), which is a general technique for exact sampling from the stationary distribution of a Markov chain. Unfortunately, CFTP depends on the landscape of D, and can therefore have a poor runtime. To improve on the basic implementation, the authors modify the CFTP strategy. Specifically, the transitions are modified so that the corresponding Markov chain has a uniform distribution. Then, rejection sampling is used to re-weight the sample in order to return to the target distribution over D.

**Questions:**

•	Do you have ideas where your modified CFTP method could be useful? Even speculation would be interesting!

•	Your CFTP method requires an estimate of the target stationary distribution. What if the stationary distribution is known up to a normalizing factor?

•	pg. 8: how can we ensure that the modified transition probabilities correspond to a valid probability distribution? Please show this explicitly.

**Limitations:**

Yes

**Strengths And Weaknesses:**

•	This paper has some significant novel ideas, notably the modified CFTP scheme. My impression prior to reading this paper was that CFTP is practical only with monotone chains, but this paper potentially widens the applicability of the technique.

•	The writing is very good; the paper was a pleasant read!

•	No experimental evaluation; would be nice to include to get an idea of how practical this is.

•	pg. 8: The beginning of the proof sketch of Lemma 5 could be broken up into more sentences.

---

> ### Author Response · Authors · 2022-08-02
> **Response to Reviewer y8bp**
>
> We thank the reviewer for the positive feedback and for appreciating our results and presentation. In the first revision of our work, we will improve the readability of the proof of Lemma 5, which was compact due to space constraints. We would like to respond to the reviewer's questions and comments, as follows:
>
> > *No experimental evaluation; would be nice to include to get an idea of how practical this is.*
>
> We agree that some experimental evaluation would be interesting. While our motivation was originally theoretical, we will update the current version and add experimental evaluation to see how practical our approach is compared to the standard CFTP method.
>
> > *Do you have ideas where your modified CFTP method could be useful? Even speculation would be interesting!*
>
> Our approach could be seen as a general methodology for boosting the convergence of various random walk procedures. In applications where either (i) the transitions are essentially unknown but learning the underlying distribution is feasible or (ii) the transitions are known (and no learning is needed), then one can use our technique to smoothen the landscape of the random walk and obtain faster convergence results. For instance, we believe that such re-weighting and rejection sampling ideas (whose motivation comes from the simulated tempering technique) are applicable even for approximate sampling algorithms (e.g., when the (known) target distribution is far from being uniform, a potential speed-up may be possible).
>
> A different perspective of our results comes from the field of truncated statistics, where a fundamental task is to design efficient procedures that remove the truncation/censoring bias from the data. Our algorithm can be considered as a way to “clean up” the bias induced from pairwise comparisons (by viewing the various induced conditional distributions as truncations of the original target distribution).
>
> A potential speculation is that this approach could be used in general settings where there exists some kind of bias in the data and not necessarily truncation (e.g., assume that we observe samples from an Ising model and some coordinates are missing). If one could construct a natural Markov chain, she could remove the induced bias potentially in an efficient way by using our re-weighting technique.
>
> In general, we believe that the CFTP/sampling community will find our parameterized approach an interesting contribution.
>
> > *Our CFTP method requires an estimate of the target stationary distribution. What if the stationary distribution is known up to a normalizing factor?*
>
> The normalization constant does not appear in the transition probabilities. In this case, one could transform the target distribution to the uniform probability measure (exactly) without using learning and repeat the same approach. However, usually, the cases where the normalizing factor is not known correspond to scenarios where the partition function is hard to compute (even to approximate) and, hence, some extra care should be taken in order to have an efficient approach. As the reviewer comments, in such settings there is some helpful underlying structure (e.g., in Ising models where monotone CFTP applies), which allows for efficiency.
>
> > *How can we ensure that the modified transition probabilities correspond to a valid probability distribution? Please show this explicitly.*
>
> Assume that $D$ is the actual unknown distribution and $\widetilde{D}(x)$ is the estimate for $D(x)$. The rescaling mechanism for the pair $(x, y)$ with $D(x) > D(y)$ works as follows: it lets the mass at the transition $x \to y$ unchanged (the transition occurs with probability $p[x \to y] = D(y)/(D(x)+D(y))$) and it downscales the weight of the (most probable) transition $y \to x$ with $\widetilde{D}(y)/\widetilde{D}(x)$.
> This operation gives rise to new transition probabilities where
> $\widetilde{p}[ x \to y] = p[x \to y] = D(y)/(D(x)+D(y)) $ and
> $\widetilde{p}[y \to x] = D(x)/(D(x)+D(y)) \cdot \widetilde{D}(y)/\widetilde{D}(x) \approx  p[x \to y]$.
> Observe that the new Markov chain remains ergodic and so it admits a unique stationary distribution $\widetilde{\pi}$. The induced probability measure at $x$ will have mass
> $
> \widetilde{\pi}(x) = \frac{ D(x)/\widetilde{D}(x) }{  \sum_{y \in [n]} D(y)/\widetilde{D}(y) }.
> $
> Note that if the estimate $\widetilde{D}$ is exact, we obtain the uniform distribution, while, when we have some noise in the estimation, $\widetilde{\pi}$ should be close to the uniform probability measure. The intuition is that we make each pair of transitions $(x \to y, y \to x)$ relatively balanced.

---

> > ### Comment · Reviewer_y8bp · 2022-08-08
> > **Reply**
> >
> > Thank you for your comments! The ideas for further applications of the CFTP technique are interesting.

---

### Official Review · Reviewer_BvYp · 2022-07-12

**Rating:** 4
**Confidence:** 3
**Soundness:** 2 fair
**Presentation:** 2 fair
**Contribution:** 2 fair

**Summary:**

This paper gives an algorithm to sample a distribution given access to pairwise comparisons from it. The technical contribution is to design a Markov chain whose stationary distribution matches the unknown one and then use the coupling from the past algorithm to obtain exact samples. The sample complexity of the resulting algorithm may depend on the structure of the distribution. Therefore the authors give another algorithm for sampling from a parametric Markov chain whose stationary distribution approximately matches the unknown one and which mixes significantly faster given a good approximation for the stationary distribution. They obtain this algorithm using efficient learning from the pairwise comparisons algorithm.

**Questions:**

None.

**Strengths And Weaknesses:**

Strength: The problem is important, the results are original, and the paper is clearly written.

Weakness:
- The paper lacks any empirical verification, especially regarding how efficient the improved algorithm is in practice.
- It would have been great to see the upper bound being matched by a lower bound.

---

> ### Author Response · Authors · 2022-08-02
> **Response to Reviewer BvYp**
>
> We thank the reviewer for recognizing the importance of the problem discussed in the paper and the fact that the work is clearly presented and written. We would like to respond to the reviewer's questions and comments, as follows:
>
> > *The paper lacks any empirical verification, especially regarding how efficient the improved algorithm is in practice.*
>
> We would like to emphasize that the main purpose of this paper is theoretical. In this work, we design a perfect sampling algorithm when the input is biased from pairwise comparisons by modifying the well-known CFTP algorithm. Indeed, an experimental evaluation of our approach compared to the standard CFTP algorithm would be very interesting and is definitely a direct future research direction. In the first revision of our work, we will provide experimental evaluation to see how practical our approach is compared to the standard CFTP method, as the reviewer suggests.
>
> > *It would have been great to see the upper bound being matched by a lower bound.*
>
> This is indeed a very interesting point that we mention in the conclusion of our work. Our work focuses on the upper bound - algorithm design; however, we agree with the reviewer that a general lower bound would be an interesting addition. The (sample) complexity of our proposed method is $ \widetilde{O}(n^2/\lambda(Q))$.  The term $1/\lambda(Q)$ appears due to the mixing time of the (transformed) Markov chain. The complexity of CFTP (Zahavy et al. 2020) is $\widetilde{\Theta}(n T_{mix})$. Hence, any CFTP-based exact sampler (or more broadly random-walk based algorithm) will have that kind of spectral dependence. Similarly, a dependence on the support’s size $n$ is necessary in general. Note that the structure of the target distribution D is crucial: if the target is uniform (or generally “flat”), then a bound of order $ \widetilde{O}(n /\lambda(Q))$ is obtained by the naive CFTP method; for distributions with modes, our algorithm achieves quadratic dependence on $n$. However, as we mention in the conclusion of our paper, it is not clear how to obtain a lower bound against **any** possible exact sampling algorithm for this $n^2$ term. We remark that the sample complexity of a potential (not necessarily random-walk based) exact sampling algorithm will depend on $Q$ (since it draws samples from $Q$ and $D$ ’s support should be observed at least once), but might achieve a sample complexity not directly dependent on $\lambda(Q)$.
>
> As a result, we believe that the main open question left for future work from our result is whether the quadratic dependence on the support’s size $n$ can be reduced to $\widetilde{O}(n)$. In this direction, it is interesting to ask whether there are efficient ways to transform the Markov chain we are dealing with in order to get a faster perfect sampling algorithm. Our learning method finds a vector $\vec p$ (input of our modified CFTP Algorithm) that makes the re-weighted distribution close to uniform. Is it possible to find another vector $\vec q$ (without our learning algorithm) such that (i) when transforming the chain with $\vec q$, we obtain a fast mixing chain; and (ii) the success probability in the rejection sampling step will be sufficiently high? This may lead to an improved sample complexity bound. To summarize, even though getting a matching lower bound is an interesting pursuit for future work, we believe that our algorithmic results require various new ideas and constitute a good addition to the existing literature.

---

### Meta-Review · Area_Chair_ZLNc · 2022-08-24

**Recommendation:** Accept
**Confidence:** Certain

**Metareview:**

The authors show how to perfectly sample a discrete distribution, given sample access to the Bradley-Terry-Luce model on subsets of size 2. While the learning problem has been previously studied extensively, this work initiates the study of sampling. Technically, the authors introduce re-weighting and rejection sampling ideas that speed up coupling from the past by utilizing an approximate learning algorithm; these techniques could be useful in other applications, as the authors hint in the rebuttal.

The reviewers agreed that the paper is technically quite strong and that it's quite well written. The authors responded to all remaining questions by the reviewers, clearing the path to the paper being accepted.

**Award:**

No

---

### Decision · Program_Chairs · 2022-09-14

Accept